

# Unravelling the impacts of precipitation, temperature and land-cover change for extreme drought over the North American High Plains

Annette Hein[1], Laura Condon[2], Reed Maxwell[1]

[1]Department of Geology and Geological Engineering, Colorado School of Mines, Golden, 80401, USA

[2]Department of Hydrology and Atmospheric Science, University of Arizona, Tucson, 85721, USA

*Correspondence to*: Annette Hein (ahein@alumni.mines.edu)

**Abstract.** Drought is a natural disaster that may become more common in the future under climate change. It involves changes to temperature, precipitation, and/or land cover, but the relative contributions of each of these factors to overall drought severity is not clear. Here we apply a high-resolution integrated hydrologic

model of the High Plains to explore the individual importance of each of these factors and the feedbacks between them. The model was constructed using ParFlow-CLM, which represents surface and subsurface processes in detail with physically based equations. Numerical experiments were run to perturb vegetation, precipitation and temperature separately and in combination. Results show that decreased precipitation caused larger anomalies in evapotranspiration, soil moisture, stream flow and water table levels than did

increased temperature or disturbed land cover. However, these factors are not linearly additive when applied in combination; some effects of multi-factor runs came from interactions between temperature, precipitation and land cover. Spatial scale was important in characterizing impacts, as unpredictable and nonlinear impacts at small scales aggregate to predictable, linear large scale behaviour.

## 1 Introduction

Improved understanding of drought is important to sustainably manage water resources and agricultural production worldwide. Agriculture depends on rainfall, especially in arid and semiarid regions, so large droughts can devastate global agriculture. As climate continues to change, droughts may happen more often and be worse than they are now (IPCC 2014; Diffenbaugh et al. 2017). Within the United States, the High Plains is a key agricultural region that is also drought-prone. Drought affected that region on many

occasions during the 20th century, including the Dustbowl of the 1930s (Hong and Kalnay 2000, Schubert et al. 2004) and the more recent 2012 drought that dried soils and lowered crop yields across most of the area (Otkin et al., 2012). Forecasts for the High Plains predict similar or worse drought in the future (Cook et al., 2015) that could result in significant declines in crop yields (Glotter and Elliot, 2016). In the past, groundwater pumping has been used to buffer the region against drought impacts, but it is becoming

depleted (McGuire 2017). A better understanding of drought gained from modeling studies will be valuable for meeting future sustainability challenges.



Two important factors that cause drought are decreased precipitation and increased temperature (Loon 2015), with vegetation acting as a subsidiary factor. Drought occurs when these changes impact evapotranspiration (ET) and soil moisture levels within a watershed. When precipitation decreases, less

water is available for any part of the water cycle including ET. If the system is already wet (energy limited), this change may cause only minor impacts if the remaining water is still sufficient to supply potential ET. If the system is water limited, then a decrease in precipitation will cause ET to decrease and soil moisture to dry. Some of the energy previously used to evaporate the water (latent heat of phase change) will instead go to sensible heat, causing a shift in the Bowen ratio (Eltahir 1998, Seniviratne 2010).

This change in the surface energy balance can carry over into atmospheric instability and changes to circulation (Eltahir 1998) creating feedbacks to meteorology (Brubaker 1996) at a variety of timescales (Betts 1996). In the present study area of the High Plains, an ensemble of climate models found a strong connection between soil moisture and the atmosphere (Koster 2004).

In contrast to the precipitation decrease, a temperature increase causes drought more indirectly, through an

increase in potential ET. In an energy limited system, the available water will supply a higher actual ET (McEvoy 2016). In a water limited system, the increase in ET is bounded by the available water. Initially, ET can still increase, but as the soils dry ET is eventually expected to drop. This is the opposite direction of the effect predicted for precipitation decrease, so in a drought where both occur, there will be a competing effect (Livneh and Hoerling 2016). If vegetation is disturbed, its buffering effect on the system is removed.

Vegetation is expected to have a buffering effect against impacts to ET because it can reach deeper sources of water to satisfy ET demands (Maxwell and Condon 2016).

Many studies have used models to explore the driving factors and possible effects of future droughts. Otkin et al. (2012) examined US Department of Agriculture metrics and Noah, Mosaic and Variable Infiltration Capacity (VIC) models to show that hot and dry conditions in the 2012 drought dried High Plains soils

within a few months. Gosling et al. (2017) used an ensemble of local and global hydrologic models and a variety of climate change scenarios to conclude there was no definite prediction for runoff in the upper Mississippi basin. Crosbie et al. (2012) also found no definite prediction for recharge in the High Plains under scenarios from 16 global climate models. Chien et al. (2012) predicted with a Soil Water Assessment Tool (SWAT) model that streamflow in Illinois watersheds will decrease under climate change. Naz et al.

(2016) modeled hydrologic response to climate change across the entire continental US with a VIC model. They found large regional differences in runoff, SWE, and rain-to-snow ratio across the country under various Climate Model Intercomparison Project 5 model projections. Meixner et al. (2015) reviewed



studies across 8 representative aquifers in the United States to anticipate effects of climate change on recharge. Recharge increased slightly in the northern High Plains, and decreased in the south.

Modelling studies typically include some combination of temperature, precipitation and land cover changes as forcing factors to drought. However, since the preceding studies are either reconstructing a natural event or forecasting future droughts, they involve many simultaneous changes in forcing variables. Although the broad theoretical importance of each variable is clear, multiple simultaneous changes in one study obscure the details of exact mechanisms or interactions between factors. To address this limitation, other studies have taken the approach of isolating and comparing factors using numerical experiments instead of reconstructing real-world events.

Livneh and Hoerling (2016) argued that precipitation was more important than temperature in causing drought impacts in the High Plains based on results from historical reconstruction and sensitivity experiments using VIC and the Unified Land Model (ULM). Maxwell and Kollet (2008) ran a ParFlow-CLM model over the Little Washita basin in Oklahoma and found that a 2.5 degree C temperature increase reduced saturation and potential recharge. If precipitation decreased, effects were much more extreme than if temperature increased alone. Groundwater tables near the land surface allowed local regions of the model to maintain saturation and potential recharge regardless of the climate perturbations. The studies summarized here suggest that precipitation changes typical of observed droughts outweigh the effects of typical temperature or land cover changes in water limited systems. However, if precipitation is stable, these secondary factors can be important; and in any case they may mitigate or exacerbate the effect of precipitation.

Existing studies have identified precipitation and temperature as important factors relating to drought, with vegetation changes as a secondary control. The studies reviewed here often reconstruct historical events, which does not allow for isolation of individual factors and their effects. A few studies have focused on individual factors with a numerical experiment approach; this study does the same, but uses a more advanced and flexible modeling code to ensure that the results are as physically based as possible. In this study we apply an integrated hydrologic model, ParFlow-CLM, to run numerical experiments across the southern High Plains. ParFlow is ideal for this study because it is a fully integrated hydrologic model that includes groundwater, soil moisture and overland flow processes. This allows a more detailed exploration than other tools that have been used in similar studies that rely on more simplified or compartmentalized approaches. In particular, the project addresses three specific questions:

1) What is the relative importance of precipitation, temperature and land cover change in response of ET, runoff and soil moisture to drought?


2) How do the hydrologic impacts of precipitation, temperature and land cover change differ when driving factors are considered together rather than in isolation?

3) How do impacts of the main drought factors and their interactions change across spatial scales?

## 2 Methods

This study explores how temperature, precipitation and land cover affect the water and energy balance of the High Plains through a series of numerical experiments where the driving factors are systematically perturbed.  The scenarios developed for these numerical experiments were modeled after an example of extreme drought in the region, the Dustbowl of the 19390s.  The goal of the study is not to reconstruct the Dustbowl or produce operational forecasting, rather to exploit the capabilities of large-scale modeling to

illuminate major features of the hydrologic system using a real world extreme drought as a test case.

The numerical simulations were conducted with ParFlow-CLM, an integrated hydrologic model. ParFlow-CLM was selected because employs a more extensive and physically based representation of subsurface processes than many other hydrologic models and is therefore well suited the water and energy dynamics that occur during drought. Here we provide more details on the modeling platform (2.1), study

domain (2.2),  drought scenarios (2.3) and metrics of analysis (2.4). Selected model inputs and outputs are presented in the model data (Hein et al 2018) on the Harvard Dataverse.

### 2.1 Model Selection

The model was constructed using ParFlow, an integrated hydrologic modeling code, coupled to the Common Land Model (CLM), a land surface modeling code. ParFlow is an integrated hydrologic model

with an explicit description of most physical processes of the water balance. It models saturated and unsaturated flow in three dimensions using Richards' Equation, with flux defined by Darcy's Law and relationships between pressure and relative saturation or permeability defined by the Van Genuchten functions. Overland flow is modeled with the kinematic wave equation, with velocity found by Manning's equation.  Energy and water balances at the surface are represented with CLM (Dai et al., 2003; Maxwell

and Miller 2005, Jefferson et al. 2017). CLM was coupled to ParFlow by passing the land surface water flux to ParFlow as a forcing in the top layer, and substituting ParFlow's computations for infiltration and streamflow routing within CLM (Maxwell and Miller 2005).

ParFlow has a number of differences with commonly used models in other drought-related studies, the Variable Infiltration Capacity (VIC) model (VIC 2016; Liang et al. 1994) and the Soil Water Assessment

Tool (SWAT) (Neisch et al. 2011). ParFlow allows any number of subsurface layers with any specified conductivity, and has vertical and lateral flow driven by pressure gradients. Soil moisture and groundwater





are not distinguished; both are represented through pressure in a cell and solved for using Richards' Equation. VIC typically has three soil layers and does not simulate lateral flow between cells, although it includes a baseflow term for water leaving a cell to enter a stream. VIC is often applied to larger scale modeling, while ParFlow can be used at any resolution. SWAT partitions groundwater into a "deep

aquifer" which can have lateral flow to other subbasins and a "shallow aquifer" which contributes only to the stream. Soil moisture and groundwater are modeled separately. In contrast, ParFlow combines all of these processes. The detailed, small scale representation of subsurface processes makes ParFlow-CLM a suitable model to run numerical experiments whose results depend on physical processes and their interactions, as opposed to statistical fitting or simplified parameterizations. However, this comes at a

tradeoff with higher computational expense due to these detailed representations.

## 2.2 Model Configuration

The model domain covers the southern High Plains and Rocky Mountains, including portions of the Arkansas and Missouri river basins (Figure 1) at a 1-km resolution. The domain is 1200 by 1124 km and extends to a depth of 102 km, with 5 layers for a total of 6,744,000 computational cells. The lowest layer is

100 m thick and the other 4 layers are 1 m, 0.6 m, 0.3 m and 0.1m thick, listed from base to top. An overland-flow boundary condition, allowing free development of a stream network, was imposed at the top layer. A no-flow boundary condition was specified at the bottom and on all sides, while surface streams were routed out of the domain at the edge. Due to computational expense, the runs in the present study were performed on the supercomputer Cori at the National Energy Research Scientific Computing Center

(NERSC). One year of the model required over 20,000 processor hours to calculate and completed in about a week of wall clock time.

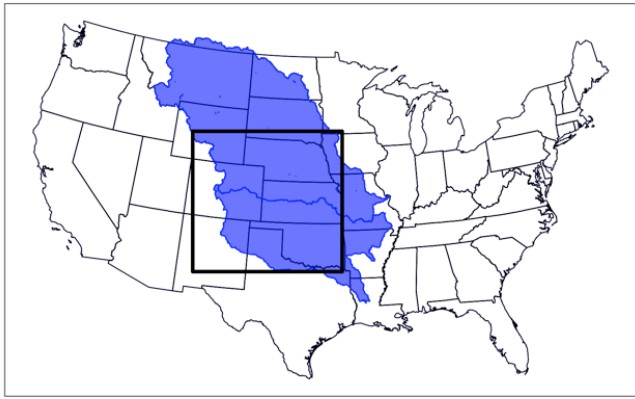

**Figure 1: The model domain (box) covers the southern High Plains of the United States. Blue shading indicates the Missouri and Arkansas continental river basins.**

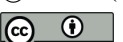



Inputs for the study were developed from previous work of Maxwell and Condon (2016) modeling hydrology across the continental US (CONUS). The basic input data and initial conditions follow Maxwell and Condon (2016). Inputs include slopes, soil types, vegetation, attributes of soils and geologic units, and

initial pressure conditions (see Figure 2). The slopes in the x (east-west) and y (north-south) directions were derived from a digital elevation model and processed to ensure the entire domain was connected. The slopes control surface flow routing. Soil types were taken from the SSURGO database. Important soil attributes include porosity, permeability, specific storage and van Genuchten parameters, which control saturated and unsaturated flow. Initial subsurface pressure conditions were taken from the original CONUS

run, to increase spinup efficiency. The vegetation dataset was taken from the USGS land cover trends dataset (Soulard et al. 2014). Important vegetative parameters include leaf and stem area index, roughness length and displacement heights, rooting distribution parameters, and reflectance and transmittance for leaves and stems (Maxwell and Condon 2016). While most inputs were drawn from Condon and Maxwell 2016, the geologic layer of that study contained features that were geologically less realistic at the scale of

the High Plains. The geology of the base layer was updated for this project using local data from the US Geological Survey (USGS, 1998; USGS, 2005).

It is important to note that water management was not included in the simulation. This includes groundwater pumping, surface water storage and diversion, and irrigation. This means that results of the project represent the system in a pre-development state not including anthropogenic impacts to the

hydrology (Maxwell and Condon 2016). The only management impact represented was land use through the vegetative layer.

The initial conditions for the simulation were obtained from the existing continental scale simulations (Maxwell and Condon 2016) and include 4 additional years of spinup prior to this project. The pressure file was subset to the High Plains domain and the geologic layer was updated as described in the

Appendix. The model was run recursively with water year 1984 North American Land Data Assimilation System (NLDAS-2) forcing until average subsurface storage change in one year was less than 1% of precipitation (achieved after three recursive runs). Ajami et al. (2014) showed that change in subsurface storage is one of the most rigorous spinup metrics for integrated hydrologic modelling. Holding this value below 1% of precipitation means that effects seen in numerical experiments can be interpreted as

meaningful, i.e. something besides spinup noise if they exceed the threshold.





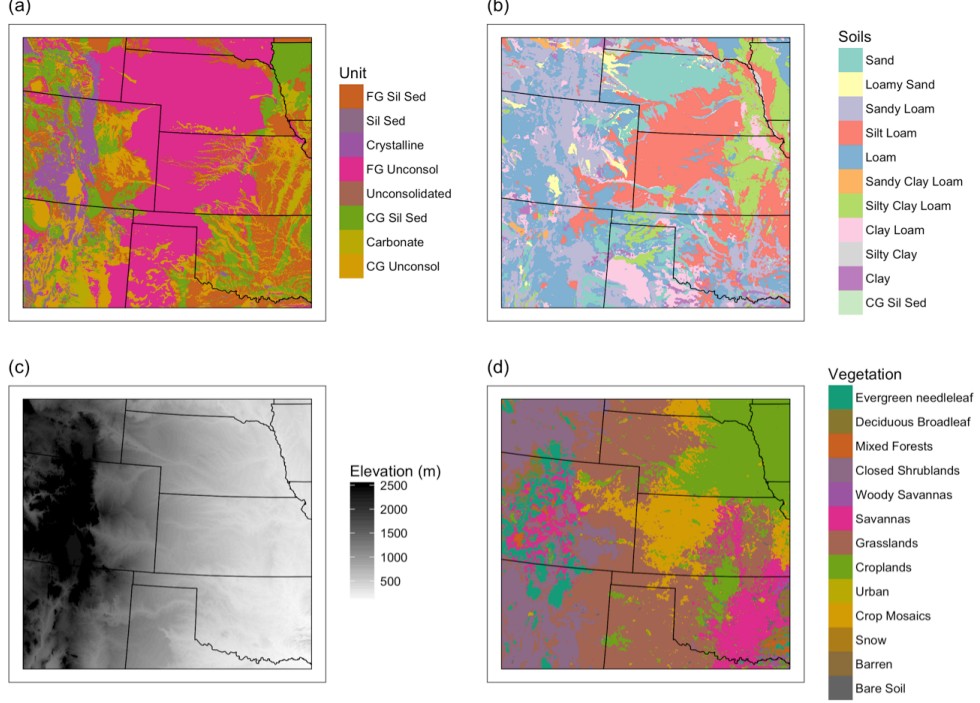

**Figure 2: Model inputs include a) geology to characterize the bottom model layer, b) soils to characterize upper layers, c) topography for flow routing and d) vegetation for surface energy and water exchanges.**

## 2.3 Numerical Experiments

A suite of synthetic drought scenarios were developed to explore the importance of precipitation, temperature and land cover change on regional drought response in the High Plains. All simulations are developed using the hourly observed NLDAS-2 historical atmospheric forcings from Water Year 1984 as a baseline (precipitation, temperature, pressure, humidity, short wave radiation, long wave radiation, wind speed). The experiments outlined in Table 2 and include a baseline run, three one-perturbation experiments

exploring the effect of precipitation, temperature and land cover changes in isolation, a combined experiment with both temperature and precipitation changes, and a worst-case scenario which also includes land cover changes.

Two further runs were also conducted to explore the importance of lateral flow as a mechanism within the model. Commonly used models including VIC and SWAT do not allow lateral flow within the model, and

including this process makes the model computationally more expensive. Creating normal runs with lateral flow and free-draining runs (i.e., without lateral flow) allows exploration of how this process affects model results. To construct a free-draining run, the water table was set at the base of the domain and all overland and subsurface lateral flow processes were turned off, while vertical flow through the soil column and



water table remained. With these settings, ParFlow-CLM mimics a traditional land surface model as described in Maxwell and Condon (2016).

Definitions and specific implementation of each numerical experiment are shown in Table 2, and the exact perturbations used are quantified in Figure 3. The baseline run and both free draining runs were conducted

5  for one year; the drought runs were conducted for three years of recursive drought forcing to simulate a transient time period a few years into a hypothetical severe drought.

**Table 1 Numerical experiments were implemented through changes to the model temperature, precipitation, vegetative cover and internal settings.**

| Term | Definition | Implementation |
| --- | --- | --- |
| Hot | Temperature above normal | Forcing input variable Temp was changed in each cell on a monthly basis. |
| Dry | Precipitation below normal | Forcing input variable APCP was changed in each cell on a monthly basis. |
| Crops | Land cover changed | Static input vegetation type was set to bare soil wherever it was normally crop or crop mosaic. |
| Free draining | Lateral flow forbidden | ParFlow keys allowing lateral flow were turned off. |

**Table 2: Model scenarios including run name and perturbations applied relative to the baseline scenario.**

| Run | Temperature | Precipitation | Vegetation | Lateral Flow |
| --- | --- | --- | --- | --- |
| 1 Baseline | | | | |
| 2 Baseline free draining | | | | Off |
| 3 Hot | Hot | | | |
| 4 Dry | | Dry | | |
| 5 Crops | | | Crops set to bare soil | |
| 6 Hot and Dry | Hot | Dry | | |
| 7 Worst case | Hot | Dry | Crops set to bare soil | |
| 8 Worst case free draining | Hot | Dry | Crops set to bare soil | Off |



Temperature and precipitation perturbations of the NLDAS forcing are based on PRISM reconstructions of the Dustbowl drought (the most extreme drought in the modern historical record). Perturbations were applied at monthly timescale for each cell of the domain. A spatial map of changes was
prepared for each month of water year 1934, one of the worst drought years historically recorded in the region, relative to the baseline decade of 1920-1929. Temperature was perturbed by adding an absolute temperature change to each cell of the hourly forcing for the relevant month. Precipitation was perturbed by multiplying each cell of the hourly forcing by a relative change for the appropriate month. Lastly, vegetation was disturbed for the crops runs by setting all crop or crop mosaic cells to bare soil (representing
the massive crop failure that occurred in response to extreme drought conditions). Figure 3 shows plots of the resulting annual anomalies in temperature, precipitation and land cover.

**2.4 Metrics of Drought Analysis**

Several metrics are applied to quantify drought impacts. First standard anomalies were calculated for all drought scenarios relative to the baseline by simply subtracting the Baseline values. The anomaly is a
simple metric of the model impact. Averaging the anomaly across the domain produced a measure of the total impact of a given factor. The single perturbation model runs allowed the calculation of average impacts ($I$) for each factor alone: $I$(temperature), $I$(precipitation) and $I$(land cover). The multi-perturbation runs allowed calculation of the impacts for the combined factors: $I$(temperature + precipitation + land cover) and $I$(temperature + precipitation).
The individual run impacts make it possible to assess whether impacts were linearly additive. If impacts are linearly additive, then the impacts of multi-perturbation runs (e.g. $I$(temperature + precipitation) ) should equal the sum of the composite individual perturbation runs (e.g. $I$(temperature) + $I$(precipitation)).   Here we quantify the nonlinearity in the combined drought response as a percent difference between the multi-perturbation impact and the expected impact assuming linear addition.



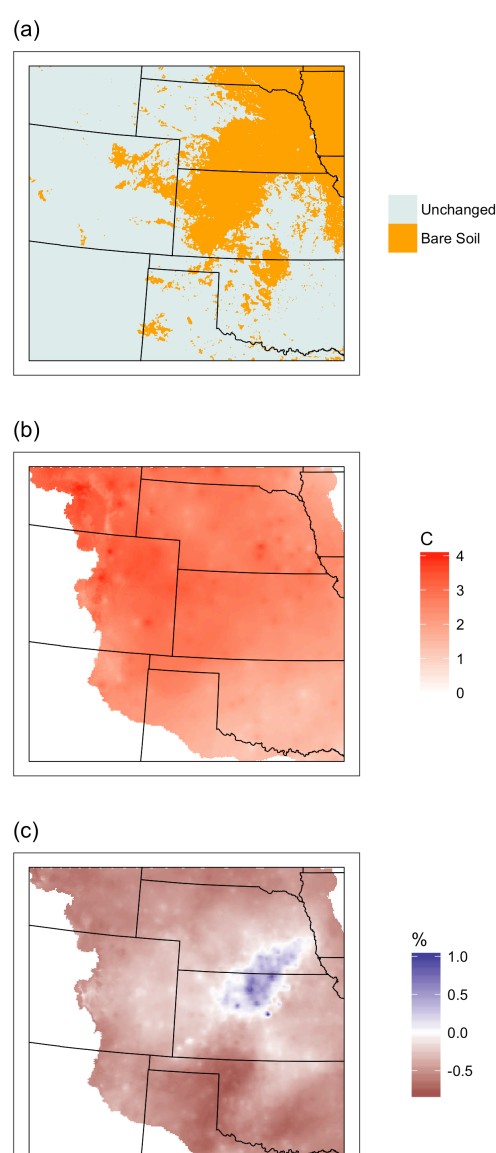

**Figure 3: Maps of the annual drought perturbations applied to the baseline scenario including (a) changes from cropland to bare soil, (b) absolute temperature increases, and (c) relative precipitation increases.**

## 3 Results and Discussion

5 Results are grouped into three sections. The first section focus on attribution of drought impacts to specific factors of temperature, precipitation and land cover. The second part quantifies how these factors combine and interact in the multi perturbation simulations, with particular focus on whether the impacts are linearly



additive. Lastly, the third section explores the importance of spatial scale to the predictability and linearity of impacts.

### 3.1 Attribution of Drought Impacts

The simulated drought runs produced large impacts to runoff, ET, soil water content and water table depth.
5   Here we compare the hydrologic impacts of different perturbation combinations to evaluate the relative importance of temperature, land cover and precipitation changes in hydrologic drought impacts. Figure 4 shows domain averaged annual values of runoff, ET, soil water content and water table depth. Relative to the baseline scenario all of the drought scenarios have decrease runoff and ET, depleted soil water content and lower water tables, as would be expected in a drought. Figure 5 maps these anomalies across the
10   domain and shows that impacts were typically most severe in the southern and eastern regions. In the central region where rainfall increased slightly (Figure 1c), this extra water partly avoided the most severe impacts. For example, in the dry and hot/dry runs, soil moisture and runoff anomalies were smaller in this central region.

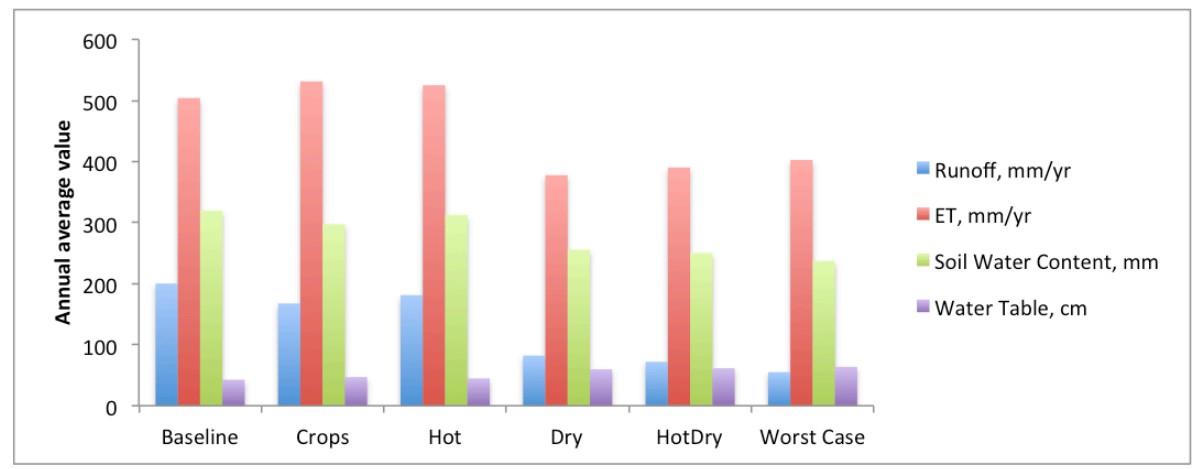

**Figure 4: Averages of runoff, ET, soil water content and water table depth, calculated on an annual basis across all cells in the model. Each colour represents a different variable and each cluster is a different model run. The baseline, crops and hot runs are generally wetter than the others.**





**Figure 5: Spatial maps of impacts to hydrologic variables, calculated by subtracting the baseline from each run. Each row represents one variable and each column is a different run. Panels (a)-(e) show runoff, (f)-(j) show ET, (k)-(o) show soil moisture and (p)-(t) show the water table.**

5        Comparison of single perturbation runs and multi-perturbation runs as shown in Figure 5 allows the impacts of the drought run to be attributed to individual effects and associated mechanisms. For example, disturbances to land cover produced strong but localized effects. Changing cropland to bare soil stopped transpiration in the affected areas, but increased total ET. Setting plant-covered areas to bare soil stops all transpiration in those cells, but the missing transpiration was more than compensated by higher ground

10   evaporation in the same seasons. No extra water was available to the system, so (as Figure 4 shows) the





increase in ET was balanced by a small decline in runoff and a small drop in soil moisture and water table levels over the two years. Spatially, Figure 5 shows that most effects were confined to the areas of vegetation change, with the exception of runoff, which decreased in downstream basins.

Increasing temperature produced small changes in several outputs across the domain. Hotter
temperatures increased ET. The increase was limited by available water as many cells remained close to their baseline ET regardless of the temperature increase. Figure 5 showed that runoff decreased by about 10% while soil moisture and water table levels remained almost the same as the baseline run.

Lowering precipitation drastically reduced all components of the water budget. ET declined, but was supported at first by declines in soil water content and water table levels. Although the second year
was just as dry, Table 4 shows that ET and runoff decreased less than in the first year. Effects were most pronounced in spring and summer. Spatially, ET and soil moisture anomalies were largest where precipitation deficits were largest.  The exception was along riparian corridors where streamflow and lateral convergence of groundwater maintained local soil moisture and supported ET (Figure 5). The larger impacts of precipitation relative to temperature are indicative of a water limited system as would be
expected for the High Plains.

It is important to note that the anomalies in the single-perturbation runs shown in Figure 5 have high spatial variance. Variability in the sensitivity to the applied drought anomalies are illustrated in Figure 5 which plots the response of individual pixels, to a given forcing change in temperature or precipitation. Variability in Figure 5 shows that the same perturbation can result in a wide range of responses. Figure 6
also shows these anomalies do not follow a simple functional relationship, which suggests that the ET anomaly for any particular cell is influenced by many other variables. Since ET is controlled by soil and vegetative resistance, soil type and land cover would be possible controls. However, the top panels of Figure 6 shows the anomaly colored by soil type, demonstrating that the wide range is independent of soil type. The lower panel shows the same for vegetative cover.



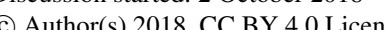

**Figure 6: Impacts for a given forcing span a wide range, independently of land cover as shown in (a) and (b) or soil type as shown in (c) and (d). In (a) and (c), annual anomalies from the dry run are plotted versus precipitation anomaly. In (b) and (d), annual anomalies from the hot run are plotted versus temperature anomaly.**

5       Overall, this demonstrates that decreasing precipitation caused the largest anomalies of any of the single factors. Relative to precipitation, temperature produced minor changes, especially in runoff. Vegetation change from crops to bare soil had large impacts on the local areas, but an intermediate effect when averaged over the domain as a whole. This applies to the relative size of the changes studied; a 2 degree C increase in temperature versus a 40% drop in precipitation and disturbance to land cover over

10     about 30% of the model area. For individual pixels, anomalies followed the expected pattern with heat



causing ET to increase and dryness causing it to decrease. The size of the anomalies, however, were relatively unpredictable even when controlling for the forcing change, soil type and land cover.

### 3.2 Factor Combinations

Section 3.1 presented the overall hydrologic impacts of individual and multi-perturbation runs. This section quantifies how these factors combine and interact. In a completely linear system, the single perturbation impacts would combine additively to equal the multi-perturbation impacts; however, because hydrologic systems are non-linear and there are many system components interacting simultaneously, some part of that total may also be due to interactions between the factors. Consideration of important mechanisms includes

land cover change and water limited behaviour. In general, individual factors can combine in a nonlinear way. For example, hotter temperatures occurring alone should increase ET, and drier weather alone should decrease ET as detailed in the introduction. If hot and dry weather occurs, the total impact to ET could be a combination of the individual impacts. There could also be an interaction between the factors in which the change due to one perturbation depends on the level of the other factor. For example, the increase in ET

due to temperature could depend on the amount of precipitation (which is a major control on available soil water) (e.g. Livneh and Hoerling 2016).

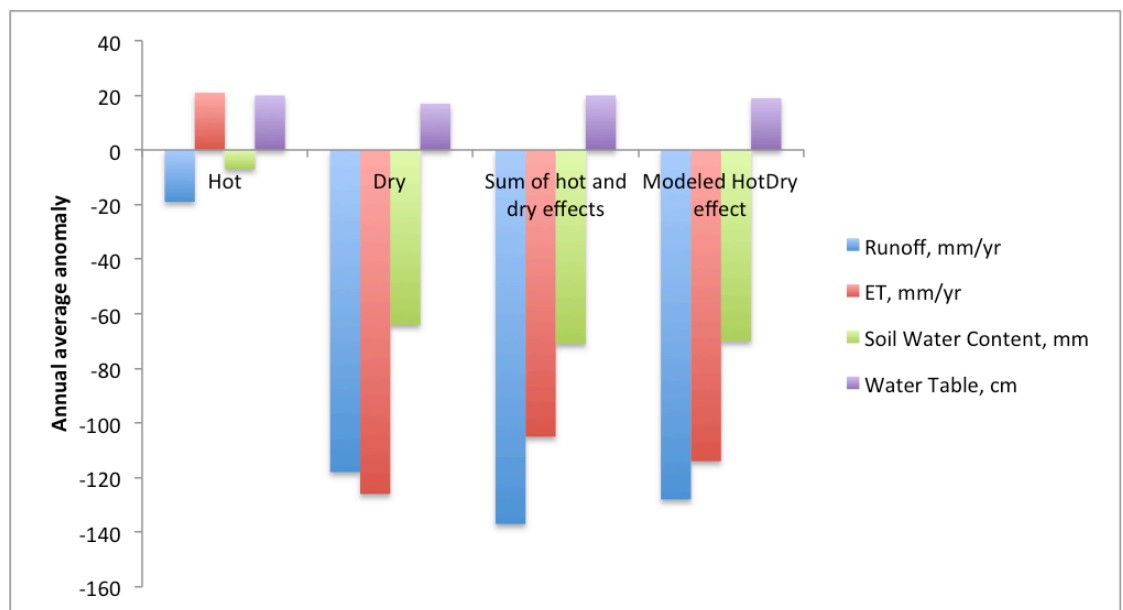

**Figure 7: Modelled anomalies between experimental runs and the baseline are shown for the hot/dry run, the hot run and the dry run. The anomalies in the hot/dry run are close to the sum of those in the hot run and the dry run. This linear behaviour is most true for**
**soil water content, less so for ET and runoff.**



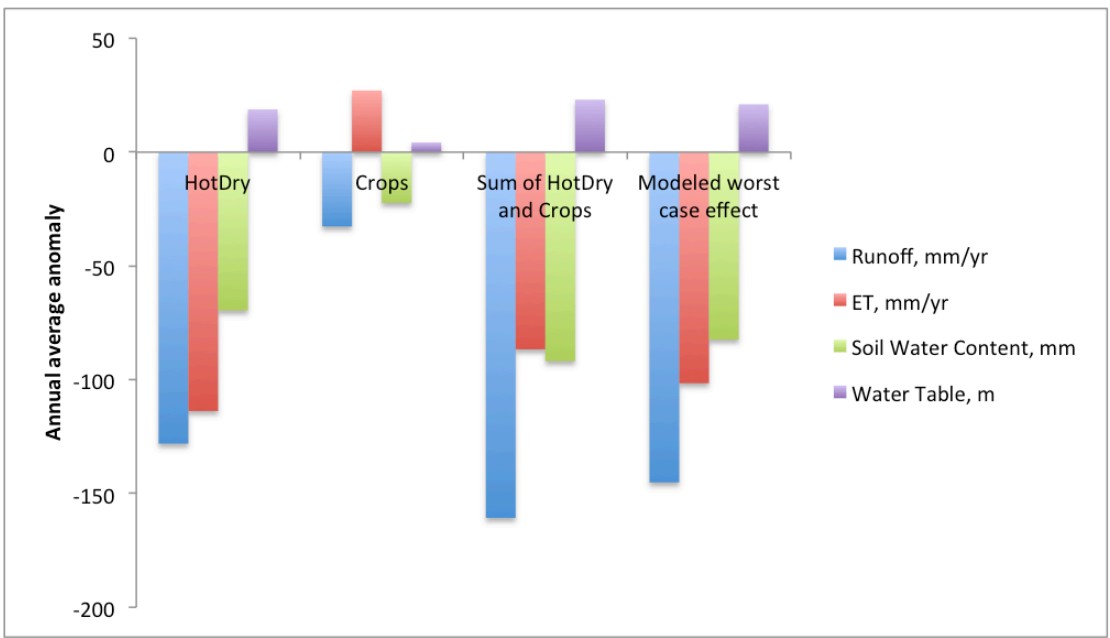

**Figure 8: Modelled anomalies between experimental runs and the baseline are shown for the worst case run, the hot/dry run and the crops run. The anomalies in the worst case run are close to the sum of those in the hot/dry run and the crops run; the largest difference is seen in runoff.**

Figure 7 and  Figure 8 compare the impacts of the single perturbation runs with those in the multi-perturbation runs. Each table compares the result of a multi-perturbation run (the "Modeled Effect" column) with the combination of single-perturbation runs (the "Sum" column). In a perfectly linear system, the two columns would be the same. The tables show that impacts of each multi-perturbation run can be attributed to individual components, but not completely. For example, Figure 7 shows that ET impacts

predicted by adding the impact in the hot run (a 21 mm/yr increase) to the impact in the dry run (a 126 mm/yr decrease) predict a combined effect in the hot/dry run of -105 mm/yr. This amounts to 93% of the 102 mm/yr decrease that was actually modelled in the hot/dry simulation. Similar patterns hold for the other variables, with the dry run contributing more than the hot run to every anomaly, as observed qualitatively in the previous section.  In  Figure 8, a similar analysis shows that the hot and dry run

contributes almost three times as much as the crops run to each anomaly. The figures also show that the individual components do not account for the entire effect. For example, in Figure 7 the hot/dry run is lacking 9 mm/yr of ET and keeps an extra 9 mm/yr of runoff compared to what would be expected by combining individual impacts.

It is instructive to examine the mismatch between modelled anomalies and those predicted by

linearity using spatial plots. Figure 9 maps this nonlinearity (percent difference, as described in the methods) across the domain for runoff, ET, soil moisture and the water table.  On average in the hot and dry run, ET generally decreased more than predicted by linearity, runoff decreased less, while water table





and soil moisture decreased less in the center, more in the north and south. Nonlinearity between the worst case run compared to the hot/dry run was naturally localized to the areas of land cover disturbance (i.e. where there were differences between the two). Runoff changed in either direction, ET generally decreased more than expected, and soil moisture and water table levels decreased less than expected. Importantly, the

nonlinearity spans a wide range of variation, with simulated multi-perturbation impacts being up to +/- 40% from the expected values in a linear system.

Antecedent soil moisture and water limited behaviour may explain some of the nonlinearity. Figure 10 shows the observed ET anomalies versus antecedent soil moisture for a given forcing change (color scale). The top panel is the dry run and the lower panel is the hot run. The clear break in both panels at

about 350 mm of soil water content indicates the importance of antecedent soil moisture. Above this cutoff, increasing temperature produces the largest increases in ET. Decreasing precipitation produces only small anomalies, because enough water is available in the wettest cells to supply the continued ET demand. Below this cutoff, however, the system is water limited. In the presence of a temperature increase, soil water content limits the possible ET increase. These observations from the single perturbation runs support

a mechanism for the nonlinearities that has been suggested in previous research (e.g. McEvoy 2016, Seniviratne 2010): when precipitation decreases, there is less available water to supply ET demand even when a rising temperature increases potential ET. Thus the simulated multi-perturbation ET is smaller (in other words, the deficit in ET is larger) than would be expected by simply combining factors.

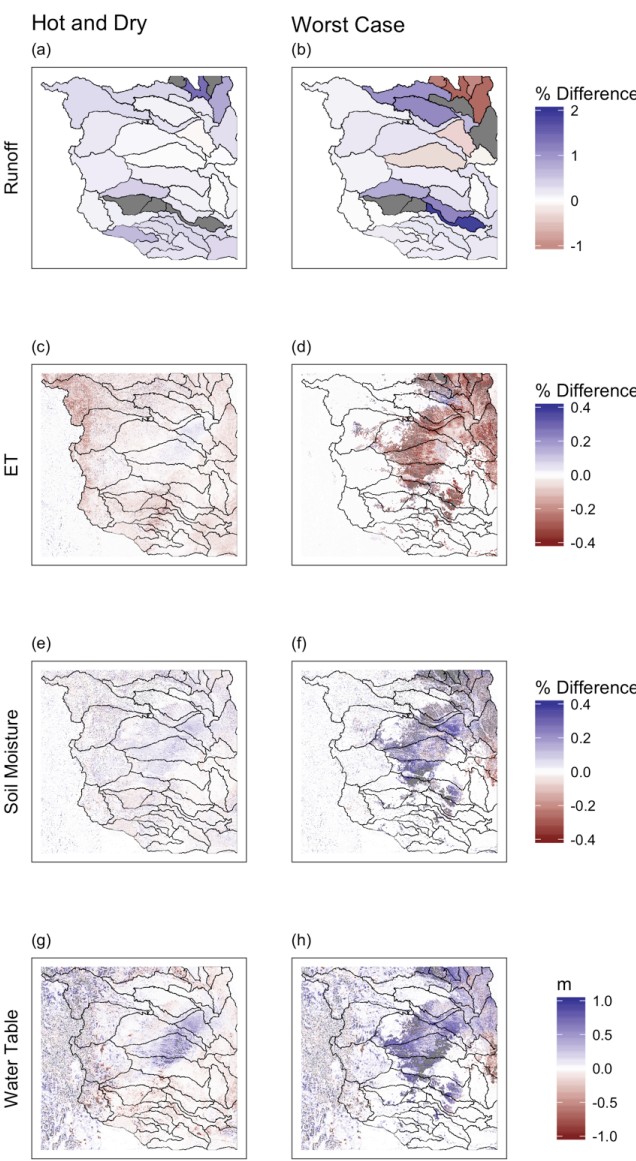

**Figure 9: Nonlinear behaviour in both multi-perturbation runs shows spatial patterns. Panels (a) and (b) show runoff, (c) and (d) show ET, (e) and (f) show soil moisture, and (g) and (h) show water table levels. The left hand column compares the hot/dry run to the single perturbation hot and dry runs. The right hand column compares the worst case run to the crops run and the hot/dry run.**





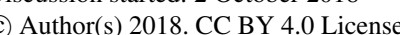

**Figure 10: Antecedent soil moisture is an important control on model ET response. The ET anomaly is plotted versus antecedent soil moisture and colored by model forcing. Panel (a) shows data from the dry run (color scale is precipitation change) and panel (b) shows data from the hot run (color scale is temperature change). Each point is one model cell. Below about 350 mm of soil water content, the cells show water limited behaviour in which drought causes decreased ET depending on severity.**

### 3.3 Importance of Spatial Scale

This section examines the importance of scale in assessing these processes. Impacts of individual factors show less variability and more dependence on model forcing at larger scales. As was discussed in section 3.1, impacts of individual factors can be unpredictable at small scales ( Figure 6); in other words, a given forcing change can produce a wide range of impacts to ET. Figure 11 shows that this variability is





greatly reduced as soon as ET anomalies are aggregated to small (HUC-8) drainage basins. By the time the ET response is aggregated to subcontinental watersheds on the scale of the Arkansas or Red river, it appears to have little remaining variability.

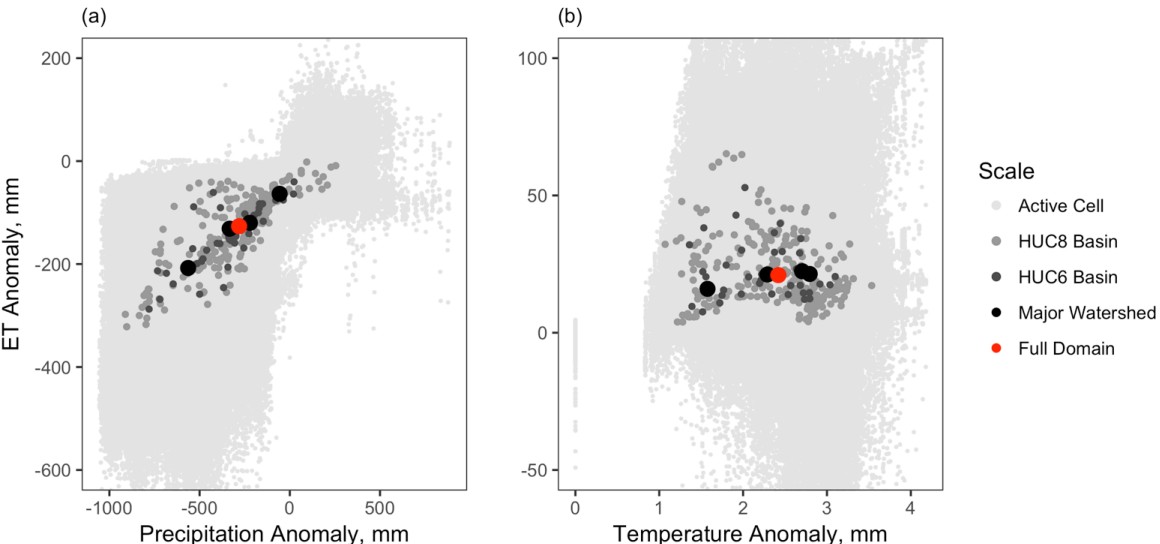

Figure 11: Impacts of individual factors become less random at larger scales. Panel (a) shows ET anomalies of the dry run and panel (b) shows ET anomalies of the hot run, plotted against their respective forcing anomalies. Impacts begin at the individual cell level and are aggregated to a series of larger scales.

Combinations of factors becomes more linear at larger scales. Section 3.2 showed that nonlinearity (i.e. the portion of the response not accounted for by the linear combination of the individual perturbations) can span +/- 40% for a given pixel ( Figure 9 ) while being much less at the entire domain scale (Figure 7 and Figure 8). Figure 12 shows that the nonlinearity is much decreased in moderately sized (HUC-6) river basins, and small in subcontinental river basins. This is especially important because it shows that treating the system as linear would fail to capture the most severe impacts occurring in individual pixels.

**Figure 12: Factors combine in a more linear way at larger scales. Each panel shows boxplots that characterize deviations from linearity from model cell to subcontinental scales. Panel (a) shows runoff, (b) shows ET, (c) shows soil water content and (d) shows the water table.**



The previous two figures may seem to contradict the message of earlier sections. If the impacts of individual factors are actually straightforward and combine in a basically linear way at subcontinental scales, perhaps simplified models would be adequate to answer big-picture questions at large scales without

involving the full complexity of an integrated hydrologic model. The free draining run provides insight to address this question. It allows no interactions between cells and can thus be considered as a package of single column models, run across a spectrum of soil types, slopes and land cover.

An example of a result that changes with the free draining run, even at large scales, is the effect of crop disturbance on ET. Figure 13 compares ET in the worst case simulation to a baseline in both the free

draining and typical configuration. Inspecting the area of crop disturbance in the lower panel of Figure 13 shows that in the normal model configuration, ET increases when crops change to bare soil (also discussed in section 3.1). However, in the same area of the upper panel, exactly the same forcing changes cause ET to decrease. In the free draining run, there is a quantitatively and qualitatively different response to a big-picture, large scale question about the impact of land cover. This shows that even large-scale predictions

depend on representation of lateral flows and interactions within the model.

Overall, this section shows that unpredictable and nonlinear small scale impacts aggregate to predictable, linear large scale behaviour. The complexity of model response depends on the scale of the area of interest, with individual km-scale pixels being complex and subcontinental river basins being much more simple. Responses at any scale nonetheless depend on representation of the processes and feedbacks

at smaller scales. At coarse resolution, linear models may provide usable results; but as the resolution increases, accounting for nonlinearities becomes more important. This means that coarser scale simulations such as those often run on VIC may capture big picture drought-related impacts, but may miss the finer scale local variation.





(a)

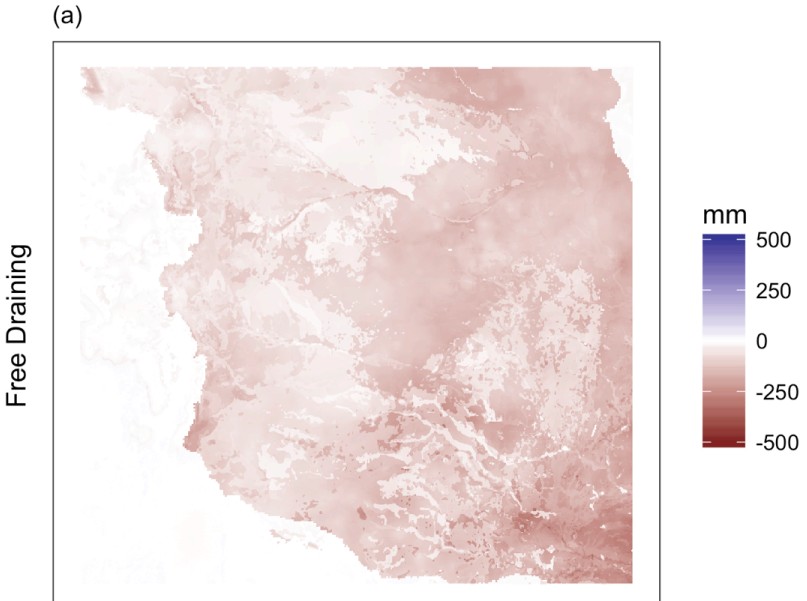

(b)

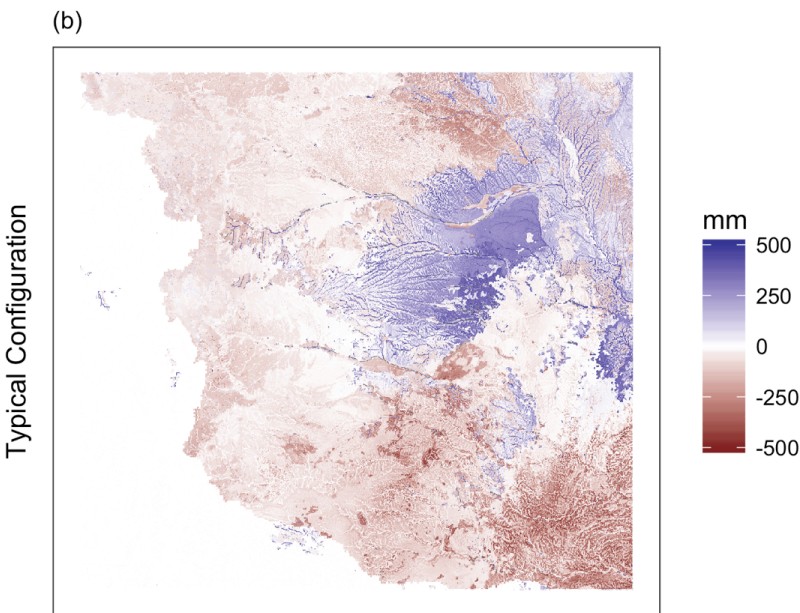

**Figure 13: Large scale impacts of crop changes depend on representation of small scale processes. Spatial maps are calculated by subtracting (a) the free draining baseline (Run 2) from the free draining worst case run (Run 8) and (b) subtracting the baseline with lateral flow (Run 1) from the worst case run (Run 7). Panel (a) shows the free draining run where ET decreases in areas of land cover disturbance. Panel (b) shows the typical configuration for comparison, where ET increases in areas of land cover disturbance..**





## Conclusions

This study explored impacts of drought-related drivers and relevant mechanisms through a series of numerical experiments using a ParFlow-CLM model. Simulations based after the example of the Dustbowl drought of the 1930s individually perturbed temperature, precipitation and land cover, followed by multi-

perturbation runs that combined these changes as would occur in a real-world scenario.

Attribution of drought effects to single factors showed that lowered precipitation causes more severe effects than increased temperature, within ranges of variation typical of major droughts. All impacts are ultimately due to forcing changes, but Sections 3.2 and 3.3 showed that moisture limitations and scale also influenced responses and produced more complex behavior. Soil types and land cover had minimal

effect. The complex behavior described above produced nonlinear impacts at small scales, however, these impacts became much more predictable and linear at large scales. Although large-scale behavior appears simple, including complex small-scale processes such as lateral flow between cells was crucial to representing the large-scale responses.

These results address the research questions presented in the introduction.

The first question focused on the relative importance of precipitation, temperature and land cover change in hydrologic response to drought. The results show that precipitation is relatively more important than temperature or land cover change in hydrologic response to drought. The effects of precipitation are on the order of 3 times the size of the effects of temperature or land cover change, for ranges plausibly seen in extreme droughts. This is in line with results of prior studies including Livneh and Hoerling (2016) and

Maxwell and Kollett (2008). However, the exact effects of forcing change are somewhat unpredictable and this broad result may not hold true for individual pixels.

The second question took these individual effects and combined them to see whether when the main drought factors (precipitation, temperature and land cover change) occur together, their effects are linearly additive. The effects can be linearly additive on a large scale for variables such as soil water content, but

they are slightly less linear for variables such as ET or runoff. For individual model pixels, the effects can be +/- 40% of the expected value. This agrees with expected system feedbacks such as those described by Eltahir (2008), and expands on the results in Maxwell and Condon (2016).

The third question asked how impacts of the main drought factors and their interactions change across spatial scales. Results showed that highly variable and nonlinear impacts modelled at small scales

aggregate to much less random, linear large scale behaviour. Even these large scale predictions depend on representation of the small scale processes and interactions. This extends the work of Maxwell and Condon (2016), which showed that lateral flow, affects ET thresholds within the system.



Future studies could build on the work shown here by incorporating more detail. Including surface water management such as dams or irrigation diversions would make streamflow more representative of a present-day water year on the High Plains where streams are heavily managed. Inclusion of irrigation and groundwater pumping would allow study of human impacts to groundwater and surface energy balance.

Further updates to the available geologic datasets would allow the model's existing detailed representation of subsurface processes to be based on better-supported parameters.

Results of this study may indicate future sustainability challenges on the High Plains. As climate change advances, temperature will almost certainly increase and precipitation may decrease. This study shows that changes in precipitation are more worrying than changes in temperature, within the ranges

simulated. Additionally, when regional drought is occurring, local impacts may be many times smaller or larger. This matters because the most severe and costly impacts may occur in such small scale, nonlinear responses. While the exact location and size of these small scale anomalies is not predictable with a model like the present, their general existence is. Even without specific predictions, plans for responding to regional drought will be more resilient and adaptive if they anticipate small-scale, severe impacts like those

modeled here.

**Competing interests**

The authors declare that they have no conflicts of interest.

**Acknowledgments**

The simulations presented here were completed using resources of the National Energy Research Scientific

Computing Center (NERSC), a U.S. Department of Energy Office of Science User Facility operated under Contract No. DE-AC02-05CH11231. Funding was provided by the National Science Foundation through its Water, Sustainability and Climate (WSC) program and NSF award number 1204787.

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
