# Peer review of "Unravelling the impacts of precipitation, temperature and land-cover change for extreme drought over the North American High Plains"

_Hydrology and Earth System Sciences, 2018_

## Short Comment (SC1) · 27 Oct 2018

Note to the editor and authors: As part of an introductory course to the Master pro-gramme Earth & Environment at Wageningen University, students get the assignment to review a scientific paper. Since several years, students have been reviewing papers that are in open online discussion for HESS or BGS, and they have been asked to submit their reports to the discussion in order to help the review process. While these reports are written in the form of official (invited) reviews, they were not requested for by the editor, and we leave it up to the editor and authors to use these reports to their advantage. While several students were often asked to review the same paper, this

was not done with the aim to provide the authors with much extra work. We hope that these reports will positively contribute to the scientific discussion and to the quality of papers published in HESS. This report/review was supervised by dr. Ryan Teuling (teacher within the ITEE course at Wageningen University and also associated editor with HESS).

Dear editors of Hydrology and Earth System Sciences,

In this paper, Hein et al. use the coupled model ParFlow-CLM to describe which of land use changes, temperature changes, and precipitation changes representative of extreme drought have the largest impact on drought severity in the southern High Plains of the United States. They then describe which feedbacks are present between these variables responsible for drought, and finally they discuss how these feedbacks change with spatial scale within the modeled region. It is novel and interesting because it successfully answers each of its research questions, providing new knowledge with respect to how causes of drought impact each other, as well as demonstrating new and sophisticated drought modeling methods. It seems appropriate to me to gain a better understanding of how drought might change across scale, and that understanding the feedbacks between variables responsible for drought is a meaningful advancement of our understanding of drought. This paper is clearly in the scope of this journal, as it directly studies phenomena in hydrology. The work is generally well-researched, and the writing quality is good. While I would strongly recommend this work for publication, there are some minor revisions that I believe need to be addressed before it is truly of publication quality. This review has two parts: more specific arguments for the work's novelty and quality, followed by revisions that I think would make the paper stronger.

This paper is relevant because it fills a clear knowledge gap in the field of drought models. The authors claim that there are many studies that use models to forecast drought scenarios or compare the accuracy of models to historical data. However, the authors claim that these studies either isolate variables responsible for drought and describe their effect on drought, or they describe or forecast the general impacts of combinations of factors without considering specifically what feedbacks might be present between variables responsible for drought. I agree with this assessment, from reading recently published literature on drought models and from reading articles the authors of the present paper cite. Zhao and Dai (2016) report on the accuracy of various drought models when compared to observed trends in Palmer index, where models have forcings in greenhouse gas concentration changes, precipitation changes, and other variables. However, in this paper, the authors make no attempt to ascribe the impacts of drought to specific forcings, rather describing model general accuracy as a result of a complex combination of forcings. Deo et al (2016) develop a method to forecast drought using a sophisticated wavelet analysis, but they only consider "water accumulation," which is based on rainfall, rather than a combination of factors, to arrive at this forecasting. Rad et al (2017) develops a new method for drought prediction incorporating both likelihood for drought onset and likelihood for drought persistence, but their prediction is based on the relationship between rainfall and streamflow, which again does not consider the feedbacks between many variables responsible for drought. Su and Dickinson (2017) consider in depth the feedback between soil moisture and precipitation in the 2011 drought in the same area studied by the present paper, but other feedback relationships are left out. By contrast, in this paper, the impacts of drought in ET, runoff, water table level, and soil moisture are compared across runs with individual perturbations and runs with combinations of hot, dry, and crop failure perturbations. In this way, we see how for example a combination of hot and dry conditions affects ET differently than the sum of impacts due to individual hot and dry scenarios, or how the sum of impacts in water table depth from single-perturbation runs with hot, dry, and crop failure conditions differs from the impact in water table depth given by a scenario that considers all three of those perturbations, etc. So, by giving a quantification of the differences (nonlinearity) between single- and multi-perturbation model runs, the authors of the present paper provide new knowledge on the feedbacks between land use changes, precipitation changes, and temperature changes.

Specifically, the paper does this via three research questions, and it clearly provides
a satisfactory answer to each. The first research question involves quantifying which of land cover change, temperature change, or precipitation change has the biggest individual effect on drought via changes in ET, groundwater depth, runoff, and soil moisture for ranges of perturbations found during extreme droughts. It achieves this by comparing single-perturbation runs to a baseline run, and determines that changes in precipitation have by far the largest affect on drought. This agrees with Luo et al. (2017), who use the variable infiltration capacity model (VIC) to demonstrate that reductions in temperature have a larger impact on agricultural drought than increases in temperature.

While the first research question is already somewhat well-studied, the second and third research questions are not. The second research question quantifies the feedbacks present between factors responsible for drought. In order to do this, it runs the model for scenarios that include multiple perturbations (hot and dry conditions, and hot, dry, and crop failure conditions). Then, it compares the drought impacts from these runs to the sum of drought impacts from single-perturbation runs. The percent difference between these impacts is called the nonlinearity, and describes how the factors responsible for drought affect each other. It goes on to provide a theoretical physical justification for these differences, and finds that the nonlinearity is largest in ET and runoff.

The third research question aims to determine how linearity changes across spatial scales. In order to do this, it aggregates linearities from the pixel level to increasingly large scales, all the way up to the sale of the entire model. The result is that variation in linearity decreases and linearity increases as the spatial scale increases. That is, there is more disagreement between sums of impacts of single-perturbation runs and impacts of multi-perturbation runs at the small scale, whereas at the large scale both approaches yield similar results. The paper then shows how inclusion of small scale detail is still meaningful for large-scale results by demonstrating that removing the small-scale process of lateral flow changes sign and magnitude of ET across a

large portion of the model area. Thus, all three research questions are thoroughly answered, providing novel quantification of feedbacks between processes responsible for drought at different spatial scales.

This paper also represents the next step in modeling sophistication. Upon reviewing the author references on model choices, I agree that the inclusion of lateral flow is more computationally expensive and yields more detailed results, that VIC does not allow lateral flow and generally has less detailed parameter options (and sees its greatest use in large-scale modeling) (as can be seen in Liang et al. 2003), and SWAT can only have non-groundwater lateral flow in depths of zero to two meters, which might not accurately reflect lateral flow in conditions where there is deeper groundwater (as can be seen in Green et al. 2006). ParFlow is evidently an advanced model in the field of drought modeling, and this study is made even more sophisticated by its fine grid resolution, which allows for a meaningful comparison of model qualities across spatial scales (albeit at computational expense).

This paper is socially relevant in that it could have important implications for drought management policies in the studied region. Specifically, the existence of large variations from regional trends at the pixel (1km) scale is well-established in this paper. Any drought response or preparation plan could take this into account, expecting variability in necessary resources or action on the small scale.

Finally, the writing style is clear and encourages rather than hinders reading.

While this is generally an excellent paper, it could be made tighter in key spots. The following recommendations are in order of decreasing importance.

In order to determine the nonlinearity of a multi-perturbation run, the authors subtract drought impacts (in ET, soil moisture, groundwater depth, and runoff) from the sum of the drought impacts resulting from single-perturbation runs. Though the single- and multi-perturbation runs generally have a duration of three years (the worst case free draining run is the exception), the nonlinearities are reported as single values, suggesting the authors aggregate the nonlinearities of the three years. I acknowledge that one of the intents of the paper is to determine the feedbacks between factors affecting drought, and this doesn't necessarily involve observing change in linearity across temporal scale. Still, I think it warrants discussion, because 1) certain physical processes in the model have long memories (groundwater, soil moisture) and thus might have varying temporal linearity trends, and 2) various factors' contributions to linearity might be obscured by aggregating over three years. So, if there are no significant temporal trends in linearity, I recommend providing maybe just a paragraph or so detailing this. If there are significant trends, I recommend reporting on them by describing how the linearity changes with each successive year of the model.

This point is simple, but important, and also pertains to linearity. Figure 12 is a key figure for this paper - in it, the authors show the deceasing variability in nonlinearity with increasing scale and the average trend towards linearity with increasing scale. They do not indicate which nonlinearities are reported in this figure. There are three options: nonlinearity between the "worst case" run and the sum of the "hot," "dry," and "crops" runs, nonlinearity between the "hot and dry" and sum of the "hot" and "dry" runs, and nonlinearity between the "worst case" and sum of "hot and dry" and "crops" runs. I'm generally willing to believe the trends in linearity that the authors report, but specifically what we're looking at in figure 12 should be established before any general trends are made. Is this figure the nonlinearity between the "worst case" run and the individual runs, the nonlinearity associated between the "hot and dry" run and the individual runs, or a sum or average of the two? Or, does it incorporate the nonlinearity between the "worst case" and "hot and dry" runs?

While the model is extremely detailed and state-of-the-art, the paper lacks comparison to actual data. I appreciate that the authors provide a physical description of the observed nonlinearity in ET (the hot and dry scenario has a nonlinear ET value because when precipitation decreases, there is less water available for ET even if the temperature rises which would normally mean a higher ET), and that this nonlinearity

is theoretically supported by McEvoy and Seniviratne. I also appreciate that it might be hard to find "real world data" on exactly what the authors study in this case - after all, you cannot create multiple droughts with exactly the conditions you want, especially not on the studied scale. But finding some physical justification aside from a theoretical validation would make the paper stronger. The authors are already well-poised to do so, in the case of antecedent soil moisture. Finding an area that has data on ET at various soil moistures under hot, dry, and hot and dry conditions, and then comparing these values, could offer further validation to the model.

While the theoretical physical description of nonlinearities is good, it doesn't entirely prove that nonlinearities aren't model artifacts. Perhaps the authors could include a few paragraphs clearly indicating why the observed nonlinearities aren't just due to the nature of the model.

There's some lack of citation. Though the paper uses in-text citations of McEvoy 2016 and Seniviratne 2010, neither of these citations occurs in the references section. So I recommend that the authors include reference information for these two papers and check that all of their references are listed.

There are differing grid scales for PRISM and model data. The ParFlow-CLM model runs at 1km cell size, while PRISM data has a 4km cell size. Because a central point of the paper is how small scale detail affects large scale properties, the paper might be made slightly stronger by addressing how the PRISM forcings are coarser than their model's resolution, and why this matters or doesn't.

The authors indicate that they compare 1920s precipitation and temperature to PRISM reconstructions for each month of 1934, but it is unclear whether they obtain the perturbations from comparing months of 1934 with average months in the 1920s, or if they compare months of 1934 to averages for the entirety of the 1920s. Also, the source for the meteorological data of the 1920s is not indicated.

ParFlow allows for a cell to have 4 of 16 different possible vegetation types present. In

the paper, they simply indicate that some of the land is covered by "crops." Perhaps they could include a more detailed parameterization of the crops, as different crops will surely have different impacts on drought (for example, via ET). However, the authors could probably safely ignore this comment, because they do indicate that they're looking for the presence and degree of linearity across scales and the relative impacts of perturbations, rather than forecasting specific events.

The authors state "combinations of factors become more linear at larger scales," but it appears as if nonlinearity actually increases specifically in runoff from the "major basin" to "full domain" scales, although variability in linearity does seem to follow the appropriate trend across scales. Though it is just a variation in one drought factor between two similar scales, perhaps it should be briefly explained.

The water year 1984 is used as a baseline. Some brief indication should be given that this is a typical water year, or at least not a year of drought.

General typos Page 3, line 32: you should consider including "water table level" in the text of the first question. Page 4, line 8: change "19390s" to "1930s" Page 4, line 12: include "it" after "because" Page 4, line 13: include "for" after "suited" Page 5, line 14: change units of km to m. Page 7, line 9: delete "and" between "2" and "include" Page 10, line 5: change "focus" to "focuses" Page 11, line 8: change "decrease" to "decreased" Page 15, figure 7: Apparently, the anomalies in water table are not summed for the "sum of hot and dry effects." Page 20, line 8: change "becomes" to "become" Page 23, line 5: delete a period at the end of the sentence. Page 24, line 32: delete the second comma.

I hope this helps the authors improve their manuscript, and I hope that this paper becomes published.

Sincerely,

Peter La Follette

---

## Referee Comment (RC1) · Anonymous Referee #1 · 28 Oct 2018

Review of Hein et al. Unravelling the impacts of precipitation, temperature and land-cover change for extreme drought over the North American High Plains.

Summary

Hein et al. present a modeling study, which evaluates the impact of (hot and dry) atmospheric forcing anomalies on the land surface/subsurface water balance over the North American High Plains. It studies the nonlinearity and scale dependency of these feedbacks through single and combined perturbations of temperature and precipitation in the atmospheric forcing. In a second step, Hein et al. evaluate the impact of land use on these feedbacks and they try to address the importance of groundwater in alleviating these feedbacks. The general idea behind this work interesting and novel. Unfortunately, the manuscript shows major flaws in the presentation of results and would benefit from more precise (and, in my opinion, intuitive) research question. Although the proposed research questions may be evaluated as adequately addressed, both, methods and findings are not adequately presented. Yet, the idea behind this work is interesting, and could, more precisely formulated, address a much more relevant topic (see below). The model simulations along with the nonlinearity aspect provide all the "ingredients" required to address this. I hence recommend major revisions. These revisions should address (1) a clear and potentially more relevant message at hand, (2) a better representation of results and (3) some minor corrections and clarifications.

(1) Message and relevance.

The manuscript misses a substantial differentiation between meteorological and hydrological drought. More precisely, it misses the differentiation between (i) the impact of meteorological drought and (ii) the response of hydrological drought to the former. I am referring to meteorological drought as hot and dry atmospheric forcings, and to hydrological drought as e.g. water shortages and anomalously low groundwater levels, soil moisture, ET and runoff. The combination of both aspects, along with a focus on the nonlinearity of feedbacks, would make this manuscript worth publishing. The authors do, however, only address the first aspect, i.e. they evaluate the impact of meteorological drought conditions (hot, dry, hot and dry) on an annually averaged water balance without referring to the land surface/subsurface state. And according to their own introduction, the fact that precipitation deficits are the main driver for (hydrological) drought, is not novel. Yet, the authors do not explicitly show that their model does simulate a hydrological drought and how the forcing perturbations impact this drought. This is why, at the end of the manuscript, the reader is left wondering what the impacts of anomalous dry and/or hot conditions on an existing drought are, and even more simple, if a hot temperature anomaly alone can initiate or aggravate a hydrological drought. I personally like the analyses of the nonlinearity of feedbacks, which should, in my opinion,

be the main topic of the manuscript and could help to increase relevance. Specifically, what would be really interesting, is the combination of both drought aspects along with the nonlinearity analysis. This would intuitively lead to interesting questions, such as: Does the nonlinearity of the land surface feedbacks to meteorological drought forcing aggravate or dampen the hydrological drought? Does it change extremes (as indicates in title!)? How does it impact severity and extent of hydrological drought? How does land use buffer the impact of (the nonlinear feedbacks of) meteorological drought on hydrological drought? The simulation experiments seem to be designed to address exactly these questions, but the manuscript does not.

(2) Methodology and presentation.

In addition, both, methodology and analyses would benefit from a more precise description. In the following, I will list a couple of (important) issues that remain unclear to me and hampered my understanding:

- Model selection. I understand the advantage of ParFlow as a numerical, physics-based model which simulates lateral flow over other models, such as VIC and SWAT, and the advantage is clear from the description. I do, however, not see the need to "badmouth" other models if they are neither being used and compared, nor validated against observations. Moreover, I do not understand the comparison of lateral flow/ no lateral flow influence on ET in Fig. 13. This is not connected to the research questions proposed and setup and results are not well explained. Do you apply a constant water table in the free drainage runs? If so, did you perform a separate spinup for those runs? Or might the differences in Fig. 13 simply arise because you have different water table depths?

- Numerical experiments. The use of different experiment names, e.g. "Hot and Dry", "hot/dry", "Hot and Dry" (are they all the same "Hot and Dry" run (6) from Tab. 2?) is really confusing and makes it hard to follow. Please unify.

- Simulation period and dependency on land surface/subsurface state. I am totally confused about the simulation period and the setup of the numerical experiments: Which years are simulated? Why is the model set up with data from 1984? Do you simulate a hydrological drought period, and if so, why do you not evaluate the impact of your forcing perturbations on evolution and extent of drought? Even if you do not simulate a hydrological drought, it is still important to evaluate the differences in relation to the land surface and subsurface states. What is the relative importance compared to the natural (modelled) variability? E.g. the reader does not know if a water table difference of ∼1 m (Fig. 5) is on the order of natural, (inter-)annual variability and if it occurs in a region of shallow or deep water tables.

- Time scales. The authors only show annually averaged differences, which do not allow to address drought and extreme impacts (as indicated by the title!). The limitation of presenting annual averages becomes evident in Fig. 10, which shows the relation between "antecedent soil moisture" on ET. First of all, I do not really see a "clear break" (p. 17, l. 9). Secondly, do you use an annual average as antecedent soil moisture? Soil moisture varies at much shorter time scales; and a grid cell (region) might move from an energy limited towards a soil moisture limited state within a year, and especially during a drought. Maybe it makes sense to look at shorter time scales . . . otherwise I do not see the merit of Fig. 10. Whether a grid cell is soil moisture or energy limited also depends on the soil texture, doesn't it? I am not sure that this can be as simplified as the text does it.

- Spatial scales. Sec. 3.3 remains unclear to me, though it could be potentially very interesting. This may be mainly due to my lack of understanding what is shown in Fig. 12. Could you please clarify? Does Fig. 12 show the same comparison as in Fig. 9 but the sum? But then, which scenario is shown? What are HUC6 and HUC8 basins? And which are the major basins? Which basins do you actually show in Fig. 5?

(3) Minor points.

- If I understand correctly, "anomalies" are neither climatological anomalies, e.g. of soil

moisture or runoff to determine drought extent, nor are they the anomalies of forcing during the 1934 (?) drought, which are used to perturb the forcing. If I understand correctly, anomalies in the manuscript are simply the difference between a scenario and the baseline simulation. Please clarify and consider rephrasing.

- A lot of references are missing (e.g., Loon 2015; Eltahir 1998; Seneviratne 2010; Betts 1996; Koster 2004; McEvoy 2016; . . .)!

- p. 11. l. 7-9: The description of Fig. 5 in the text is misleading. If I see this correctly (and as is later on in the manuscript mentioned), Hot (and Crops) have higher ET! Also, please be precise what "lower" WTD means. . . it's deeper?

- Fig. 9: The differences in Fig. 9 (c-d-e-f) are not percentages, are they? 0.4 % difference in Fig. 9 seems rather small and not significant (or do you mean 40% as mentioned in the text on e.g. p. 17?). What do grey colors in Fig. 9 mean? Do I understand correctly, that Fig. 9 shows (a,c,e,g): Hot (3) + Dry (4) - "Hot and Dry" (6) ? (b,d,f,h): Hot (3) + Dry (4) + Crops (5) - "Worst case" (7) ? (with numbers referring to runs from Table 2)

- There are no tables representing results; hence the paragraphs on p. 13 l. 8-11 and p. 16

- l. 6-9 are confusing. Please remove/rephrase.

- p. 14 l. 8-10: this relates to the forcing perturbations, does it? So, if I sum up the precipitation changes from Fig. X over the domain, they correspond to a total change of 40% ?

- p. 17 l. 17-18: Is this related to the results presented in Fig. 9c? (Here, radiation probably has a stronger impact than temperature).

- Conclusions. (i) contradiction of "real world scenario" to what you describe in the introduction and the methods. (ii) "ranges of variation typical of major droughts" (p. 24 l. 7) - I cannot find this classification in the presentation of the results. Could you

please expand?

- Might the perturbation of radiation be more important than temperature perturbations? (because this really indicates a limitation of energy)

- p. 11 l. 11: Figure 3c?

- p. 13 l. 10: there is no Table 4.

- p. 13 l. 16 and 19: Figure 6?

- p. 16 l. 11: How do you end up with 93% with 102 and 105 mm?

- p. 16 l. 6-9: there is no table.

- p. 16 l. 16: do you mean Figure 8?

- p. 17 l.8: observed is "modeled" ?

- p. 5 l. 9: 102 m?

- p. 4 l.3: 1930s ?

———————————————————

---

## Author Comment (AC1) · 1 Nov 2018

Author Response to Peter La Follette Interactive Comment:

The authors appreciate the thoughtful review and critique of the manuscript, with many constructive suggestions. Here we provide a general response to points raised in this review. The following discussion responds in the order points were presented in the review.

**The reviewer notes that simulations lasted for 3 model years and suggests that we discuss the temporal evolution of nonlinearities.**
- We agree that temporal evolution of drought is an important topic; however, in this paper we focus on results in the last year of model simulations in order to emphasize spatial scaling and factor interactions.
- Prior to the runs used for simulation, the model was initialized at an equilibrium state. That is, the simulation was run until annual storage change dropped below a threshold of 1% annual precipitation. Thus, the model began from a steady state configuration.
- Temporal evolution of drought is a large topic in itself and we feel that it is out of the scope of the present study; however, we will add text clarifying that we focus on the last (3rd) year and explaining the rationale.

**The reviewer notes that the caption of Figure 12 does not state exactly what "nonlinearity" is plotted.**
We agree this should be more explicitly stated and will revise to clarify that the nonlinearity is a comparison between the multi-factor hot/dry run versus the single factor hot run and dry run.

**The reviewer notes that the paper lacks comparisons to measured data, and suggests that we should add such a comparison in the case of antecedent soil moisture.**
- We agree that comparison to data is an important and challenging step of model studies. There are observations that could be used to explore the same research questions that are addressed here; however, none of them are directly comparable to the present model because the present model is not a reconstruction of any historical drought.
- The findings of this model, therefore, cannot be taken as a direct prediction for central North America. Rather, their value lies in suggesting system-scale phenomena such as the nonlinear combination of factors.
- It would be possible to design experimental studies or analyses of existing measurements to further explore these phenomena; however, we feel that is a future research topic and not possible within the scope of this study.
- We will add further discussion of this important question to the paper.

**The reviewer queries whether the nonlinearities could be model artifacts, and suggests further discussion.**
- We acknowledge that all models include simplifications and assumptions that in

some cases produce artifacts, results that lack a physical basis. In this case, we have employed an integrated hydrologic model that minimizes these assumptions because it represents all important processes of the hydrologic cycle using physically based equations, as we discuss in the methods.

- Additionally, the perturbation experiment has the same forcing, subsurface configuration, etc for each case and alters one factor at a time, so that differing model outputs result from a difference in one input only.
- We acknowledge that the precise results of the model do not constitute a direct prediction, but we believe the system-scale results will not be artifacts of the model itself. We will add more discussion of this point.

**The reviewer points out some missing citations in the reference list;** we will add those and check that all cited references are listed.

**The reviewer notes that PRISM climatological data and the model inputs and results have different grid resolutions, and queries how this was handled.**
We agree that PRISM data has a coarser resolution than the model grid. We resampled PRISM rasters to the model grid before preparing forcing data. Additionally, lateral flow of groundwater in the model has the effect of smoothing resolution artifacts. This lateral flow of groundwater is primarily driven by topography, and the topography dataset is at the model resolution. We will add text to the methods clarifying this point.

**The reviewer asks for more details about how the drought forcing data was prepared.**
- We began with hourly North American Land Data Assimilation System (NLDAS) reconstructions of temperature and precipitation from a baseline water year. For our baseline we chose water year 1984, which is one of the most average water years for the United States in recent decades. We then increased temperature and decreased precipitation using anomalies drawn from a major drought in the region.
- To find the drought anomalies, we used PRISM data for water year 1934, a year of severe drought, and the 1920s, the non-drought, immediately preceding decade. We took months of water year 1934 to represent a "drought January" "drought February" etc. We averaged months of the 1920s to arrive at a baseline for that region at that time, a "non-drought January" "non-drought February" etc.
- We compared the months to create anomalies. For example, we subtracted "non-drought January" temperature from "drought January" temperature to find the January temperature anomaly. The averaging and subtraction was done for each pixel of the model grid, producing a spatial map for each month.
- As the last step, we modified the hourly baseline temperature data by adding the anomaly for the appropriate month in each cell.
- Precipitation data was processed in the same way except that we found a percent change for each month instead of an absolute difference, to avoid negative precipitation values.
- We will add a clarification of this point to the methods.

**The reviewer asks for more detail regarding crop types and their representation, and suggests we could include a more detailed parameterization of the crops.**

- We agree that different crops will affect the details of drought evolution in different ways, but feel it is important to note that ParFlow-CLM is not an agricultural model. Each cell is assigned 1 vegetation type and all crops are represented as the same "croplands" vegetation type. This is an approximation, as is our choice to remove all vegetation in these cells for a simulated crop failure.
- Analyzing the details of crop type and its impact is, we believe, outside the scope of the present study and not critical to address the study questions.
- We will add a brief discussion of this point to the methods.

**The reviewer notes that Figure 12a shows that the median of the boxplots increases slightly in nonlinearity across scales, and queries whether this is a contradiction to our point that "nonlinearity is less at large scales."**

- We agree that the median nonlinearity becomes more positive for runoff, and more negative for ET as scale increases. Our interpretation focuses on the spread of the boxplots, and on noting that extremely nonlinear responses happen more at small scales.
- As the scale increases, these responses average out to become less extreme, as shown by the decreasing spread of the boxplots.
- In a subcontinental basin as a whole, there is indeed a small positive nonlinearity in runoff: The change in runoff under both temperature and precipitation increase is slightly larger than that due to the separate effects of temperature and precipitation. However, in small basins, the change in runoff due to both variables can be much smaller or larger than expected.
- We will add further discussion to clarify this point.

**The reviewer queries why Water Year 1984 was selected as our baseline;** we will clarify 1984 was an average water year in the United States.

**The reviewer identified a number of typos;** we appreciate this and will make the corrections in revision.

---

## Referee Comment (RC2) · Anonymous Referee #2 · 2 Nov 2018

The paper titled "Unravelling the impacts of precipitation, temperature, and land-cover change for extreme drought over the North American High Plains" uses the ParFlow-CLM model and a set of synthetic climate scenarios to investigate the role of different extreme climate conditions and their interactions in the severity of drought. The novelty in the paper is in its objective to quantify how comprehensively integrating subsurface flow into the modeling of the land surface acts to enhance or reduce drought conditions and the role that temperature, precipitation, and land cover play in those emergent conditions. The paper is appropriate for HESS and I believe it will eventually be suitable for publication. However, the scenarios need to be revisited and rerun before publication.

[Figure]

Main issues: The meteorological scenarios used in this study are physically inconsistent. One cannot just add or subtract from the temperature without accounting for how that impacts the rest of the meteorological states or fluxes. For example, downward longwave radiation is very strongly correlated to temperature and specific humidity is strongly connected to temperature. As such, if you are modifying one you have to modify the others. If not, the meteorology with which you force the model is mostly fantasy and not realistic. There are methods out there to deal with this. This needs to be addressed before publication by first making physically consistent meteorological scenarios and then rerunning the simulations.

Other comments:

Abstract, Line 10 - physically based equations? A lot of land surface and hydrologic models have physically-based equations. I really don't understand the need to use this terminology here. ParFlow is indeed an advanced groundwater model that uses our best understanding of how to model the flow of subsurface water over regional scales. However, that does not mean that other models do not use physically based equations. As long as models are not just complete black boxes, they will always have physically based equations. Those equations might be over-simplistic but they are still physically-based. In my opinion, the better analogy of ParFlow is the "superparameterization" term used in climate models. Anyway, I don't say this to lessen the value of ParFlow but using this terminology distracts the reader from the real strengths of ParFlow.

Abstract, Line 13 - I don't think there was a need to run such a complex model to prove that decreases in precipitation will cause the largest negative anomalies in the evapotranspiration, soil moisture, streamflow, and water table levels. The only flux that would compete with precipitation would be shortwave radiation. However, fortunately for us, the sun is not going anywhere anytime soon.

Page 1, Line 22 - There are studies over the past few years that show that drought recurrence has changed very little over the past half century. Droughts have devastating impacts regardless of if we consider climate change or not. The value of studying the impact of subsurface flow on droughts is critical regardless of the impact of climate change.

Page 3, Line 6 - It is true that the different meteorological factors are strongly interconnected. However, there is a reason for that. The states and fluxes are inherently interconnected. You can't think about precipitation without thinking about temperature. And you especially can't think about specific humidity, radiation fluxes, among others without thinking about temperature. As such, although separating them is a nice thought experiment it is not any realistic representation of what is going on in the observed physical system.

Page 3, Line 27 - Could we move away from the term "fully integrated hydrologic model"? I understand that you are reemphasizing the coupled groundwater, land surface model, surface flow advantages of the model. However, "fully integrated" has a very comprehensive meaning that leads to expecting too much from ParFlow-CLM and leads to stronger criticisms for the model than there should be. Fully integrated would mean a complete characterization of urban hydrology, water management, dynamic coupling with the ecosystem processes (i.e., ecohydrology), complete representation of flooding dynamics, among many many other processes. Again, ParFlow-CLM has significant advantages that should not be discounted or disregarded over more simplistic approaches and those should be emphasized. But you can do that in a more direct way instead of broad terminology such as "fully integrated".

Page 4, Line 23 - Van Genuchten functions should be Van Genuchten water retention curve. Also, it would be a good idea to cite the corresponding VG paper.

Page 5, Line 2 - It is certainly true that VIC does not model the lateral flow between macroscale grid cells and this is a feature that should have been addressed a long time ago. Thankfully most land surface models are finally moving in the right direction and starting to include groundwater flow between macroscale cells. In any case, VIC does

account for sub-grid flow of water even though it only does so implicitly in the variable infiltration curve. Stating that there is no lateral flow at all is not entirely accurate.

Page 5, Line 7 - I agree with this statement to a certain extent. Although it is fairly comprehensive down to the 1km scale grid scale (if we disregard the role of faults and karst terrain in the regional flow), it still completely disregards finer scales. There are plenty of hydrologic processes that play a fundamental role in ecosystem functionand structure that are far below the spatial resolution that you capture in ParFlow. As a result, there is still an underlying need to parameterize meter-scale coupling between the water, energy, and biogeochemical cycles. CLM does that somewhat but it is certainly still far from the objective.

Page 6, Line 6 - This should say sink-filled instead of processed. However, why is this necessary? The sinks in elevation maps can be real features. For example depressions in the landscape certainly exist. This is one of the large advantages in my opinion of ParFlow vs more simplified models since it allows you to directly model those depressions through subsurface redistribution or through accumulation over the surface and redistribution. Maybe I am misunderstanding something here. Please clarify.

Page 6, Line 17 - This is problematic. You can't assume pre-industrial conditions in water management but then use land use change from the past 30 years. They are fairly inconsistent. The agriculture over the plains is inherently dependent on irrigation. If you are going to have one you shouldn't throw out the other especially given the argument of a fully integrated approach. As you revisit the scenarios I would recommend also trying to address this inconsistency.

Page 6, Line 25 - Recursively? Do you mean repeatedly? Recursively has a very specific meaning in computing that doesn't make much sense here.

Table 2 - It is important to realize that the moment that you start varying the climatic conditions you will start also varying the response of the biogeochemical cycles. Certain plant species will die while others will flourish. As a result, the land use scenarios

that you use are inconsistent with what would actually occur with your prescribed forcing. I don't expect you to account for this because this is actually a problem in climate models. However, it might be a good topic to address this explicitly in the discussion section.

Page 13, Line 23 - I would say not strongly correlated. "Independent" is too strong of a word here.

Page 22, Line 16 - I would stay km-scale instead of small scale. That term is fairly nebulous and means different things to different communities. For example, small scale in watershed hydrology is cm to meter scale.
* * *

---

## Author Comment (AC2) · 16 Nov 2018

Author Response to Anonymous Reviewer #1

The authors appreciate the thoughtful review and critique of the manuscript, and the many constructive suggestions provided by the reviewer. Here we provide a general response, addressing points in the order they were presented in the review. Review comments are in bold face.

**(1) Message and relevance.**
**The manuscript misses a substantial differentiation between meteorological and hydrological drought. More precisely, it misses the differentiation between (i) the impact of meteorological drought and (ii) the response of hydrological drought to the former. I am referring to meteorological drought as hot and dry atmospheric forcings, and to hydrological drought as e.g. water shortages and anomalously low groundwater levels, soil moisture, ET and runoff. The combination of both aspects, along with a focus on the nonlinearity of feedbacks, would make this manuscript worth publishing. The authors do, however, only address the first aspect, i.e. they evaluate the impact of meteorological drought conditions (hot, dry, hot and dry) on an annually averaged water balance without referring to the land surface/subsurface state.**
In the current manuscript we chose to focus our analysis on the propagation of meteorological droughts through the hydrologic system (i.e. into a hydrologic drought). While we agree that it would also be interesting to apply the same meteorological stresses to a system that is already in a state of hydrologic drought this is outside the scope of our current analysis.  Here we specifically focus on the distinguishing the impacts of various meteorological drought signatures on ET and quantify the role of lateral groundwater flow in the response, because this is where we feel our integrated modeling approach provides the most novel results to the existing body of literature. However, in Figures 4,5,7 & 8 we do also quantify all of the hydrologic drought metrics outlined by the reviewer (i.e. groundwater levels soil moisture, ET and runoff).  For space reasons, we kept this analysis to an annual level. We agree with the reviewer that with this focus we may be missing an opportunity to comment more directly on other hydrologic drought impacts.  We appreciate the suggestion and plan the following revisions to better highlight the connections between hydrologic and meteorological drought which are present in our simulations:
- We will expand the introduction to include this terminology and discuss existing research on pathways for drought evolution from meteorological to hydrologic drought.
- We will expand the results section by adding a new subsection at the beginning, that focuses specifically on characterizing the impacts to the water balance in more detail and quantifying the relative importance of each hydrologic drought characteristic under different meteorological drought conditions.
- We will revise the title of the manuscript to better clarify that we are evaluating the propagation from meteorological drought to hydrologic drought as follows: "Evaluating the relative importance of precipitation, temperate and land-cover

changes in the hydrologic response to extreme meteorological drought conditions in the North American High Plains".

**And according to their own introduction, the fact that precipitation deficits are the main driver for (hydrological) drought, is not novel. Yet, the authors do not explicitly show that their model does simulate a hydrological drought and how the forcing perturbations impact this drought.**
**This is why, at the end of the manuscript, the reader is left wondering what the impacts of anomalous dry and/or hot conditions on an existing drought are, and even more simple, if a hot temperature anomaly alone can initiate or aggravate a hydrological drought**.
In Figures 4,5,7&8 we directly show the impacts of a hot temperature anomaly on ET, soil moisture, groundwater depth and streamflow. In Figure 7 we compare the impacts of the hot temperature anomaly to the dry anomaly and the combined hot and dry anomaly for all of the variables mentioned above to quantify the relative importance of temperature and precipitation stresses as the reviewer suggests. We agree with the reviewer that separating out these impacts is an important step and we feel that is one of the strengths of this modeling approach. Perhaps we are misunderstanding this comment though. If the main point of the reviewer is that we are not considering the impacts of hot and dry anomalies on aggravating an existing drought, the reviewer is correct in pointing out that we chose to evaluate impacts relative to non-drought conditions rather than starting from a system that was already stressed. Still we are able to do the types of relative comparison that the reviewer is outlining here. We do appreciate the comment and we agree that we could be more explicit in our characterization of hydrologic and meteorological drought and that the manuscript would benefit from a more detailed quantification and analysis of the hydrologic drought that we simulate. Per our response above, we plan to expand our background and results sections to better highlight these points and re-frame our discussion in terms of hydrologic and meteorological drought.

**I personally like the analyses of the nonlinearity of feedbacks, which should, in my opinion, be the main topic of the manuscript and could help to increase relevance. Specifically, what would be really interesting, is the combination of both drought aspects along with the nonlinearity analysis. This would intuitively lead to interesting questions, such as: Does the nonlinearity of the land surface feedbacks to meteorological drought forcing aggravate or dampen the hydrological drought? Does it change extremes (as indicates in title!)? How does it impact severity and extent of hydrological drought? How does land use buffer the impact of (the nonlinear feedbacks of) meteorological drought on hydrological drought? The simulation experiments seem to be designed to address exactly these questions, but the manuscript does not.**
We appreciate the suggestions and agree that the nonlinearity analysis is a novel component of our work which we should perhaps highlight more strongly. Because we are not coupled to an atmospheric model for these simulations we cannot directly evaluate some of the reviewer's suggested research questions here that refer to

feedbacks from the hydrologic system up to the atmosphere. We like the ideas suggested here though and in response to this comment we will do the following:

- Currently, nonlinearity is addressed under our research question 2, in results section 3.2. We will expand the discussion and directly evaluate the effects of vegetation change/land use on hydrologic response (i.e. the final question suggested by the reviewer.

  In the title we used the term 'extreme' drought because we have selected a historically extreme drought perturbation to apply to our system not because we were evaluating the role of hydrologic drought in changing this extreme meteorological condition. In response to this comment and the previous comments we will revise the title as to: "Evaluating the relative importance of precipitation, temperate and land-cover changes in the hydrologic response to extreme meteorological drought conditions in the North American High Plains".

**(2) Methodology and presentation.**
**In addition, both, methodology and analyses would benefit from a more precise description. In the following, I will list a couple of (important) issues that remain unclearto me and hampered my understanding:**
**- Model selection. I understand the advantage of ParFlow as a numerical, physics-based model which simulates lateral flow over other models, such as VIC and SWAT, and the advantage is clear from the description. I do, however, not see the need to "badmouth" other models if they are neither being used and compared, nor validated against observations.**
We completely agree that there is no need to 'badmouth' different modeling approaches and that was not our intent here. We added discussion of VIC and SWAT approaches because they are frequently used in similar studies of drought. Our intent was to be clear about the key differences in the physical approaches of these models and the specific advantages that ParFlow for the questions we wanted to ask in this study. We also noted in this section that ParFlow is the most computationally demanding and that this is a limitation for our approach. Indeed, our results suggest that less complex models may be adequate for large scale questions, in some circumstances. We will revise this section to ensure that we are not doing this in a way that disparages any other models, but simply weighs advantages and disadvantages.

**Moreover, I do not understand the comparison of lateral flow/ no lateral flow influence on ET in Fig. 13. This is not connected to the research questions proposed and setup and results are not well explained. Do you apply a constant water table in the free drainage runs? If so, did you perform a separate spinup for those runs? Or might the differences in Fig. 13 simply arise because you have different water table depths?**
We included the free draining runs to address our third research question, on spatial scaling and complexity. Many of the results in Section 3.3 suggest that model responses at large scales are linear and predictable, which in turn raises the question whether big-picture questions, at subcontinental scales, could be addressed without integrated hydrologic modeling and its km-scale complexity. The free-draining run tests

this by removing interactions between cells. Without lateral flow, the grid cells can be considered as a package of single column models, run across various soil types, slopes and land cover.

- We did not apply a constant water table or a separate spinup for the free draining run, and it does have lower water tables than the other runs. We accounted for this by using a free draining baseline run in calculating free draining ET impacts; however, it is still possible that a generally lower water table resulted in a water limited system and decreased ET once plant transpiration stopped. We will revise the methods to clarify our setup of the free draining run, and add details about the effect of a lower water table on the differences in Fig 13.
- We tested the same big-picture question (what is the average effect of crop disturbance on ET?) with and without lateral flow. The results in Fig 13 showed that crop disturbance increased ET in the normal case, but decreased ET in the free draining case. This is an example where even a large scale prediction depends on the representation of lateral flows and interactions within the model. We will revise to include more discussion of how the free draining runs relate to the research questions and overall point of the paper.

**- Numerical experiments. The use of different experiment names, e.g. "Hot and Dry", "hot/dry", "Hot and Dry" (are they all the same "Hot and Dry" run (6) from Tab. 2?) is really confusing and makes it hard to follow. Please unify.**
We will review for consistent terminology in the revised manuscript.

**- Simulation period and dependency on land surface/subsurface state. I am totally confused about the simulation period and the setup of the numerical experiments: Which years are simulated? Why is the model set up with data from 1984?**
We have summarized our explanation in response to these questions below. We will also expand the methodology to include these points in the revised manuscript.

- We use synthetic drought scenarios that are not an exact reconstruction of any historical drought, but rather an example of severe meteorological drought. We start with a baseline water year, then add perturbations singly and in combination, as shown in the manuscript Table 1.
- We used Water Year 1984 as the baseline because it is one of the most average water years in the United States in recent decades.
- The forcing data was prepared as described below:
    - We began with hourly North American Land Data Assimilation System (NLDAS) reconstructions of temperature and precipitation from a baseline water year. For our baseline we chose water year 1984, which is one of the most average water years for the United States in recent decades. We then increased temperature and decreased precipitation using anomalies drawn from a major drought in the region.
    - To find the drought anomalies, we used PRISM data for water year 1934, a year of severe drought, and the 1920s, the non-drought, immediately preceding decade. We took months of water year 1934 to represent a

"drought January" "drought February" etc. We averaged months of the 1920s to arrive at a baseline for that region at that time, a "non-drought January" "non-drought February" etc.

- o We compared the months to create anomalies. For example, we subtracted "non-drought January" temperature from "drought January" temperature to find the January temperature anomaly. The averaging and subtraction was done for each pixel of the model grid, producing a spatial map for each month.
- o As the last step, we modified the hourly baseline temperature data by adding the anomaly for the appropriate month in each cell.
- o Precipitation data was processed in the same way except that we found a percent change for each month instead of an absolute difference, to avoid negative precipitation values.

**Do you simulate a hydrological drought period, and if so, why do you not evaluate the impact of your forcing perturbations on evolution and extent of drought?**

- ● We do not reconstruct any specific historical drought; however, the forcing is derived from the 1930s, which was one of the most extreme meteorological and hydrological droughts recorded in the study area. We feel it is reasonable to expect significant hydrological impacts from this meteorological forcing, and we characterize those impacts in the results section as discussed below.
- ● We agree that temporal evolution of drought is an important topic; however, in this paper we focus on annually averaged results in the last year of model simulations in order to emphasize spatial scaling and factor interactions. Temporal evolution of drought is a large topic in itself and we feel that it is out of the scope of the present study; however, we will add text clarifying that we focus on the last (3rd) year and explaining the rationale.
- ● We currently characterize the spatially distributed impacts in Section 3.1 with Figure 5. We agree that the spatial extent of impacts is an important topic and we will expand the discussion to include more characterization of the spatial extent/patterns of impacts including various thresholds to characterize the severity of drought impact.

**Even if you do not simulate a hydrological drought, it is still important to evaluate the differences in relation to the land surface and subsurface states. What is the relative importance compared to the natural (modelled) variability? E.g. the reader does not know if a water table difference of ~1 m (Fig. 5) is on the order of natural, (inter-)annual variability and if it occurs in a region of shallow or deep water tables.**

Here too we are not entirely sure what the reviewer means when they say we are not simulating a hydrologic drought. It is true that our simulation starts from a baseline non-drought condition, but we apply severe impacts to precipitation and temperature (i.e. forcing the model with a meteorological drought) in order to simulate a hydrologic drought. That point aside though we appreciate the suggestion and we agree that some

additional discussion of our simulated impacts in the context of model variability would help put results into context.

- We will use our added results section (noted in our first response) to place impacts in context of seasonal and spatial variability within the model. We think this will help us better characterize the severity of our simulated impacts and improve the discussion of hydrologic drought per the reviewer's previous comments.

**- Time scales. The authors only show annually averaged differences, which do not allow to address drought and extreme impacts (as indicated by the title!). The limitation of presenting annual averages becomes evident in Fig. 10, which shows the relation between "antecedent soil moisture" on ET. First of all, I do not really see a "clear break" (p. 17, l. 9).**

To keep our discussion and figure count manageable we chose to focus on spatial variability more than temporal variability (in this case seasonal oscillations). We highlight the impact of spatial averaging on muting the simulated impacts similar to what the reviewer is noting for temporal averaging. As noted in earlier responses, the term 'extreme' refers to the drought perturbation we chose to apply. In Figure 10, approximately 350 mm of soil moisture marks a transition between energy limited and water limited behavior in both the hot and the dry run. We will revise the discussion to use a different term to describe this instead of "clear break."

**Secondly, do you use an annual average as antecedent soil moisture? Soil moisture varies at much shorter time scales; and a grid cell (region) might move from an energy limited towards a soil moisture limited state within a year, and especially during a drought. Maybe it makes sense to look at shorter time scales...otherwise I do not see the merit of Fig. 10.**

- We acknowledge that annual averages will mask the most extreme values of any given year, and that soil moisture has large variation within a year. We chose to use annual averages in order to condense a large amount of data (hourly values for each cell of the model grid) from each run into a summary that would allow us to focus on the project research questions: comparisons between runs that show interactions between variables, nonlinearity and spatial scaling.
- The merit of Figure 10 is in providing a summary of water limited versus energy limited behavior. We agree that it does not address annual variability; however, it provides a general illustration of the mechanism of energy limitation at work in the model, which is used in later discussion to explain some of the nonlinear responses.
- We will change the antecedent soil moisture to be the value at the beginning of the year, rather than the annual average.
- We will improve and expand the discussion to better connect Figure 10 and its mechanism to later sections, to clarify its place in the analysis.

**Whether a grid cell is soil moisture or energy limited also depends on the soil texture, doesn't it? I am not sure that this can be as simplified as the text does it.**

- Cells become moisture limited when the demand for evaporation exceeds the available water. Soil texture changes water retention, so two soils with the same water content but different textures might have different amounts of water available for evaporation. Thus, in a few cases, the same water content might lead one cell to be energy limited but another to be water limited.
- We agree that plotting all points together regardless of texture is a simplification. However, we argue that this simplification does not change the overall message of the figure; some cells are water limited, some are energy limited, and the pre-existing soil moisture affects the response of model cells to forcing changes.

**- Spatial scales. Sec. 3.3 remains unclear to me, though it could be potentially very interesting. This may be mainly due to my lack of understanding what is shown in Fig. 12. Could you please clarify? Does Fig. 12 show the same comparison as in Fig. 9 but the sum? But then, which scenario is shown?**
- We agree this should be more explicitly stated and will revise the caption of Figure 12, to clarify that the nonlinearity is a comparison between the multi-factor hot/dry run versus the single factor hot run and dry run.
- As the reviewer surmised, the boxplots show the spread of the data that is plotted in Figure 9 a), c), e) and g), at a variety of scales. We will clarify that connection in the text.

**What are HUC6 and HUC8 basins? And which are the major basins? Which basins do you actually show in Fig. 5?**
- HUC-6 and HUC-8 stand for Hydrologic Unit Code levels, a system used to classify drainages in the United States. HUC-8 basins are smaller basins that nest within HUC-6 drainages.
- We use the term major basins to mean the Arkansas, Red, and Missouri subcontinental basins.
- Figure 5 and other spatial maps show HUC-6 basins such as the North Platte, South Platte, Upper Cimarron, Republican, etc.
- We appreciate the reminder that not all readers are familiar with this terminology, and will add a clarification.

**(3) Minor points.**
**- If I understand correctly, "anomalies" are neither climatological anomalies, e.g. of soil moisture or runoff to determine drought extent, nor are they the anomalies of forcing during the 1934 (?) drought, which are used to perturb the forcing. If I understand correctly, anomalies in the manuscript are simply the difference between a scenario and the baseline simulation. Please clarify and consider rephrasing.**
This is correct; we use the term "anomalies" to describe the difference between a scenario and the baseline simulation. [We also use the term "impacts" and will review to ensure that the language is clear throughout the discussion.

**- A lot of references are missing (e.g., Loon 2015; Eltahir 1998; Seneviratne 2010;**

**Betts 1996; Koster 2004; McEvoy 2016;...)!**
We will review the manuscript to ensure that all citations appear in the reference list.

**- p. 11. l. 7-9: The description of Fig. 5 in the text is misleading. If I see this correctly**
**(and as is later on in the manuscript mentioned), Hot (and Crops) have higher ET!**
**Also, please be precise what "lower" WTD means...it's deeper?**
We appreciate being alerted to this oversight. The Hot and Crops runs have slightly higher ET and we will revise accordingly. Lower WTD means a water table depth further below the land surface.

**- Fig. 9: The differences in Fig. 9 (c-d-e-f) are not percentages, are they? 0.4 % difference in Fig. 9 seems rather small and not significant (or do you mean 40% as**
**mentioned in the text on e.g. p. 17?).**
This is correct; c-d-e-f show fractional differences. We will revise the legend.
**What do grey colors in Fig. 9 mean?**
Grey colors show a limited amount of extreme values that fall outside the color scale. We set the color scale to show the most relevant variability, which required excluding some outliers.
**Do I understand correctly, that Fig. 9 shows (a,c,e,g): Hot (3) + Dry (4) - "Hot and Dry" (6) ? (b,d,f,h): Hot (3) + Dry (4) + Crops (5) - "Worst case" (7) ? (with numbers referring to runs from Table 2)**
- Subpanels a,c,e,g compare the hot and dry run (6) to the individual hot (3) and dry (4) runs and subpanels b,d,f, and h compare the worst case run (7) to the hot and dry (6) and crops (5) runs.
- The nonlinearity is calculated as described on page 9, lines 13-24. First, individual impacts were calculated for each run relative to the baseline, by simply subtracting baseline values. Then a fractional difference was found between the impacts in a multifactor run, and the expected impacts found by adding together individual runs.
- Red colors in Figure 9 mean that the multifactor impact was smaller than expected, and blue colors mean that it was larger.
- We will expand the caption and/or discussion to clarify.

**- There are no tables representing results; hence the paragraphs on p. 13 l. 8-11 and p. 16 - l. 6-9 are confusing. Please remove/rephrase.**
This discussion is supposed to refer to the data in Figures 7 and 8 (which was presented in table form in an earlier draft); we will revise and review the text for similar oversights.

**- p. 14 l. 8-10: this relates to the forcing perturbations, does it? So, if I sum up the precipitation changes from Fig. X over the domain, they correspond to a total change of 40% ?**
This is correct; we will clarify the sentence.

**- p. 17 l. 17-18: Is this related to the results presented in Fig. 9c? (Here, radiation probably has a stronger impact than temperature).**
This is correct; Figure 10 illustrates a mechanism that relates to Figure 9c. We also note that radiation did not change between the baseline and drought scenarios.

**- Conclusions. (i) contradiction of "real world scenario" to what you describe in the**
**introduction and the methods.**
We agree that the model is not reconstructing a real world scenario and will rephrase the sentence. Our intent is to convey that real world scenarios include multiple factors, not to claim that this model represents a usable reconstruction of some known drought.

**(ii) "ranges of variation typical of major droughts" (p. 24 l. 7) - I cannot find this classification in the presentation of the results. Could you please expand?**
This phrase refers to the development of the drought scenarios, which use temperature, precipitation and vegetation changes derived from a historical drought scenario. It is meant to acknowledge that the relative importance of the factors holds for the ranges tested in the study, but not necessarily for ranges outside those simulated. For example, if temperature was increased far enough, it could have a larger impact than a small precipitation change. We will clarify this point in the methods and discussion.

**- Might the perturbation of radiation be more important than temperature perturbations? (because this really indicates a limitation of energy)**
Radiation was not perturbed in the study; temperature was the only change between the baseline and the hot scenario. We will clarify this point in the methods.

**- p. 11 l. 11: Figure 3c?**
Correct; figure 3c. We will revise the sentence.

**- p. 13 l. 10: there is no Table 4.**
This refers to a table that was removed in an earlier draft; we will delete this passage.

**- p. 13 l. 16 and 19: Figure 6?**
Figure 5 refers to the spatial map of anomalies; figure 6 plots some of these anomalies against the forcing anomalies that produced them. There is a typo in the current draft which should read: "variability in the sensitivity...is illustrated in **Figure 6** which plots...Variability in **Figure 6** shows that the same perturbation can result in a wide range of responses"

**- p. 16 l. 11: How do you end up with 93% with 102 and 105 mm?**
There is a typo in the current draft; it should read 105 mm, which is 93% of **114** mm. (The percentage calculation is also done with unrounded values)

**- p. 16 l. 6-9: there is no table.**

As noted above, this should refer to Figures 7 and 8.

**- p. 16 l. 16: do you mean Figure 8?**
Correct, this is supposed to say Figure 8.
**- p. 17 l.8: observed is "modeled" ?**
Correct, we will revise the sentence.

**- p. 5 l. 9: 102 m?**
Correct, we will revise the typo.

**- p. 4 l.3: 1930s ?**
Correct, we will revise the typo.

---

## Author Comment (AC3) · 16 Nov 2018

Author Response to Anonymous Reviewer #2
The authors appreciate the thoughtful review and critique of the manuscript, with many constructive suggestions. Here we provide a general response, addressing points in the order they were presented in the review. Review comments are in bold face.

**The paper titled "Unravelling the impacts of precipitation, temperature, and land-cover change for extreme drought over the North American High Plains" uses the ParFlow-CLM model and a set of synthetic climate scenarios to investigate the role of different extreme climate conditions and their interactions in the severity of drought. The novelty in the paper is in its objective to quantify how comprehensively integrating subsurface flow into the modeling of the land surface acts to enhance or reduce drought conditions and the role that temperature, precipitation, and land cover play in those emergent conditions. The paper is appropriate for HESS and I believe it will eventually be suitable for publication. However, the scenarios need to be revisited and rerun before publication.**

**Main issues: The meteorological scenarios used in this study are physically inconsistent. One cannot just add or subtract from the temperature without accounting for how that impacts the rest of the meteorological states or fluxes. For example, downward longwave radiation is very strongly correlated to temperature and specific humidity is strongly connected to temperature. As such, if you are modifying one you have to modify the others. If not, the meteorology with which you force the model is mostly fantasy and not realistic. There are methods out there to deal with this. This needs to be addressed before publication by first making physically consistent meteorological scenarios and then rerunning the simulations.**

- We agree the meteorological changes are an approximation, but we believe they are suitable for the proposed research questions in this paper. For the goals of this study, we are not simulating a projected climate change (with, for example, a Global Climate Model prediction) nor are we predicting the actual drought. This is a difference between the present study and many published studies, as outlined in the literature review. Instead, we are using one-factor perturbations to the model (forcing, land cover) in a systematic way to run numerical experiments.
- By making a single change at a time, we can attribute any differences between the baseline and the perturbed run to the single variable that was perturbed. This is an advantage of modeling studies over real world observations, as we can assess process interaction with much greater precision and detail. If we changed humidity and pressure with associated temperature changes we would be modifying three things at once, which limit the linearity arguments and the strength of the experiment as such.
- It is worth noting that the model runs are computationally expensive. For this paper, each year used about a week of wall-clock time and over 20,000 processor hours. This expense means that understanding individual impacts is

valuable from a modeling perspective, to lay a foundation before proceeding to more complex simulations.
- Other studies have also used the approach of changing individual variables or small groups of variables. For example, Kollet and Maxwell perturbed temperature without changing other meteorological variables for a study of drought in the Little Washita watershed. Rasmussen et al (2011) also employed a simplified approach they called "pseudo-global warming", adding estimated climate perturbations in temperature, vapor mixing ratio, boundary layer height and windspeed to a forcing dataset. Markovich et al (2016) perturbed temperature alone in a similar study of climate change in California, and Pribulick et al (2016) perturbed temperature and land cover in studying the impacts of vegetation change under global warming in a Colorado watershed. While the reviewer is correct that changing one meteorological variable does not fully represent the physical system, it is a documented simplification used in multiple published papers.
- We will explain these points more clearly and add more details and justification on this modeling approach and perturbations to the methods section. We will also note that model runs with changes in all meteorological forcing is a good possibility for future study.

**Other comments:**
**Abstract, Line 10 - physically based equations? A lot of land surface and hydrologic models have physically-based equations. I really don't understand the need to use this terminology here. ParFlow is indeed an advanced groundwater model that uses our best understanding of how to model the flow of subsurface water over regional scales.**
**However, that does not mean that other models do not use physically based equations. As long as models are not just complete black boxes, they will always have physically based equations. Those equations might be over-simplistic but they are still physically- based. In my opinion, the better analogy of ParFlow is the "superparameterization" term used in climate models. Anyway, I don't say this to lessen the value of ParFlow but using this terminology distracts the reader from the real strengths of ParFlow.**
We agree with the reviewer that most models have some physical basis for their equations. We used the term "physically based" to distinguish from statistically based models. While we believe this is useful shorthand for the abstract, we appreciate the suggestion from the reviewer and we don't want this terminology to distract from our message. We will expand our discussion in the methods to make it clear that we understand ParFlow is not the only physically based model.

**Abstract, Line 13 - I don't think there was a need to run such a complex model to prove that decreases in precipitation will cause the largest negative anomalies in the evapotranspiration, soil moisture, streamflow, and water table levels. The only flux that would compete with precipitation would be shortwave radiation. However, fortunately for us, the sun is not going anywhere anytime soon.**

Indeed, the first research question has been addressed in other literature; reviewing impacts of different factors with this model shows that our results are consistent with other literature. Additionally, in this exercise we are able to quantify the impacts in detail, rather than making qualitative arguments. Characterizing the impacts in general terms is a preliminary to more detailed analysis of nonlinearity and scaling of these impacts in research questions 2 and 3.

**Page 1, Line 22 - There are studies over the past few years that show that drought recurrence has changed very little over the past half century. Droughts have devastating impacts regardless of if we consider climate change or not. The value of studying the impact of subsurface flow on droughts is critical regardless of the impact of climate change.**
This is a good point; we will expand the discussion on importance to include it.

**Page 3, Line 6 - It is true that the different meteorological factors are strongly interconnected. However, there is a reason for that. The states and fluxes are inherently interconnected. You can't think about precipitation without thinking about temperature. And you especially can't think about specific humidity, radiation fluxes, among others without thinking about temperature. As such, although separating them is a nice thought experiment it is not any realistic representation of what is going on in the observed physical system.**
We agree with the reviewer's point, but perhaps not their conclusion. In addition to referring back to our response to this point, above, we would like to note that our model goal is not to represent all the complexity of meteorology, but to test narrowly defined specific scenarios for the purpose of a numerical experiment. The intent of experiments like these is not to represent the observed physical system in all ways, but rather to lay the groundwork for more complex experiments and for interpreting observations in the future. We have attempted to be transparent in these goals and the simplifications we intentionally made to the system throughout the manuscript. We appreciate the reviewer's comments and we will make sure that the limitations of this simplification and our reasons for making it are more clear in the revised manuscript.

**Page 3, Line 27 - Could we move away from the term "fully integrated hydrologic model"? I understand that you are reemphasizing the coupled groundwater, land surface model, surface flow advantages of the model. However, "fully integrated" has a very comprehensive meaning that leads to expecting too much from ParFlow-CLM and leads to stronger criticisms for the model than there should be. Fully integrated would mean a complete characterization of urban hydrology, water management, dynamic coupling with the ecosystem processes (i.e., ecohydrology), complete representation of flooding dynamics, among many many other processes. Again, ParFlow-CLM has significant advantages that should not be discounted or disregarded over more simplistic approaches and those should be emphasized. But you can do that in a more direct way instead of broad terminology such as "fully integrated".**

We will replace "fully integrated" with "integrated hydrologic model" for greater clarity, as the reviewer suggests, and add a brief discussion of terminology to the methods.

**Page 4, Line 23 - Van Genuchten functions should be Van Genuchten water retention curve. Also, it would be a good idea to cite the corresponding VG paper.**
We will revise to make it clear that Van Genuchten functions refers to the Van Genuchten pressure-saturation and pressure-relative permeability relationship, and add a citation to the Van Genuchten 1980 paper.

**Page 5, Line 2 - It is certainly true that VIC does not model the lateral flow between macroscale grid cells and this is a feature that should have been addressed a long time ago. Thankfully most land surface models are finally moving in the right direction and starting to include groundwater flow between macroscale cells. In any case, VIC does account for sub-grid flow of water even though it only does so implicitly in the variable infiltration curve. Stating that there is no lateral flow at all is not entirely accurate.**
We will make the statement more precise to state that VIC "does not model lateral flow between macroscale grid cells", as suggested.

**Page 5, Line 7 - I agree with this statement to a certain extent. Although it is fairly comprehensive down to the 1km scale grid scale (if we disregard the role of faults and karst terrain in the regional flow), it still completely disregards finer scales. There are plenty of hydrologic processes that play a fundamental role in ecosystem function and structure that are far below the spatial resolution that you capture in ParFlow. As a result, there is still an underlying need to parameterize meter-scale coupling between the water, energy, and biogeochemical cycles. CLM does that somewhat but it is certainly still far from the objective.**
We agree this is an important point and will add some discussion on fine-scale processes that are not captured in the current representation.

**Page 6, Line 6 - This should say sink-filled instead of processed. However, why is this necessary? The sinks in elevation maps can be real features. For example depressions in the landscape certainly exist. This is one of the large advantages in my opinion of ParFlow vs more simplified models since it allows you to directly model those depressions through subsurface redistribution or through accumulation over the surface and redistribution. Maybe I am misunderstanding something here. Please clarify.**
This is correct; the terrain was processed with a sink-filling algorithm and we will be more explicit about this in the revised manuscript. We agree with the reviewer that natural depressions in the landscape exist; however, literature suggests that it is difficult to distinguish these natural depressions and sinks from noise within the processed DEM (eg Kenny et al 2008). For the simulations presented here we use the kinematic wave approximation of the overland flow equations and therefore we require a domain with a connected drainage network. Additionally, we would like to note that this approach is

consistent with the previous work modeling the continental United States by Maxwell and Condon (2016) that our domain was built from.

**Page 6, Line 17 - This is problematic. You can't assume pre-industrial conditions in water management but then use land use change from the past 30 years. They are fairly inconsistent. The agriculture over the plains is inherently dependent on irrigation. If you are going to have one you shouldn't throw out the other especially given the argument of a fully integrated approach. As you revisit the scenarios I would recommend also trying to address this inconsistency.**
We agree that the agriculture over the plains depends on irrigation, and that this water use is very different from pre-industrial water management. Ideally, we agree that water use and land use would be consistent within the model; unfortunately, there is limited pre-industrial data on land use. In the case of this project, we are not reconstructing any specific historical drought, so it is less important in addressing the research questions to match all forcings and settings to one period of time.

**Page 6, Line 25 - Recursively? Do you mean repeatedly? Recursively has a very specific meaning in computing that doesn't make much sense here.**
We will change recursively to repeatedly for clarity.

**Table 2 - It is important to realize that the moment that you start varying the climatic conditions you will start also varying the response of the biogeochemical cycles. Certain plant species will die while others will flourish. As a result, the land use scenarios that you use are inconsistent with what would actually occur with your prescribed forcing. I don't expect you to account for this because this is actually a problem in climate models. However, it might be a good topic to address this explicitly in the discussion section.**
This is an important point and we will add discussion of this point to the paper.
We would also note that the historical Dustbowl was marked by widespread crop failures and massive topsoil loss. While we are not reconstructing the Dustbowl, this does imply that our land use scenario of replacing "crops" with "bare soil" is not unreasonable in the face of extreme drought.

**Page 13, Line 23 - I would say not strongly correlated. "Independent" is too strong of a word here.**
We will make the suggested terminology change.

**Page 22, Line 16 - I would stay km-scale instead of small scale. That term is fairly nebulous and means different things to different communities. For example, small scale in watershed hydrology is cm to meter scale.**
We agree that km-scale is more precise and will revise to clarify.

References Cited

Maxwell, R. M., & Condon, L. E.: Connections between groundwater flow and transpiration partitioning, *Science*, 353(6297), 377-380, 2016.

Maxwell, R. M., & Kollet, S. J.: Interdependence of groundwater dynamics and land-energy feedbacks under climate change. *Nature Geoscience*,1(10), 665-669, 2008

Kenny, F., Matthews, B., & Todd, K.: Routing overland flow through sinks and flats in interpolated raster terrain surfaces. *Computers & Geosciences* 34(11), 1417-1430, 2008

Markovich, K.H., Maxwell, R.M. and Fogg, G.E., Hydrogeological response to climate change in alpine hillslopes. *Hydrological Processes*, 30(18), 3126-3138, 2016

Pribulick, C.E., Foster, L.M., Bearup, L.A., Navarre-Sitchler, A.K., Williams, K.H., Carroll, R.W. and Maxwell, R.M. Contrasting the hydrologic response due to land cover and climate change in a mountain headwaters system. *Ecohydrology*, 9(8), 1431-1438, 2016

Rasmussen, R., Liu, C., Ikeda, K., Gochis, D., Yates, D., Chen, F., Tewari, M., Barlage, M., Dudhia, J., Yu, W. and Miller, K. High-resolution coupled climate runoff simulations of seasonal snowfall over Colorado: A process study of current and warmer climate. *Journal of Climate*, 24(12), 3015-3048, 2011

---

## Author Response (AR1)

Dear Dr Wanders,

On behalf of all coauthors, I submit our revised manuscript "Evaluating the relative importance of precipitation, temperature and land-cover change in the hydrologic response to extreme meteorological drought conditions over the North American High Plains" for your consideration in HESS.

The authors would like to thank you, Peter LaFollette and the two anonymous reviewers for constructive criticism and review. In response, we have carried out extensive revisions of the manuscript.

In particular, we have expanded the methods section with a discussion of forcing perturbations and added supporting literature for the method of single factor perturbations. We rephrased the introduction and results sections in terms of meteorological and hydrological drought, and added an additional section to the results with two new figures to place drought impacts in context of seasonal and spatial variability. We included more in-depth discussion of the specific impact of land cover change on the transition from meteorological to hydrologic drought.

On the following pages, please find our point-by-point response to reviewers and list of all changes, followed by the marked-up manuscript. A clean version of the revised manuscript has been uploaded separately.

Thank you for your time and consideration.

Sincerely,

Annette Hein

**Hein et al, list of revisions:**

The authors thank the reviewers and editor for their constructive feedback on the manuscript. Here we present the revisions of the manuscript in response to the comments.

Note that line numbers in the reviewer comments refer to the original manuscript, whereas line numbers in the author responses refer to the Word manuscript with tracked changes.

In presenting our revisions, unchanged text is shown in normal format, deleted text is presented with a strikethrough and added text is underlined.

The list of responses and revisions is presented first, followed by the manuscript showing markup.

**Editor Comments:**

Comments to the Author:

Dear Authors,

From my own reading of the manuscript and the reviewers' comment I would like to recommend major revisions for the manuscript to be published in HESS. Following the point by point review and my own reading, there seem to be two major points that I would advise to addressed in the manuscript, either by additional simulation, analysis or discussion.

- The consistency in the forcing data a suggested by Reviewer #2, either needs to be addressed or discussed in the revised manuscript. I understand that computational demand will be an issue, but if feasible it would be a great addition to the paper to look into the effect of the different meteorological components and their correlated impacts.

- The translation from meteorological to hydrological drought needs to be addressed. As stated by reviewer #1, their are not linearly related and I feel this study could be research that really contributes and help to improve our understanding in the propagation from extreme meteorological to hydrological droughts.

I would encourage the authors to use the reviewers' comments and improve the manuscript.

Kind regards,

Niko

**Author Response:**

We thank the editor for the recommendation and appreciate the comments.

**In response to the first point about consistency of forcing data, we agree the**

meteorological changes are an approximation, but we believe they are suitable for the proposed research questions in this paper. For the goals of this study, we are not simulating a projected climate change (with, for example, a Global Climate Model prediction) nor are we predicting the actual drought. Instead, we are using one-factor perturbations to the model (forcing, land cover) in a systematic way to run numerical experiments. By making a single change at a time, we can attribute any differences between the baseline and the perturbed run to the single variable that was perturbed. This is an advantage of modeling studies over real world observations, as we can assess process interaction with much greater precision and detail. If we changed humidity and pressure with associated temperature changes we would be modifying three things at once, which limit the linearity arguments and the strength of the experiment as such. It is worth noting that the model runs are computationally expensive. For this paper, each year

It is worth noting that the model runs are computationally expensive. For this paper, each year used about a week of wall-clock time and over 20,000 processor hours. This expense means that understanding individual impacts is valuable from a modeling perspective, to lay a foundation before proceeding to more complex simulations.

Other studies have also used the approach of changing individual variables or small groups of variables. For example, Kollet and Maxwell (2008) perturbed temperature without changing other meteorological variables for a study of drought in the Little Washita watershed. Rasmussen et al (2011) also employed a simplified approach they called "pseudo-global warming", adding estimated climate perturbations in temperature, vapor mixing ratio, boundary layer height and windspeed to a forcing dataset. Markovich et al (2016) perturbed temperature alone in a similar study of climate change in California, and Pribulick et al (2016) perturbed temperature in a Colorado watershed. While the reviewer is correct that changing one meteorological variable does not fully represent the physical system, it is a documented simplification used in multiple published papers.

In response to the first point, we have revised the manuscript with a discussion of these points in the methods section (page 12, line 2 to page 12, line 27) as detailed below in response to the reviewer.

In response to the second point about meteorological and hydrological drought, we note that in the current manuscript, we chose to focus our analysis on the propagation of meteorological droughts through the hydrologic system (i.e. into a hydrologic drought). While we agree that it would also be interesting to apply the same meteorological stresses to a system that is already in a state of hydrologic drought, this is outside the scope of our current analysis. Here we specifically focus on the distinguishing the impacts of various meteorological drought signatures on ET and quantify the role of lateral groundwater flow in the response, because this is where we feel our integrated modeling approach provides the most novel results to the existing body of literature.

However, in Figures 6,7,9 & 10 (formerly Figures 4,5,7 and 8) we do also quantify all of the hydrologic drought metrics outlined by the reviewer (i.e. groundwater levels, soil moisture, ET and runoff). In Figure 9 (formerly Figure 7) we compare the impacts of the hot temperature anomaly to the dry anomaly and the combined hot and dry anomaly for all of the variables mentioned above to quantify the relative importance of temperature and precipitation stresses, as the reviewer suggests. We agree with the reviewer that separating out these impacts is an important step and we feel that is one of the strengths of this modeling approach. We do appreciate the comment and we agree that we could be more explicit in our characterization of hydrologic and meteorological drought and that the manuscript would benefit from a more detailed quantification and analysis of the hydrologic drought that we simulate.

In response to the second point, we have revised the title of the manuscript, added an additional section to the results, and rewritten the introduction and results sections in order to highlight the connections between hydrologic and meteorological drought which are present in our simulations. Revisions are detailed below in response to the reviewer.

We believe that we have strengthened the manuscript by incorporating both major concerns into the revision, and we thank the editor and reviewers for their comments.

Non-public comments to the Author: Dear authors, I had some minor addition comments for you to consider. Regards, Niko

Introduction, it might be very insightful to provide some schematic overview of the effect of P and T on ET and ultimately drought. This is just a suggestion. Author Response:

While we have elected not to add another figure due to space, we do discuss the conceptual impact in some detail in the introduction, (see, eg, Page 2, line 12 to page 3, line 6) and have rephrased the discussion to emphasize the transition from meteorological to hydrologic drought.

**Page 4 line 11-15 some sentence are not correct here.**

**...selected because IT employs a MORE...**

**.. Is therefore well suited TO SIMULATE the water and....**

**Revision:**

Page 5, Line 7: revised as follows:

"ParFlow-CLM was selected because it employs a more extensive and physically based representation of subsurface processes than many other hydrologic models and is therefore well suited to simulate the water and energy dynamics that occur during drought."

**Page 4 line 30 change to (SWAT, Neisch..) Revision:**

Page 5, line 29: Revised in response to this and other comments:

"It is instructive to compare ParFlow with the Variable Infiltration Capacity (VIC) model (VIC 2016; Liang et al. 1994) and the Soil Water Assessment Tool (SWAT,) Neisch et al. 2011)..."

**Page 5 line 14 102km should be 102m Revision:**

Page 6, line 17: revised as follows:

"The domain is 1200 by 1124 km and extends to a depth of 102 km, with 5 layers for a total of 6,744,000 computational cells."

**Page 19 annual average or antecedent sm? Slightly confusing Revision:**

Figure 12: This was addressed in other revisions to the figure. As detailed below, the Figure 12 (formerly Figure 10) x-axis has been revised to show antecedent soil moisture from day 1 of year 3 of the simulations.

**Figure 12 could more scale be included? Would be interesting to see some cutoffs, maybe drainage area as a metric? Then give 25-75 range instead of boxplot. This could help inform others on characteristic lengths of linearity.**

**Author Response:** We feel that the approach of aggregating results by river basin is valuable because it follows the natural scaling of the landscape, with small drainages nesting inside larger ones. However, we agree that the 25-75 range may be more informative than the boxplot for conveying the figure's message.

**Revision:**

Figure 14: We replotted Figure 14 (formerly Figure 12) to show the 25-75% range of the data instead of complete boxplots.

· · ·

**Comments and revisions for Reviewer 1:**

**RC: (1) Message and relevance.**

The manuscript misses a substantial differentiation between meteorological and hydrological drought. More precisely, it misses the differentiation between (i) the impact of meteorological drought and (ii) the response of hydrological drought to the former. I am referring to meteorological drought as hot and dry atmospheric forcings, and to hydrological drought as e.g. water shortages and anomalously low groundwater levels, soil moisture, ET and runoff. The combination of both aspects, along with a focus on the nonlinearity of feedbacks, would make this manuscript worth publishing. The authors do, however, only address the first aspect, i.e. they evaluate the impact of meteorological drought conditions (hot, dry, hot and dry) on an annually averaged water balance without referring to the land surface/subsurface state.

**Author Response**: In the current manuscript we chose to focus our analysis on the propagation of meteorological droughts through the hydrologic system (i.e. into a hydrologic drought). While we agree that it would also be interesting to apply the same meteorological stresses to a system that is already in a state of hydrologic drought this is outside the scope of our current analysis. Here we specifically focus on the distinguishing the impacts of various meteorological drought signatures on ET and quantify the role of lateral groundwater flow in the response, because this is where we feel our integrated modeling approach provides the most novel results to the existing body of literature.

However, in Figures 6,7,9 & 10 (formerly Figures 4,5,7 and 8) we do also quantify all of the hydrologic drought metrics outlined by the reviewer (i.e. groundwater levels, soil moisture, ET and runoff). For space reasons, we originally kept this analysis to an annual level. We agree with the reviewer that with this focus we may be missing an opportunity to comment more directly on other hydrologic drought impacts.

**Revisions:**

We made the following revisions to better highlight the connections between hydrologic and meteorological drought which are present in our simulations:

Title: We revised the title of the manuscript to clarify that we are evaluating the propagation from meteorological drought to hydrologic drought as follows:

"Unravelling the impacts Evaluating the relative importance of precipitation, temperature and land-cover change in the hydrologic response to for extreme meteorological drought conditions in the North American High Plains".

Page 1, line 25: We added the following text:

"Because there are many ways to characterize drought, researchers often make a distinction between meteorological and hydrologic drought (Van Loon, 2015). Meteorological drought is defined as weather changes such as decreased precipitation or increased temperature. These changes may produce hydrologic drought, which is defined as impacts to the hydrologic system such as decreased runoff or soil moisture."

Introduction, Results: The text throughout these sections was rephrased in terms of meteorological and hydrologic drought.

Page 1, line 26 and References: We added the following citation: "Van Loon, A.F. Hydrological drought explained. WIREs Water, 2(4) 359-392, 2015."

Page 15, line 4 to page 17, line 6: We expanded the results section by adding a new subsection at the beginning, that focuses specifically on characterizing the impacts to the water balance in more detail under different meteorological drought conditions. This section includes two new figures showing seasonal changes in the water balance and a spatial snapshot of drought impacts.

RC: And according to their own introduction, the fact that precipitation deficits are the main driver for (hydrological) drought, is not novel. Yet, the authors do not explicitly show that their model does simulate a hydrological drought and how the forcing perturbations impact this drought.

**This is why, at the end of the manuscript, the reader is left wondering what the impacts of anomalous dry and/or hot conditions on an existing drought are, and even more simple, if a hot temperature anomaly alone can initiate or aggravate a hydrological drought.**

Author Response: In Figures 6,7,9 and 10 (formerly figures 4,5,7&8) we directly show the impacts of a hot temperature anomaly on ET, soil moisture, groundwater depth and streamflow. In Figure 9 (formerly Figure 7) we compare the impacts of the hot temperature anomaly to the dry anomaly and the combined hot and dry anomaly for all of the variables mentioned above to quantify the relative importance of temperature and precipitation stresses as the reviewer suggests. We agree with the reviewer that separating out these impacts is an important step and we feel that is one of the strengths that this modeling approach brings to our manuscript. Perhaps we are misunderstanding this comment though. If the main point of the reviewer is that we are not considering the impacts of hot and dry anomalies on aggravating an existing drought, the reviewer is correct in pointing out that we chose to evaluate impacts relative to nondrought conditions rather than starting from a system that was already stressed. Still we are able to do the types of relative comparison that the reviewer is outlining here. We do appreciate the comment and we agree that we could be more explicit in our characterization of hydrologic and meteorological drought and that the manuscript would benefit from a more detailed quantification and analysis of the hydrologic drought that we simulate. **Revisions:**

Introduction, Results: As detailed above, the introduction and results sections have been expanded to better highlight these points and re-frame our discussion in terms of hydrologic and meteorological drought.

RC: I personally like the analyses of the nonlinearity of feedbacks, which should, in my opinion, be the main topic of the manuscript and could help to increase relevance. Specifically, what would be really interesting, is the combination of both drought aspects along with the nonlinearity analysis. This would intuitively lead to interesting questions, such as: Does the nonlinearity of the land surface feedbacks to meteorological drought forcing aggravate or dampen the hydrological drought? Does it change extremes (as indicates in title!)? How does it impact severity and extent of hydrological drought? How does land use buffer the impact of (the nonlinear feedbacks of) meteorological drought on hydrological drought? The simulation experiments seem to be designed to address exactly these questions, but the manuscript does not.

**Author Response:** We appreciate the suggestions and agree that the nonlinearity analysis is a novel component of our work which we should perhaps highlight more strongly. Because we are not coupled to an atmospheric model for these simulations we cannot directly evaluate some of the reviewer's suggested research questions here that refer to feedbacks from the hydrologic system up to the atmosphere. We like the ideas suggested here though and in response to this comment we have made the following changes:

**Revisions:**

Page 24, line 13 to page 24, line 25: Currently, nonlinearity is addressed under our research question 2, in results section 3.3. We expanded the discussion and directly evaluate the effects of vegetation change/land use on hydrologic response (i.e. the final question suggested by the reviewer.)

Title: Additionally, as noted above, we have revised the title to emphasize the concepts of meteorological and hydrologic drought.

**RC: (2) Methodology and presentation.**

In addition, both, methodology and analyses would benefit from a more precise description. In the following, I will list a couple of (important) issues that remain unclear to me and hampered my understanding:

- Model selection. I understand the advantage of ParFlow as a numerical, physics-based model which simulates lateral flow over other models, such as VIC and SWAT, and the advantage is clear from the description. I do, however, not see the need to "badmouth" other models if they are neither being used and compared, nor validated against observations.

**Author Response:** We completely agree that there is no need to 'badmouth' different modeling approaches and that was not our intent here. We added discussion of VIC and SWAT approaches because they are frequently used in similar studies of drought. Our intent was to be clear about the key differences in the physical approaches of these models and the specific advantages that ParFlow for the questions we wanted to ask in this study. We also noted in this section that ParFlow is the most computationally demanding and that this is a limitation for our approach. Indeed, our results suggest that less complex models may be adequate for large scale questions, in some circumstances. We have revised this section to be a more clearly neutral weighing of advantages and disadvantages.

**Revisions**:**

Page 5, line 27-page 6, line 2: Revised as follows:

"ParFlow has a number of differences with commonly used models in other drought-related studies. It is instructive to compare ParFlow with, the Variable Infiltration Capacity (VIC) model (VIC 2016; Liang et al. 1994) and the Soil Water Assessment Tool (SWAT,) Neisch et al. 2011), not in order to criticize any model, but to illustrate the reasoning for model choice in this study. ParFlow allows any number of subsurface layers with any specified conductivity, and has vertical and lateral flow driven by pressure gradients. Soil moisture and groundwater are not distinguished; both are represented through pressure in a cell and solved for using Richards' Equation. This 3 dimensional variably saturated flow is the main difference between ParFlow and other models. While this detailed representation is an advantage for this study, it also leads to higher computational expense in ParFlow runs when compared to other models."

**RC: Moreover, I do not understand the comparison of lateral flow/ no lateral flow influence on ET in Fig. 13. This is not connected to the research questions proposed and setup and results are not well explained. Do you apply a constant water table in the free drainage runs? If so, did you perform a separate spinup for those runs? Or might the differences in Fig. 13 simply arise because you have different water table depths? Author Response: We included the free draining runs to address our third research question,**

on spatial scaling and model complexity. Many of the results in Section 3.4 suggest that model responses at large scales are linear and predictable, which in turn raises the question whether big-picture questions, at subcontinental scales, could be addressed without integrated hydrologic modeling and its km-scale complexity. The free-draining run tests this by removing interactions between cells. Without lateral flow, the grid cells can be considered as a package of single column models, run across various soil types, slopes and land cover; this configuration behaves as a more simple model.

•We did not apply a constant water table or a separate spinup for the free draining run, and it does have lower water tables than the other runs. We accounted for this by using a free draining baseline run in calculating free draining ET impacts; however, it is still possible that a generally lower water table resulted in a water limited system and decreased ET once plant transpiration stopped.

•We tested the same big-picture question (what is the average effect of crop disturbance on ET?) with and without lateral flow. The results in Fig 15 (formerly Fig 13) showed that crop disturbance increased ET in the normal case, but decreased ET in the free draining case. This

is an example where even a large scale prediction depends on the representation of lateral flows and interactions within the model.

**Revisions:**

Page 9, line 13 to page 10, line 5: The Methods section was revised as follows: "Two further runs were also conducted to explore the importance of lateral flow as a mechanism within the model as part of addressing the third research question on spatial scaling and complexity. Commonly used models including VIC and SWAT do not allow lateral flow within the model, and including this process makes the model computationally more expensive. Creating normal runs with lateral flow and free-draining runs (i.e., without lateral flow) allows exploration of how this process affects model results. To construct a free-draining run, the water table was set at the base of the domain and all overland and subsurface lateral flow processes were turned off, while vertical flow through the soil column and water table remained. (No separate spinup was conducted for the free-draining run, which has a lower water table than the other runs. To account for this, the free draining drought run was compared to a free draining baseline.) With these settings, ParFlow-CLM mimics a traditional land surface model as described in Maxwell and Condon (2016)."

Page 34, line 13 to page 34, line 35: The discussion was revised to add details about the effect of a lower water table on the differences in Fig 13, and to include more discussion of how the free draining runs relate to the research questions and overall point of the paper.

**RC: - Numerical experiments. The use of different experiment names, e.g. "Hot and Dry", "hot/dry", "Hot and Dry" (are they all the same "Hot and Dry" run (6) from Tab. 2?) is really confusing and makes it hard to follow. Please unify.**

Author Response: We will review for consistent terminology in the revised manuscript. Revisions:

Throughout the manuscript, Run 6 is now referred to as the "hot/dry" run.

**RC- Simulation period and dependency on land surface/subsurface state. I am totally confused about the simulation period and the setup of the numerical experiments: Which years are simulated? Why is the model set up with data from 1984?**

Author Response: We have summarized our explanation in response to these questions below. We will also expand the methodology to include these points in the revised manuscript.
We use synthetic drought scenarios that are not an exact reconstruction of any historical drought, but rather an example of severe meteorological drought. We start with a baseline water year, then add perturbations singly and in combination, as shown in the manuscript Table 1.

•We used Water Year 1984 as the baseline because it is one of the most average water years in the United States in recent decades.

•The forcing data was prepared as described below:

O We began with hourly North American Land Data Assimilation System (NLDAS) reconstructions of temperature and precipitation from a baseline water year. For our baseline we chose water year 1984, which is one of the most average water years for the United States in recent decades. We then increased temperature and decreased precipitation using anomalies drawn from a major drought in the region.

O To find the drought anomalies, we used PRISM data for water year 1934, a year of severe drought, and the 1920s, the non-drought, immediately preceding decade. We took months of water year 1934 to represent a "drought January" "drought February" etc. We averaged months of the 1920s to arrive at a baseline for that region at that time, a "non-drought January" "non-drought February" etc.

O We compared the months to create anomalies. For example, we subtracted "nondrought January" temperature from "drought January" temperature to find the January temperature anomaly. The averaging and subtraction was done for each pixel of the model grid, producing a spatial map for each month.

O As the last step, we modified the hourly baseline temperature data by adding the anomaly for the appropriate month in each cell.

O Precipitation data was processed in the same way except that we found a percent change for each month instead of an absolute difference, to avoid negative precipitation values.

**Revisions:**

Page 11, line 2 to page 12, line 2: the methods were revised to incorporate the above points.

**RC: Do you simulate a hydrological drought period, and if so, why do you not evaluate the impact of your forcing perturbations on evolution and extent of drought? Author Response:**

•We do not reconstruct any specific historical drought; however, the forcing is derived from the 1930s, which was one of the most extreme meteorological and hydrological droughts recorded in the study area. We feel it is reasonable to expect significant hydrological impacts from this meteorological forcing, and we characterize those impacts in the results section as discussed below.

•We agree that temporal evolution of drought is an important topic; however, in this paper we focus on annually averaged results in the last year of model simulations in order to emphasize spatial scaling and factor interactions. Temporal evolution of drought is a large topic in itself and we feel that it is out of the scope of the present study; however, we will add text clarifying that we focus on the last (3rd) year and explaining the rationale.

•We currently characterize the spatially distributed impacts in Section 3.1 with Figure 5. We agree that the spatial extent of impacts is an important topic and we will expand the discussion to include more characterization of the spatial extent/patterns of impacts including various thresholds to characterize the severity of drought impact.

**Revisions:**

Page 10, line 9: The following text was added to the methods section:

"Although the simulation included three years, the analysis focuses on annually averaged results in the third and last year of model simulations in order to emphasize spatial scaling and factor interactions. Temporal evolution of drought is a large topic in itself and while interesting, falls outside the scope of the present study."

Page 16, line 4 to page 17, line 6: The added results section and new Figure 5 address spatial impacts of results.

**RC: Even if you do not simulate a hydrological drought, it is still important to evaluate the differences in relation to the land surface and subsurface states. What is the relative importance compared to the natural (modelled) variability? E.g. the reader does not know if a water table difference of 1 m (Fig. 5) is on the order of natural, (inter-)annual variability and if it occurs in a region of shallow or deep water tables.**

**Author Response:** Here too we are not entirely sure what the reviewer means when they say we are not simulating a hydrologic drought. It is true that our simulation starts from a baseline non-drought condition, but we apply severe impacts to precipitation and temperature (i.e. forcing the model with a meteorological drought) in order to simulate a hydrologic drought. That point aside though we appreciate the suggestion and we agree that some additional discussion of our simulated impacts in the context of model variability would help put results into context. **Revisions:**

Page 15, line 4 to Page 17, line 6: The added results section 3.1 places impacts in context of seasonal (Figure 4) and spatial (Figure 5) variability within the model. We believe this section better characterizes the severity of our simulated impacts and improves the discussion of hydrologic drought per the reviewer's previous comments.

**RC: - Time scales. The authors only show annually averaged differences, which do not allow to address drought and extreme impacts (as indicated by the title!). The limitation of presenting annual averages becomes evident in Fig. 10, which shows the relation between "antecedent soil moisture" on ET. First of all, I do not really see a "clear break" (p. 17, I. 9).**

**Author Response:** To keep our discussion and figure count manageable we chose to focus on spatial variability more than temporal variability (in this case seasonal oscillations). We highlight the impact of spatial averaging on muting the simulated impacts similar to what the reviewer is noting for temporal averaging. As noted in earlier responses, the term 'extreme' refers to the drought perturbation we chose to apply. In Figure 12 (formerly Figure 10), approximately 350 mm of soil moisture marks a transition between energy limited and water limited behavior in both the hot and the dry run.

**Revisions:**

Page 25, line 17: The passage was revised as follows:

"The transition clear break in both panels at about 350 mm of soil water content indicates the importance of antecedent soil moisture. Above this transition point cutoff, increasing temperature produces the largest increases in ET."

**RC: Secondly, do you use an annual average as antecedent soil moisture? Soil moisture varies at much shorter time scales; and a grid cell (region) might move from an energy limited towards a soil moisture limited state within a year, and especially during a drought. Maybe it makes sense to look at shorter time scales...otherwise I do not see the merit of Fig. 10.**

**Author Response:** We agree with the reviewer that short time scales are an important part of the system behavior. Our model is run with an hourly time step and writes hourly output; however, we have chosen to summarize this large amount of information using annual averages.

We acknowledge that annual averages will mask the most extreme values of any given year, and that soil moisture has large variation within a year. We chose to use annual averages in order to condense a large amount of data (hourly values for each cell of the model grid) from each run into a summary that would allow us to focus on the project research questions: comparisons between runs that show interactions between variables, nonlinearity and spatial scaling.

The merit of Figure 12 (previously Figure 10) is in providing a summary of water limited versus energy limited behavior. We agree that it does not address annual variability; however, it provides a general illustration of the mechanism of energy limitation at work in the model, which is used in later discussion to explain some of the nonlinear responses.

While we believe the annual average approach to be generally valuable, we will revise the axis of this figure to capture a snapshot in time at the start of the simulation instead of an annual average.

**Revisions:**

Figure 12: (Previously figure 10) We changed the soil moisture axis to be the value at the first day of the simulation year 3, rather than the annual average across the year. We note that this change does not materially impact the discussion or conclusions based on the figure.

Page 25, line 8: The passage was revised in response to this and other comments:

"Antecedent soil moisture and water limited behaviour may explain some of the nonlinearity shown in Figure 11.Figure 12 Figure 10 shows the observed modeled ET anomalies versus antecedent soil moisture for a given forcing change (color scale), providing a general illustration of these mechanisms within the model."

**RC: Whether a grid cell is soil moisture or energy limited also depends on the soil texture, doesn't it? I am not sure that this can be as simplified as the text does it.**

**Author Response**: Cells become moisture limited when the demand for evaporation exceeds the available water. Soil texture changes water retention, so two soils with the same water content but different textures might have different amounts of water available for evaporation. Thus, in a few cases, the same water content might lead one cell to be energy limited but another to be water limited.

We agree that plotting all points together regardless of texture is a simplification. However, we argue that this simplification does not change the overall message of the figure; some cells are water limited, some are energy limited, and the pre-existing soil moisture affects the response of model cells to forcing changes.

**Revisions:**

Page 25, line 11: This text was added to the discussion of Figure 12:

"In this figure, all cells are plotted regardless of soil texture. This is a slight simplification as soil texture changes water retention, so two soils with the same water content but different textures might have different amounts of water available for evaporation. Thus, in a few cases, the same water content might lead one cell to be energy limited but another to be water limited as the demand for evaporation exceeds the available water. This does not change the overall finding that pre-existing soil moisture partially controls the response of model cells to forcing changes."

**RC: - Spatial scales. Sec. 3.3 remains unclear to me, though it could be potentially very interesting. This may be mainly due to my lack of understanding what is shown in Fig. 12. Could you please clarify? Does Fig. 12 show the same comparison as in Fig. 9 but the sum? But then, which scenario is shown?**

Author Response: We agree this should be more explicitly stated and will revise the caption to clarify that the nonlinearity is a comparison between the multi-factor hot/dry run versus the single factor hot run and dry run.

As the reviewer surmised, the boxplots show the spread of the data that is plotted in Figure 11 (formerly Figure 9) a), c), e) and g), at a variety of scales.

**Revisions:**

Figure 14: The caption of Figure 14 (formerly Figure 12) was revised as follows:

"Factors combine in a more linear way at larger scales. The multi-factor hot/dry run is compared to the single factor hot and dry runs. Each panel shows boxplots that characterize deviations from linearity in the hot/dry run from model cell to subcontinental scales. Panel (a) shows runoff, (b) shows ET, (c) shows soil water content and (d) shows the water table."

Page 30, line 13: We clarified the connection between figure 11 (formerly figure 9) and figure 14 (formerly figure 12):

"Figure 12 Figure 14 examines this nonlinearity across a variety of spatial scales, comparing the multifactor hot/dry run to the single factor hot and dry runs. The boxplots show the spread of the data plotted in the left-hand column of Figure 11, averaged at several different scales. Overall, Figure 14 shows that the nonlinearity is much decreased in moderately sized (HUC-6) river basins, and small in subcontinental river basins."

**RC: What are HUC6 and HUC8 basins? And which are the major basins? Which basins do you actually show in Fig. 5?**

**Author Response:** HUC-6 and HUC-8 stand for Hydrologic Unit Code levels, a system used by the US Geological Survey to classify drainages in the United States. HUC-8 basins are smaller basins that nest within HUC-6 drainages. We use the term major basins to mean the Arkansas, Red, and Missouri subcontinental basins. Figure 5 and other spatial maps show HUC-6 basins such as the North Platte, South Platte, Upper Cimarron, Republican, etc. We appreciate the reminder that non-United States readers may be unfamiliar with this terminology, and will add a clarification.

**Revisions:**

Page 20, line 4:This text was added:

"(Basins in Figure 5 are HUC-6 basins such as the North Platte, South Platte, Upper Cimarron, Republican, etc. The HUC or Hydrologic Unit Code system is used to classify drainages in the United States.)"

Page 30, line 17: This text was added:

"(HUC-8 basins are smaller basins that nest within HUC-6 drainages. The term major basins is used here to mean the Arkansas, Red, and Missouri subcontinental basins.)"

**RC: (3) Minor points.**

- If I understand correctly, "anomalies" are neither climatological anomalies, e.g. of soil moisture or runoff to determine drought extent, nor are they the anomalies of forcing during the 1934 (?) drought, which are used to perturb the forcing. If I understand correctly, anomalies in the manuscript are simply the difference between a scenario and the baseline simulation. Please clarify and consider rephrasing.

**Author Response:** This is correct; we use the term "anomalies" to describe the difference between a scenario and the baseline simulation. We also use the term "impacts" and will review to ensure that the language is clear throughout the discussion.

**Revisions:**

Throughout: The term "anomaly" was reviewed for consistency throughout the paper.

Page 12, line 30: The following clarification was added:

"The term anomaly is used in this discussion to describe changes to hydrologic processes resulting from the perturbations applied to the simulations. The subtraction produces a simple metric of the model impact."

**RC: - A lot of references are missing (e.g., Loon 2015; Eltahir 1998; Seneviratne 2010; Betts 1996; Koster 2004; McEvoy 2016;...)!**

Author Response: We will review the manuscript to ensure that all citations appear in the reference list.

**Revisions:**

Reference section: The reference list was updated.

**RC: - p. 11. I. 7-9: The description of Fig. 5 in the text is misleading. If I see this Correctly (and as is later on in the manuscript mentioned), Hot (and Crops) have higher ET! Also, please be precise what "lower" WTD means...it's deeper?**

**Author Response:** We appreciate being alerted to this oversight. The Hot and Crops runs have slightly higher ET and we will revise accordingly. Lower WTD means a water table depth further below the land surface.

**Revisions: The passage was revised as follows:**

Page 18, line 3: The passage was revised as follows:

"Relative to the baseline scenario most all of the drought scenarios have decreased runoff and ET, depleted soil water content and lower water tables (ie, water tables that are further below

the land surface), as would be expected in a drought. The exception is the Hot and the Crops runs, which have slightly higher ET as discussed later."

RC - Fig. 9: The differences in Fig. 9 (c-d-e-f) are not percentages, are they? 0.4 % difference in Fig. 9 seems rather small and not significant (or do you mean 40% As mentioned in the text on e.g. p. 17?).

**Author Response:** This is correct; in Figure 11 (formerly Figure 9) a-b-c-d-e-f show fractional differences. We will revise the legend.

**Revisions:**

Figure 11: The legend is updated to reflect that the maps show fractional differences instead of percentages.

**RC:What do grey colors in Fig. 9 mean?**

**Author Response:** Grey colors show a limited amount of extreme values that fall outside the color scale. We set the color scale to show the most relevant variability, which required excluding

some outliers.

**Revisions:** This was clarified in the discussion of Figure 9 in response to the next comment, below.

**RC: Do I understand correctly, that Fig. 9 shows (a,c,e,g): Hot (3) + Dry (4) -"Hot and Dry" (6) ? (b,d,f,h): Hot (3) + Dry (4) + Crops (5) -"Worst case" (7) ? (with numbers referring to runs from Table 2)**

**Author Response:** Subpanels a,c,e,g compare the hot and dry run (6) to the individual hot (3) and dry (4) runs and subpanels b,d,f, and h compare the worst case run (7) to the hot and dry (6) and crops (5) runs. The nonlinearity is calculated as described on page 9, lines 13-24. First, individual impacts were calculated for each run relative to the baseline, by simply subtracting baseline values. Then a fractional difference was found between the impacts in a multifactor run, and the expected impacts found by adding together individual runs.

Red colors in Figure 9 mean that the multifactor impact was smaller than expected, and blue colors mean that it was larger.

**Revisions:** Discussion of Figure 11 (formerly Figure 9) was expanded:

Page 24, line 27: Discussion of Figure 11 (formerly Figure 9) was expanded:

"Figure 11 Figure 9 maps this nonlinearity (percent difference, as described in the methods) across the domain for runoff, ET, soil moisture and the water table. The left hand column compares the hot/ dry run to the individual hot and dry runs, and the right hand column compares the worst case run to the hot/dry and crops runs. The nonlinearity is calculated as a percent difference, as described in the methods. Red colors mean that the multifactor impact was smaller than expected, while blue colors mean that it was larger. Gray denotes a small number of outlier pixels."

**RC: - There are no tables representing results; hence the paragraphs on p. 13 l. 8-11 and p. 16 - l. 6-9 are confusing. Please remove/rephrase.**

**Author Response:** This discussion is supposed to refer to the data in Figures 9 and 10 (previously Figures 7 and 8, which was presented in table form in an earlier draft); we will revise and review the text for similar oversights.

**Revisions:**

Page 20, line 11: revised as follows:

"Lowering precipitation drastically reduced all components of the water budget. ET declined, but was supported at first by declines in soil water content and water table levels. Although the second year was just as dry, Table 4 shows that ET and runoff decreased less than in the first year. Effects were most pronounced in spring and summer. "

**Page 23, line 9: revised as follows**

"Figure 9 Figure 7 and Figure 10 Figure 8 compare the impacts of the single perturbation runs with those in the multi-perturbation runs. Each figure table compares the result of a multi-perturbation run (the "Modeled Effect" column) with the combination of single-perturbation runs (the "Sum" column). In a perfectly linear system, the two results columns would be the same. Instead, t The figurestables show that impacts of each multi-perturbation run can be attributed to individual components, but not completely."

**RC:- p. 14 I. 8-10: this relates to the forcing perturbations, does it? So, if I sum up the precipitation changes from Fig. X over the domain, they correspond to a total change of 40% ?**

Author Response: This is correct; we will clarify the sentence.

**Revisions:**

Page 21, line 8: revised as follows:

"This applies to the relative size of the changes studied; a 2 degree C increase in temperature versus a 40% drop in precipitation (when summed over the domain) and disturbance to land cover over about 30% of the model area."

**RC:- p. 17 I. 17-18: Is this related to the results presented in Fig. 9c? (Here, radiation probably has a stronger impact than temperature).**

**Author Response:** This is correct; Figure 12 (previously Figure 10) illustrates a mechanism that relates to Figure 11c (previously Figure 9c). We also note that radiation did not change between the baseline and drought scenarios.

**Revisions:**

Page 25, line 8: revised as follows:

"Antecedent soil moisture and water limited behaviour may explain some of the nonlinearity shown in Figure 11."

**RC: - Conclusions. (i) contradiction of "real world scenario" to what you describe in The introduction and the methods.**

**Author Response:** We agree that the model is not reconstructing a real world scenario and will rephrase the sentence. Our intent is to convey that real world scenarios include multiple factors, not to claim that this model represents a usable reconstruction of some known drought. **Revisions:**

Page 31, line 3: The phrase was deleted:

"Simulations based after the example of the Dustbowl drought of the 1930s individually perturbed temperature, precipitation and land cover, followed by multi-perturbation runs that combined these changes as would occur in a real-world scenario."

**RC: (ii) "ranges of variation typical of major droughts" (p. 24 l. 7) - I cannot find this classification in the presentation of the results. Could you please expand?**

**Author Response:** This phrase refers to the development of the drought scenarios, which use temperature, precipitation and vegetation changes derived from a historical drought scenario. It is meant to acknowledge that the relative importance of the factors holds for the ranges tested in the study, but not necessarily for ranges outside those simulated. For example, if temperature was increased far enough, it could have a larger impact than a small precipitation change. We will clarify this point in the methods and discussion.

Page 12, line 6: The following clarification was added to the methods:"Finally, real droughts could exceed the ranges of drought simulated within the study (for example, if temperature was increased far enough, it could have a larger impact than a small precipitation change.)"

**RC: - Might the perturbation of radiation be more important than temperature perturbations? (because this really indicates a limitation of energy)**

Author Response: Radiation was not perturbed in the study; temperature was the only change between the baseline and the hot scenario.

**Revisions:** This point was clarified in the methods section in response to Reviewer 2, below.

**RC: - p. 11 I. 11: Figure 3c?**

Author Response: Correct; figure 3c. We will revise the sentence.

**Revisions:**

Page 18, line 7: The sentence now reads: "In the central region where rainfall increased slightly (Figure 3cFigure1c), this extra water partly avoided the most severe impacts."

**RC: - p. 13 I. 10: there is no Table 4.**

Author Response: This refers to a table that was removed in an earlier draft; we will delete this passage.

**Revisions**:**

Page 20, line 11: revised as follows:

"Lowering precipitation drastically reduced all components of the water budget. ET declined, but was supported at first by declines in soil water content and water table levels. Although the second year was just as dry, Table 4 shows that ET and runoff decreased less than in the first year. Effects were most pronounced in spring and summer. "

**RC: - p. 13 l. 16 and 19: Figure 6?**

**Author Response:** Figure 7 (formerly Figure 5) refers to the spatial map of anomalies; Figure 8 (formerly figure 6) plots some of these anomalies against the forcing anomalies that produced them

**Revisions:**

Page 20, line 20: The typo was corrected as follows:

"Variability in the sensitivity to the applied drought anomalies are illustrated in Figure 8 Figure5 which plots the response of individual grid cells <del>pixels</del>, to a given forcing change in temperature or precipitation. Variability in Figure 8 Figure5 shows that the same perturbation can result in a wide range of responses."

**RC: - p. 16 I. 11: How do you end up with 93% with 102 and 105 mm?**

**Author Response:** There is a typo in the current draft; it should read 105 mm, which is 93% of 114 mm. (The percentage calculation is also done with unrounded values)

**Revisions:**

Page 24, line 5: The typo was corrected:

"This amounts to 93% of the 114 <del>102</del> mm/yr decrease that was actually modelled in the hot/dry simulation."

**RC: - p. 16 l. 6-9: there is no table.**

Author Response: As noted above, this should refer to Figures 7 and 8.

**Revisions:**

Revised in response to comments, above.

**RC: - p. 16 l. 16: do you mean Figure 8?**

**Author Response:** On review, this reference is accurate; the sentence refers to Figure 9 (formerly Figure 7) and the linearity of the hot/dry run compared to the individual components. **Revisions:**

Page 24, line 9: the sentence was clarified: "For example, in Figure 9 Figure 7 the hot/dry run is lacking 9 mm/yr of ET and keeps an extra 9 mm/yr of runoff compared to what would be expected by combining individual impacts calculated in the hot run and the dry run."

**RC: - p. 17 I.8: observed is "modeled" ?**

Author Response:Correct, we will revise the sentence. Revisions:

Page 25, line 9 (note that figure 10 is now figure 12)

"Figure 12 Figure 10 shows the modeled observed ET anomalies versus antecedent soil moisture for a given forcing change (color scale)."

**RC: - p. 5 l. 9: 102 m?**

Author Response: Correct, we will revise the typo. Revisions:

Corrected in response to editor's comments above.

**RC:- p. 4 l.3: 1930s ?**

Author Response:Correct, we will revise the typo.

**Revisions:**

Page 5, line 2: The typo was revised:

"The scenarios developed for these numerical experiments were modeled after an example of extreme drought in the region, the Dustbowl of the 1930s1<del>9390s</del>."

- - -

**Comments and revisions for Reviewer 2:**

RC: Main issues: The meteorological scenarios used in this study are physically inconsistent. One cannot just add or subtract from the temperature without accounting for how that impacts the rest of the meteorological states or fluxes. For example, downward longwave radiation is very strongly correlated to temperature and specific humidity is strongly connected to temperature. As such, if you are modifying one you have to modify the others. If not, the meteorology with which you force the model is mostly fantasy and not realistic. There are methods out there to deal with this. This needs to be addressed before publication by first making physically consistent meteorological scenarios and then rerunning the simulations.

**Author Response:** We agree the meteorological changes are an approximation, but we believe they are suitable for the proposed research questions in this paper. For the goals of this study, we are not simulating a projected climate change (with, for example, a Global Climate Model prediction) nor are we predicting the actual drought. This is a difference between the present study and many published studies, as outlined in the literature review. Instead, we are using one-factor perturbations to the model (forcing, land cover) in a systematic way to run numerical experiments.

• By making a single change at a time, we can attribute any differences between the baseline and the perturbed run to the single variable that was perturbed. This is an advantage of modeling studies over real world observations, as we can assess process interaction with much greater precision and detail. If we changed humidity and pressure with associated temperature changes we would be modifying three things at once, which limit the linearity arguments and the strength of the experiment as such.

- It is worth noting that the model runs are computationally expensive. For this paper, each year used about a week of wall-clock time and over 20,000 processor hours. This expense means that understanding individual impacts is valuable from a modeling perspective, to lay a foundation before proceeding to more complex simulations.
- Other studies have also used the approach of changing individual variables or small groups of variables. For example, Kollet and Maxwell (2008) perturbed temperature without changing other meteorological variables for a study of drought in the Little Washita watershed. Rasmussen et al (2011) also employed a simplified approach they called "pseudo-global warming", adding estimated climate perturbations in temperature, vapor mixing ratio, boundary layer height and windspeed to a forcing dataset. Markovich et al (2016) perturbed temperature alone in a similar study of climate change in California, and Pribulick et al (2016) perturbed temperature and land cover in studying the impacts of vegetation change under global warming in a Colorado watershed. While the reviewer is correct that changing one meteorological variable does not fully represent the physical system, it is a documented simplification used in multiple published papers.

**Revisions**:**

Page 12, line 3 to page 12, line 27: The above explanation was incorporated into the Methods section.

Page 38, line 10: This passage was added to the Conclusions:

"Future studies could also combine an individual-factor approach as done here, with a more realistic approach where meteorological variables like pressure or humidity change with temperature, or with complete future climate scenarios."

**RC: Other comments:**

Abstract, Line 10 - physically based equations? A lot of land surface and hydrologic models have physically-based equations. I really don't understand the need to use this terminology here. ParFlow is indeed an advanced groundwater model that uses our best understanding of how to model the flow of subsurface water over regional scales. However, that does not mean that other models do not use physically based equations. As long as models are not just complete black boxes, they will always have physically based equations. Those equations might be over-simplistic but they are still physicallybased. In my opinion, the better analogy of ParFlow is the "superparameterization" term used in climate models. Anyway, I don't say this to lessen the value of ParFlow but using this terminology distracts the reader from the real strengths of ParFlow.

**Author Response**: We agree with the reviewer that most models have some physical basis for their equations. We used the term "physically based" to distinguish from statistically based models. While we believe this is useful shorthand for the abstract, we appreciate the suggestion from the reviewer and we don't want this terminology to distract from our message. **Revisions:**

Page 5, line 13 to page 6, line 13: The discussion of ParFlow was revised to make it clear that we understand ParFlow is not the only physically based model.

**RC: Abstract, Line 13 - I don't think there was a need to run such a complex model to prove that decreases in precipitation will cause the largest negative anomalies in the evapotranspiration, soil moisture, streamflow, and water table levels. The only flux that would compete with precipitation would be shortwave radiation. However, fortunately for us, the sun is not going anywhere anytime soon.**

**Author Response:** Indeed, the first research question has been addressed in other literature; reviewing impacts of different factors with this model shows that our results are consistent with other literature. Additionally, in this exercise we are able to quantify the impacts in detail, rather

than making qualitative arguments. Characterizing the impacts in general terms is a preliminary to more detailed analysis of nonlinearity and scaling of these impacts in research questions 2 and 3.

**Revisions:**

Page 4, line 15: We added this sentence to the introduction:

"This study builds on the previous studies that compare different meteorological factors and their impact on hydrology by quantifying those impacts in detail."

Page 1, Line 22 - There are studies over the past few years that show that drought recurrence has changed very little over the past half century. Droughts have devastating impacts regardless of if we consider climate change or not. The value of studying the impact of subsurface flow on droughts is critical regardless of the impact of climate change.

Author Response: This is a good point and we agree with the reviewer. Revisions:

n/a

RC: Page 3, Line 6 - It is true that the different meteorological factors are strongly interconnected. However, there is a reason for that. The states and fluxes are inherently interconnected. You can't think about precipitation without thinking about temperature. And you especially can't think about specific humidity, radiation fluxes, among others without thinking about temperature. As such, although separating them is a nice thought experiment it is not any realistic representation of what is going on in the observed physical system.

**Author Response**: We agree with the reviewer's point, but perhaps not their conclusion. In addition to referring back to our response to this point, above, we would like to note that our model goal is not to represent all the complexity of meteorology, but to test narrowly defined specific scenarios for the purpose of a numerical experiment. The intent of experiments like these is not to represent the observed physical system in all ways, but rather to lay the groundwork for more complex experiments and for interpreting observations in the future. We have attempted to be transparent in these goals and the simplifications we intentionally made to the system throughout the manuscript. We appreciate the reviewer's comments and we will make sure that the limitations of this simplification and our reasons for making it are more clear in the revised manuscript.

**Revisions:**

Page 12, line 3 to page 12, line 27: The points above were incorporated into the expanded methods section.

RC: Page 3, Line 27 - Could we move away from the term "fully integrated hydrologic model"? I understand that you are reemphasizing the coupled groundwater, land surface model, surface flow advantages of the model. However, "fully integrated" has a very comprehensive meaning that leads to expecting too much from ParFlow-CLM and leads to stronger criticisms for the model than there should be. Fully integrated would mean a complete characterization of urban hydrology, water management, dynamic coupling with the ecosystem processes (i.e., ecohydrology), complete representation of flooding dynamics, among many many other processes. Again, ParFlow-CLM has significant advantages that should not be discounted or disregarded over more simplistic approaches and those should be emphasized. But you can do that in a more direct way instead of broad terminology such as "fully integrated".

**Author Response:** We will replace "fully integrated" with "integrated hydrologic model" for greater clarity, as the reviewer suggests, and add a brief discussion of terminology to the methods.

**Revisions:**

Page 5, line 15: This text was added to the methods:

"The terminology "integrated hydrologic model" used here refers to the globally implicit solution of Richard's Equation and the shallow water equations, thus integrating variably saturated subsurface and overland flow processes. It is not intended to imply that the model includes anthropogenic or biologic processes."

Page 4, line 18: The sentence was deleted as part of a broader revision to the paragraph: "ParFlow is ideal for this study because it is a fully integrated hydrologic model that includes groundwater, soil moisture and overland flow processes."

**RC: Page 4, Line 23 - Van Genuchten functions should be Van Genuchten water retention curve. Also, it would be a good idea to cite the corresponding VG paper.**

**Author Response:** We will revise to make it clear that Van Genuchten functions refers to the Van Genuchten pressure-saturation and pressure-relative permeability relationship, and add a citation to the Van Genuchten 1980 paper.

**Revisions:**

Page 5, line 18: The sentence was revised as follows:

"ParFlow simulates It models saturated and unsaturated flow in three dimensions using Richards' Equation, with flux defined by Darcy's Law and relationships between pressure and relative saturation or permeability defined by the Van Genuchten pressure-saturation and pressure-relative permeability relationships (Van Genuchten, 1980) functions."

Page 5, line 21 and References: The following citation was added:

"Van Genuchten, M.T., 1980. A closed-form equation for predicting the hydraulic conductivity of unsaturated soils 1. Soil Science Society of America Journal, 44(5), 892-898."

RC:Page 5, Line 2 - It is certainly true that VIC does not model the lateral flow between macroscale grid cells and this is a feature that should have been addressed a long time ago. Thankfully most land surface models are finally moving in the right direction and starting to include groundwater flow between macroscale cells. In any case, VIC does account for sub-grid flow of water even though it only does so implicitly in the variable infiltration curve. Stating that there is no lateral flow at all is not entirely accurate. Author Response: We will make the statement more precise to state that VIC "does not model lateral flow between macroscale grid cells", as suggested.

**Revisions:**

Page 6 line 2: The sentence was revised as follows:

"VIC typically has three soil layers and does not simulate lateral flow between macroscale grid cells, although it includes a baseflow term for water leaving a cell to enter a stream."

RC: Page 5, Line 7 - I agree with this statement to a certain extent. Although it is fairly comprehensive down to the 1km scale grid scale (if we disregard the role of faults and karst terrain in the regional flow), it still completely disregards finer scales. There are plenty of hydrologic processes that play a fundamental role in ecosystem function and structure that are far below the spatial resolution that you capture in ParFlow. As a result, there is still an underlying need to parameterize meter-scale coupling between the water, energy, and biogeochemical cycles. CLM does that somewhat but it is certainly still far from the objective.

Author Response: We agree this is an important point and will add some discussion on finescale processes that are not captured in the current representation.

**Revisions:**

Page 6, line 7: The passage was revised as follows:

"In contrast, ParFlow combines all of these processes at a variety of scales; in this model, scales are included from km-scale to subcontinental. (This does omit meter and cm-scale processes such as biogeochemical cycles). The detailed, <del>small scale</del> representation of subsurface processes makes ParFlow-CLM a suitable model to run numerical experiments whose results depend on physical processes and their interactions, as opposed to statistical fitting or simplified parameterizations. "

RC: Page 6, Line 6 - This should say sink-filled instead of processed. However, why is this necessary? The sinks in elevation maps can be real features. For example depressions in the landscape certainly exist. This is one of the large advantages in my opinion of ParFlow vs more simplified models since it allows you to directly model those depressions through subsurface redistribution or through accumulation over the surface and redistribution. Maybe I am misunderstanding something here. Please clarify. Author Response: This is correct; the terrain was processed with a sink-filling algorithm and we will be more explicit about this in the revised manuscript. We agree with the reviewer that natural depressions in the landscape exist; however, literature suggests that it is difficult to distinguish these natural depressions and sinks from noise within the processed DEM (eg Kenny et al 2008). For the simulations presented here we use the kinematic wave approximation of the overland flow equations and therefore we require a domain with a connected drainage network. Additionally, we would like to note that this approach is consistent with the previous work modeling the continental United States by Maxwell and Condon (2016) that our domain was built from.

**Revisions:**

Page 7, line 8: The passage was revised as follows:

"The slopes in the x (east-west) and y (north-south) directions were derived from a digital elevation model and sink-filled processed to ensure the entire domain was connected. The simulations presented here use the kinematic wave approximation of the overland flow equations and therefore require a domain with a connected drainage network, as slopes control surface flow routing. While natural depressions in the landscape exist, literature suggests that it is difficult to distinguish these natural depressions and sinks from noise within the processed DEM (eg Kenny et al 2008). The slopes control surface flow routing."

**RC: Page 6, Line 17 - This is problematic. You can't assume pre-industrial conditions in water management but then use land use change from the past 30 years. They are fairly inconsistent. The agriculture over the plains is inherently dependent on irrigation. If you are going to have one you shouldn't throw out the other especially given the argument of a fully integrated approach. As you revisit the scenarios I would recommend also trying to address this inconsistency.**

**Author Response:** We agree that the agriculture over the plains depends on irrigation, and that this water use is very different from pre-industrial water management. Ideally, we agree that water use and land use would be consistent within the model; unfortunately, there is limited pre-industrial data on land use. In the case of this project, we are not reconstructing any specific historical drought, so it is less important in addressing the research questions to match all forcings and settings to one period of time.

**Revisions**:**

Page 8, line 6: The following was added:

"The use of modern day vegetation is not temporally consistent with pre-industrial water management, but unfortunately, there is limited pre-industrial data on land use. In the case of this project, we are not reconstructing any specific historical drought, so it is less important in addressing the research questions to match all forcings and settings to one period of time."

**RC: Page 6, Line 25 - Recursively? Do you mean repeatedly? Recursively has a very specific meaning in computing that doesn't make much sense here.**

Author Response: We will change recursively to repeatedly for clarity. Revisions:

Page 8, line 17: revised as follows:

"The model was run repeatedly <del>recursively</del> with water year 1984 North American Land Data Assimilation System (NLDAS-2) forcing until average subsurface storage change in one year was less than 1% of precipitation (achieved after three repeated <del>recursive</del> runs)."

Page 10, line 7: revised as follows:

"The baseline run and both free draining runs were conducted for one year; the drought runs were conducted for three years of repeated recursive drought forcing to simulate a transient time period a few years into a hypothetical severe drought."

RC: Table 2 - It is important to realize that the moment that you start varying the climatic conditions you will start also varying the response of the biogeochemical cycles. Certain plant species will die while others will flourish. As a result, the land use scenarios that you use are inconsistent with what would actually occur with your prescribed forcing. I don't expect you to account for this because this is actually a problem in climate models. However, it might be a good topic to address this explicitly in the discussion section. Author Response: This is an important point and we will add discussion of this point to the paper. We would also note that the historical Dustbowl was marked by widespread crop failures and massive topsoil loss. While we are not reconstructing the Dustbowl, this does imply that our land use scenario of replacing "crops" with "bare soil" is not unreasonable in the face of extreme drought.

**Revisions**:**

Page 11, line 24: The following text was added:

"Lastly, vegetation was disturbed for the crops runs by setting all crop or crop mosaic cells to bare soil (this approximation was inspired by the documented representing the massive crop failure that occurred in response to extreme drought conditions during the historic Dustbowl). "Page 12, line 4: The following text was added:

"Additionally the vegetative changes could be more complex, as shifting temperature and rainfall patterns can produce a range of responses across several vegetative types."

**RC: Page 13, Line 23 - I would say not strongly correlated. "Independent" is too strong of a word here.**

**Author Response:** We will make the suggested terminology change. **Revisions:**

Page 20, line 26: revised as follows:

"However, the top panels of Figure 8 Figure 6 shows the anomaly colored by soil type, demonstrating that the wide range is not strongly correlated with independent of soil type."

**RC: Page 22, Line 16 - I would stay km-scale instead of small scale. That term is fairly nebulous and means different things to different communities. For example, small scale in watershed hydrology is cm to meter scale.**

Author Response: We agree that km-scale is more precise and will revise to clarify.

**Revisions:**

Page 35, line 1: revised in response to this and other comments:

"Overall, this section shows that unpredictable and nonlinear kilometre-small scale impacts aggregate to <del>predictable, linear</del> large scale <del>behaviour</del> changes which can be well predicted by linear combinations of single factor simulations."

\_\_\_\_\_

**Comments and Revisions for Peter LaFollette:**

**RC: The reviewer notes that simulations lasted for 3 model years and suggests that we discuss the temporal evolution of nonlinearities.**

**Author Response:** We agree that temporal evolution of drought is an important topic; however, in this paper we focus on results in the last year of model simulations in order to emphasize spatial scaling and factor interactions. Temporal evolution of drought is a large topic in itself and we feel that it is out of the scope of the present study; however, we will add text clarifying that we focus on the last (3rd) year and explaining the rationale.

Revisions: This was addressed in response to reviewer comments above.

**RC: The reviewer notes that the caption of Figure 12 does not state exactly what "nonlinearity" is plotted.**

**Author Response:** We agree this should be more explicitly stated and will revise to clarify that the nonlinearity is a comparison between the multi-factor hot/dry run versus the single factor hot run and dry run.

**Revisions:** This was revised in response to reviewer comments above.

**RC: The reviewer notes that the paper lacks comparisons to measured data, and suggests that we should add such a comparison in the case of antecedent soil moisture. Author Response:**

- We agree that comparison to data is an important and challenging step of model studies. There are observations that could be used to explore the same research questions that are addressed here; however, none of them are directly comparable to the present model because the present model is not a reconstruction of any historical drought.
- The findings of this model, therefore, cannot be taken as a direct prediction for central North America. Rather, their value lies in suggesting system-scale phenomena such as the nonlinear combination of factors.
- It would be possible to design experimental studies or analyses of existing measurements to further explore these phenomena; however, we feel that is a future research topic and not possible within the scope of this study.
- We will add further discussion of this important question to the paper.

**Revisions:**

Page 13, line 9: The following text was added to the methods:

"In analyzing model results, it is important to note that these results do not constitute a direct prediction or reconstruction of a specific historical drought, therefore we do not directly validate the simulated drought impacts against observations. Comparison to data is an important and challenging step of model studies. There are observations that could be used to explore the same research questions that are addressed here; however, none of them are directly comparable to the present model because the present model is not a reconstruction of any historical drought. The findings of this model, therefore, cannot be taken as a direct prediction for central North America. Rather, their value lies in suggesting system-scale phenomena such as the nonlinear combination of factors."

**RC: The reviewer queries whether the nonlinearities could be model artifacts, and suggests further discussion.**

**Author Response:** We acknowledge that all models include simplifications and assumptions that in some cases produce artifacts, results that lack a physical basis. In this case, we have employed an integrated hydrologic model that minimizes these assumptions because it represents all important processes of the hydrologic cycle using physically based equations, as we discuss in the methods. We acknowledge that the precise results of the model do not constitute a direct prediction, but we believe the system-scale results will not be artifacts of the model itself. We will add more discussion of this point.

**Revisions:**

Page 25, line 27: Text was added to the results section:

"A final possible explanation is that the nonlinearity is an artifact of the modeling. This model captures important processes of the hydrologic cycle using physically based equations, as discussed in the methods. Additionally, all runs were conducted with the same modeling environment and compared to a baseline from that model. However, we acknowledge there could be additional nonlinearities and feedbacks in the system that are not captured in our model."

**RC: The reviewer points out some missing citations in the reference list; Revisions:**

Reference list: The reference list was revised in response to this and other comments.

**RC: The reviewer notes that PRISM climatological data and the model inputs and results have different grid resolutions, and queries how this was handled.**

**Author Response**: We agree that PRISM data has a coarser resolution than the model grid. We resampled PRISM rasters to the model grid before preparing forcing data, and will add text to the methods clarifying this point.

**Revisions:**

Page 11, line 9: Text was added to the methods.

"Since PRISM data has a coarser resolution than the model grid, the PRISM rasters were resampled to the model grid before preparing forcing data."

**RC: The reviewer asks for more details about how the drought forcing data was prepared.**

**Author Response:**

- We began with hourly North American Land Data Assimilation System (NLDAS) reconstructions of temperature and precipitation from a baseline water year. For our baseline we chose water year 1984, which is one of the most average water years for the United States in recent decades. We then increased temperature and decreased precipitation using anomalies drawn from a major drought in the region.
- To find the drought anomalies, we used PRISM data for water year 1934, a year of severe drought, and the 1920s, the non-drought, immediately preceding decade. We took months of water year 1934 to represent a "drought January" "drought February" etc. We averaged months of the 1920s to arrive at a baseline for that region at that time, a "non-drought January" "non-drought February" etc.
- We compared the months to create anomalies. For example, we subtracted "nondrought January" temperature from "drought January" temperature to find the January temperature anomaly. The averaging and subtraction was done for each pixel of the model grid, producing a spatial map for each month.

- As the last step, we modified the hourly baseline temperature data by adding the anomaly for the appropriate month in each cell.
- Precipitation data was processed in the same way except that we found a percent change for each month instead of an absolute difference, to avoid negative precipitation values.
- We will add a clarification of this point to the methods.

Revisions: Revised in response to reviewer comments above

**RC: The reviewer asks for more detail regarding crop types and their representation, and suggests we could include a more detailed parameterization of the crops. Author Response:**

- We agree that different crops will affect the details of drought evolution in different ways, but feel it is important to note that ParFlow-CLM is not an agricultural model. Each cell is assigned 1 vegetation type and all crops are represented as the same "croplands" vegetation type. This is an approximation, as is our choice to remove all vegetation in these cells for a simulated crop failure.
- Analyzing the details of crop type and its impact is, we believe, outside the scope of the present study and not critical to address the study questions.
- We will add a brief discussion of this point to the methods.

**Revisions:**

Page 8, line 9: The following passage was added:

"Additionally, different crops may affect the details of drought evolution in different ways, but this is not represented here because ParFlow-CLM is not an agricultural model. Each cell is assigned 1 vegetation type and all crops are represented as the same "croplands" vegetation type. Analyzing the details of crop type and its impact is outside the scope of the present study and not critical to address the study questions."

**RC: The reviewer notes that Figure 12a shows that the median of the boxplots increases slightly in nonlinearity across scales, and queries whether this is a contradiction to our point that "nonlinearity is less at large scales." Author Response:**

**Author Response: We agree that the media**

- We agree that the median nonlinearity becomes more positive for runoff, and more negative for ET as scale increases. Our interpretation focuses on the spread of the boxplots, and on noting that extremely nonlinear responses happen more at small scales.
- As the scale increases, these responses average out to become less extreme, as shown by the decreasing spread of the boxplots.
- In a subcontinental basin as a whole, there is indeed a small positive nonlinearity in runoff: The change in runoff under both temperature and precipitation increase is slightly larger than that due to the separate effects of temperature and precipitation. However, in small basins, the change in runoff due to both variables can be much smaller or larger than expected.
- We will add further discussion to clarify this point.
- Note that Figure 12 is now Figure 14 due to other revisions.

**Revisions:**

Page 30, line 20: Text was added to the discussion:

"Closer inspection of Figure 14 shows that the median nonlinearity becomes more positive for runoff, and more negative for ET as scale increases. In a subcontinental basin as a whole, there is a small positive nonlinearity in runoff, meaning that the change in runoff under both temperature and precipitation increase is slightly larger than that due to the separate effects of temperature and precipitation. The interpretation that responses are more linear at larger scales

**focuses not on the median magnitude, but rather on the decreasing interquartile range of the boxplots at larger scales."**

**RC: The reviewer queries why Water Year 1984 was selected as our baseline;**

Author Response: we will clarify 1984 was an average water year in the United States. Revisions: This was addressed in response to comments, above.

**RC: Page 3, line 32: you should consider including "water table level" in the text of the first question.**

**Revisions:**

Page 4, line 22: The sentence was revised in response to this and other comments: "What is the relative importance of precipitation, temperature and land cover change in response of ET, runoff, and soil moisture and water table levels to meteorological drought?"

**RC: Page 4, line 8: change "19390s" to 1930s**

**Revisions:** Addressed in response to comments above.

**RC: Page 4, line 12: include "it" after "because**

Revisions: Addressed in response to comments above

**RC: Page 4, line 13: include "for" after "suited"**

**Revisions:** Addressed in response to comments above.

**RC: Page 7, line 9: delete "and" between "2" and "include"**

**Revisions**:**

Page 9, line 9: revised in response to this and other comments: "The experiments are outlined in Table 1 and Table 2 and include a baseline run, ..."

**RC: Page 10, line 5: change "focus:" to "focuses"**

**Revisions:** The sentence was revised in response to this and other comments: Page 14, line 6: The secondfirst section focuses on attribution of drought impacts to specific factors of temperature, precipitation and land cover.

**RC: Page 11, line 8: change "decrease" to "decreased" Revisions:**

Page 18, line 3: revised in response to this and other comments:

"Relative to the baseline scenario mostall of the drought scenarios have decreased runoff and ET, depleted soil water content and lower water tables (ie, water tables that are further below the land surface), as would be expected in a drought."

**RC: Page 15, figure 7: Apparently, the anomalies in water table are not summed for the "sum of hot and dry effects"**

**Author Comment:** The value 2 for water table anomalies in the hot run was inadvertently entered as 20 when creating the figure. We appreciate being alerted to the oversight. Note that Figure 7 is now Figure 9.

**Revisions:**

Figure 9: The figure was replaced by an updated version containing the corrected value.

**RC: Page 20, line 8: change "becomes" to "become" Revisions:**

Page 30, line 9: The sentence was deleted in response to other comments:

"Combinations of factors becomes more linear at larger scales."

**RC: Page 23, line 5: delete a period at the end of the sentence Revisions:**

Figure 15: "Panel (b) shows the typical configuration for comparison, where ET increases in areas of land cover disturbance.."

**RC: Page 24, line 32: delete the second comma.**

**Revisions:**

Page 38, line 3: "This extends the work of Maxwell and Condon (2016), which showed that lateral flow,-affects ET thresholds within the system."

**Unravelling the impacts Evaluating the relative importance of precipitation, temperature and land-cover change in the hydrologic response to for extreme meteorological drought conditions over the **North American High Plains**

Annette Hein1, Laura Condon2, Reed Maxwell1 5

1Department of Geology and Geological Engineering, Colorado School of Mines, Golden, 80401, USA 2Department of Hydrology and Atmospheric Science, University of Arizona, Tucson, 85721, USA

Correspondence to: Annette Hein (ahein@alumni.mines.edu)

**Abstract.** Drought is a natural disaster that may become more common in the future under climate change.

- 10 It involves changes to temperature, precipitation, and/or land cover, but the relative contributions of each of these factors to overall drought severity is not clear. Here we apply a high-resolution integrated hydrologic model of the High Plains to explore the individual importance of each of these factors and the feedbacks between them. The model was constructed using ParFlow-CLM, which represents surface and subsurface processes in detail with physically based equations. Numerical experiments were run to perturb vegetation,
- 15 precipitation and temperature separately and in combination. Results show that decreased precipitation caused larger anomalies in evapotranspiration, soil moisture, stream flow and water table levels than did increased temperature or disturbed land cover. However, these factors are not linearly additive when applied in combination; some effects of multi-factor runs came from interactions between temperature, precipitation and land cover. Spatial scale was important in characterizing impacts, as unpredictable and nonlinear impacts at small scales aggregate to predictable, linear large scale behaviour.
- 20

**1** Introduction

Improved understanding of drought is important to sustainably manage water resources and agricultural production worldwide. Agriculture depends on rainfall, especially in arid and semiarid regions, so large droughts can devastate global agriculture.

25

Because there are many ways to characterize drought, researchers often make a distinction between meteorological and hydrologic drought (Van Loon, 2015). Meteorological drought is defined as weather changes such as decreased precipitation or increased temperature. These changes may produce hydrologic drought, which is defined as impacts to the hydrologic system such as decreased runoff or soil moisture.

As climate continues to change, meteorological droughts are projected to occur may happen more often and with greater severity than we have seen in the past. These meteorological changes will propagate 30

to increased hydrologic drought as watersheds become increasingly stressed be worse than they are now (IPCC 2014; Diffenbaugh et al. 2017).

Within the United States, the High Plains is a key agricultural region that is also drought-prone. Drought affected that region on many occasions during the 20th century, including the Dustbowl of the 5 1930s (Hong and Kalnay 2000, Schubert et al. 2004) and the more recent 2012 drought that dried soils and lowered crop yields across most of the area (Otkin et al., 2012). Forecasts for the High Plains predict similar or worse droughts in the future (Cook et al., 2015) that could result in significant declines in crop yields (Glotter and Elliot, 2016). In the past, groundwater pumping has been used to buffer the region against hydrologic drought impacts, but groundwaterit is becoming depleted (Scanlon et al 2012, McGuire

10 2017). A better understanding of the effects of meteorological drought and the resulting hydrologic drought gained from modeling studies will be valuable for meeting future sustainability challenges.

Meteorological droughts are often characterized by some combination of Two important factors that cause drought are decreased precipitation and increased temperature (Van Loon 2015), with vegetation acting as a subsidiary factor. Hydrologic Ddrought occurs when these changes propagate through the watershed resulting in streamflow losses, changes in impact evapotranspiration (ET) and decreased soil moisture levels within a watershed. Sustained hydrologic drought can ultimately lead to crop failure for managed systems or changes in vegetation for natural systems.

There are a variety of pathways by which a meteorological drought can evolve to a hydrologic drought. When precipitation decreases, less water is available for any part of the water cycle including ET. If the system is already wet (energy limited), this change may cause only minor impacts if the remaining water is still sufficient to supply potential ET. If the system is water limited, then a meteorological drought with a decrease in precipitation will cause a hydrologic drought in which ET to decrease and soil moisture to dry both decrease. Some of the energy previously used to evaporate the water (latent heat of phase change) will instead go to sensible heat, causing a shift in the Bowen ratio (Eltahir 1998,Seneviratne 2010). Although feedbacks between the land surface and atmosphere are outside the scope of the current study, it should be noted that Tthis change in the surface energy balance can carry over into atmospheric instability and changes to circulation (Eltahir 1998) creating feedbacks to meteorology (Brubaker and Entekhabi 1996) at a variety of timescales (Betts et al 1996). In the present study area of the High Plains, an ensemble of climate models found a strong connection between soil moisture and the atmosphere (Koster 2004).

In contrast to the precipitation decrease, a temperature increases causes hydrologic drought more indirectly, through an increase in potential ET. In an energy limited system, the available water will supply a higher actual ET (McEvoy 2016). In a water limited system, the increase in ET is bounded by the

available water. Initially, ET can still increase, but as the soils dry ET is eventually expected to drop due to water limitations. This initial increase of ET is the opposite direction of the effect predicted for precipitation decrease, so in a drought where both occur, there will be a competing effect on the hydrologic system (Livneh and Hoerling 2016). If vegetation is disturbed, its buffering effect on the system is

5 removed. Vegetation is expected to have a buffering effect against impacts to ET because it can reach deeper sources of water to satisfy ET demands when the surface soil moisture is depleted (Maxwell and Condon 2016).

Many studies have used models to explore the driving factors and possible effects of future droughts. Otkin et al. (2012) examined US Department of Agriculture metrics and Noah, Mosaic and

- 10 Variable Infiltration Capacity (VIC) models to show that hot and dry conditions in the 2012 drought dried High Plains soils within a few months. Gosling et al. (2017) used an ensemble of local and global hydrologic models and a variety of climate change scenarios to conclude there was no definite prediction for runoff in the upper Mississippi basin. Crosbie et al. (2012) also found no definite prediction for recharge in the High Plains under scenarios from 16 global climate models. Chien et al. (2012) predicted
- 15 with a Soil Water Assessment Tool (SWAT) model that streamflow in Illinois watersheds will decrease under climate change. Naz et al. (2016) modeled hydrologic response to climate change across the entire continental US with a VIC model. They found large regional differences in runoff, SWE, and rain-to-snow ratio across the country under various Climate Model Intercomparison Project 5 model projections. Meixner et al. (2015) reviewed studies across 8 representative aquifers in the United States to anticipate
- 20 effects of climate change on recharge. Recharge increased slightly in the northern High Plains, and decreased in the south.

Modelling studies typically include some combination of temperature, precipitation and land cover changes as forcing factors to drought. However, since the preceding studies are either reconstructing a natural event or forecasting future droughts, they involve many simultaneous changes in forcing variables. 25 Although the broad theoretical importance of each variable is clear, multiple simultaneous changes in one study obscure the details of exact mechanisms or interactions between factors. To address this limitation, other studies have taken the approach of isolating and comparing factors using numerical experiments instead of reconstructing real-world events.

Livneh and Hoerling (2016) argued that precipitation was more important than temperature in 30 causing hydrologic drought impacts in the High Plains based on results from historical reconstruction and sensitivity experiments using VIC and the Unified Land Model (ULM). Maxwell and Kollet (2008) ran a ParFlow-CLM model over the Little Washita basin in Oklahoma and found that a 2.5 degree C temperature increase reduced saturation and potential recharge. If precipitation decreased, eEffects were much more extreme with precipitation decreases than if temperature increased alone. They showed that this relationship was caused by shallow groundwater supported by lateral convergence in the subsurface, which Groundwater tables near the land surface allowed local regions of the model to maintain saturation and potential recharge regardless of the climate perturbations. These studies summarized here suggest that

5 precipitation changes, typical of observed droughts, outweigh the effects of typical temperature or land cover changes in water limited systems. However, if precipitation is stable, these secondary factors can be important; and in any casewhen considered together with precipitation changes they may mitigate or exacerbate the effect of precipitation.

Previous work has Existing studies have identified precipitation and temperature as the most important controls of watershed drought response, important factors relating to drought, with vegetation changes as a secondary controlimpact. The studies reviewed here often reconstruct historical events, which does not allow for isolation of individual factors and their effects. A few studies have focused on individual factors with a numerical experiment approach; this study does the same, but uses aHere we focus on isolating individual drought factors using an more-advanced and flexible-modeling code hydrologic model

- 15 to ensure that the results are as physically based as possible. This study builds on previous studies that compare different meteorological factors and their impact on hydrology by quantifying those impacts in detail. In this study we apply an integrated hydrologic model, ParFlow-CLM, to run numerical experiments across the southern High Plains. ParFlow is ideal for this study because it is a fully integrated hydrologic model that includes groundwater, soil moisture and overland flow processes. This allows a more detailed
- 20 exploration than other tools that have been used in similar studies that rely on more simplified or compartmentalized approaches. In particular, the project addresses three specific questions:

1) What is the relative importance of precipitation, temperature and land cover change in response of ET, runoff,and\_soil moistureand water table levels tometeorological drought?

2) How do the hydrologic impacts of precipitation, temperature and land cover change differ when drivingfactors are considered together rather than in isolation?

3) How do impacts of the main drought factors and their interactions change across spatial scales?

**2 Methods**

This study explores how temperature, precipitation and land cover affect the water and energy balance of the High Plains through a series of numerical experiments where the driving factors (precipitation and temperature) and land cover are systematically perturbed. While land cover change can be viewed as a response to drought, it can also exacerbate system response to further drought. We include land cover change in our perturbations here to incorporate systemic watershed changes in addition to the

meteorological forcing difference. The scenarios developed for these numerical experiments were modeled after an example of extreme drought in the region, the Dustbowl of the 19390s1930s. The goal of the study is not to reconstruct the Dustbowl or produce operational forecasting, rather to exploit the capabilities of large-scale modeling to illuminate major features of the hydrologic system using a real world extreme drought as a test case

5 drought as a test case.

The numerical simulations were conducted with ParFlow-CLM, an integrated hydrologic model. ParFlow-CLM was selected because it employs a more extensive and physically based representation of subsurface processes than many other hydrologic models and is therefore well suited to simulate the water and energy dynamics that occur during drought. Here we provide more details on the modeling platform

10 (2.1), study domain (2.2), drought scenarios (2.3) and metrics of analysis (2.4). Selected model inputs and outputs are presented in the model data (Hein et al 2018) on the Harvard Dataverse.

**2.1 Model Selection**

The model was constructed using ParFlow, an integrated hydrologic modeling code, coupled to the Common Land Model (CLM), a land surface modeling code. ParFlow is an integrated hydrologic model

- 15 with an explicit description of most physical processes of the water balance. The terminology "integrated hydrologic model" used here refers to the globally implicit solution of Richards' Equation and the shallow water equations, thus integrating variably saturated subsurface and overland flow processes. It is not intended to imply that the model includes anthropogenic or biologic processes. ParFlow simulates It models saturated and unsaturated flow in three dimensions using Richards' Equation, with flux defined by Darcy's
- 20 Law-and relationships between pressure and relative saturation or permeability defined by the Van Genuchten pressure-saturation and pressure-relative permeability relationships (Van Genuchten, 1980)functions. Overland flow is modeled with the kinematic wave equation, with velocity found by Manning's equation. Energy and water balances at the surface are represented with CLM (Dai et al., 2003; Maxwell and Miller 2005, Jefferson et al. 2017). CLM is was coupled to ParFlow by passing the land surface water flux to ParFlow as a forcing in the top layer, and substituting ParFlow's computations for
- 25 surface water flux to ParFlow as a forcing in the top layer, and substituting ParFlow's computations for infiltration and streamflow routing within CLM (Maxwell and Miller 2005). ParFlow has a number of differences with commonly used models in other drought-related studies. It is instructive to compare ParFlow with, the Variable Infiltration Capacity (VIC) model (VIC 2016; Liang et al. 1994) and the Soil Water Assessment Tool (SWAT, ) (Neisch et al. 2011), not in order to criticize any
- 30 model, but to illustrate the reasoning for model choice in this study. ParFlow allows any number of subsurface layers with any specified conductivity, and has vertical and lateral flow driven by pressure gradients. Soil moisture and groundwater are not distinguished; both are represented through pressure in a cell and solved for using Richards' Equation. This 3 dimensional variably saturated flow is the main

difference between ParFlow and other models. While this detailed representation is an advantage for this study, it also leads to higher computational expense in ParFlow runs when compared to other models. VIC typically has three soil layers and does not simulate lateral flow between macroscale grid cells, although it includes a baseflow term for water leaving a cell to enter a stream. VIC is often applied to larger scale

- 5 modeling, while ParFlow can be used at any resolution. SWAT partitions groundwater into a "deep aquifer" which can have lateral flow to other subbasins and a "shallow aquifer" which contributes only to the stream. Soil moisture and groundwater are modeled separately. In contrast, ParFlow combines all of these processes at a variety of scales; in this model, scales are included from km-scale to subcontinental. (This does omit meter and cm-scale processes such as biogeochemical cycles.) The detailed, small scale
- 10 representation of subsurface processes makes ParFlow-CLM a suitable model to run numerical experiments whose results depend on physical processes and their interactions, as opposed to statistical fitting or simplified parameterizations. However, this comes at a tradeoff with higher computational expense due to these detailed representations.

**2.2 Model Configuration**

15 The model domain covers the southern High Plains and Rocky Mountains, including portions of the Arkansas and Missouri river basins (Figure 1) at a 1-km resolution. The domain is 1200 by 1124 km and extends to a depth of 102 km, with 5 layers for a total of 6,744,000 computational cells. The lowest layer is 100 m thick and the other 4 layers are 1 m, 0.6 m, 0.3 m and 0.1m thick, listed from base to top. An overland-flow boundary condition, allowing free development of a stream network, was imposed at the top layer. A no-flow boundary condition was specified at the bottom and on all sides, with the exception of while surface streams which can exit the domain were routed out of the domain at the edge. Due to computational expense, the runs in the present study were performed on the supercomputer Cori at the National Energy Research Scientific Computing Center (NERSC). One year of the model required over 20,000 processor hours to calculate and completed in about a week of wall clock time.

Figure 1: The model domain (box) covers the southern High Plains of the United States. Blue shading indicates the Missouri and Arkansas continental river basins.

- 5 Inputs for the study were developed from previous work of Maxwell and Condon (2016) modeling hydrology across the continental US (CONUS). The basic input data and initial conditions follow Maxwell and Condon (2016). Inputs include slopes, soil types, vegetation, attributes of soils and geologic units, and initial pressure conditions (see Figure 2). The slopes in the x (east-west) and y (north-south) directions were derived from a digital elevation model and sink-filledprocessed to ensure the entire domain was connected.
- 10 The simulations presented here use the kinematic wave approximation of the overland flow equations and therefore require a domain with a connected drainage network, as slopes control surface flow routing. While natural depressions in the landscape exist, literature suggests that it is difficult to distinguish these natural depressions and sinks from noise within the processed DEM (eg Kenny et al 2008). The slopes control surface flow routing. Soil types were taken from the SSURGO database. Important soil attributes
- 15 include porosity, permeability, specific storage and van Genuchten parameters, which control saturated and unsaturated flow. Initial subsurface pressure conditions were taken from the original CONUS run, to increase spinup efficiency. The vegetation dataset was taken from the USGS land cover trends dataset (Soulard et al. 2014). Important vegetative parameters include leaf and stem area index, roughness length and displacement heights, rooting distribution parameters, and reflectance and transmittance for leaves and
- 20 stems (Maxwell and Condon 2016). While most inputs were drawn from Condon and Maxwell 2016, the geologic layer of that study contained features that were geologically less realistic at the scale of the High Plains. The geology of the base layer was updated for this project using local data from the US Geological Survey (USGS, 1998; USGS, 2005).

It is important to note that water management was not included in the simulation. This includessuch as groundwater pumping, surface water storage and diversion, and irrigationare not included in these simulations. This means that results of the project represent the system in a pre-development state not including anthropogenic impacts to the hydrology (Maxwell and Condon 2016). The only management

- 5 impact represented was land use-and its changes applied by setting and changing the land cover type in the modelthrough the vegetative layer. The use of modern day vegetation is not temporally consistent with pre-industrial water management, but unfortunately, there is limited pre-industrial data on land use. In the case of this project, we are not reconstructing any specific historical drought, so it is less important in addressing the research questions to match all forcings and settings to one period of time. Additionally, different crops
- 10 may affect the details of drought evolution in different ways, but this is not represented here because ParFlow-CLM is not an agricultural model. Each cell is assigned 1 vegetation type and all crops are represented as the same "croplands" vegetation type. Analyzing the details of crop type and its impact is outside the scope of the present study and not critical to address the study questions.
- The initial conditions for the simulation were obtained from the existing continental scale 15 simulations (Maxwell and Condon 2016) and include 4 additional years of spinup prior to this project. The pressure file was subset to the High Plains domain and the geologic layer was updated as described in the Appendix. The model was run recursively repeatedly with water year 1984 North American Land Data Assimilation System (NLDAS-2) forcing until average subsurface storage change in one year was less than 1% of precipitation (achieved after three recursive repeated runs). Ajami et al. (2014) showed that change 20 in subsurface storage is one of the most rigorous spinup metrics for integrated hydrologic modelling. Holding this value below 1% of precipitation means that effects seen in numerical experiments can be interpreted as meaningful, i.e. something besides spinup noise if they exceed the threshold.

8

---

## Author Response (AR2)

Dear Dr Wanders,

On behalf of all coauthors, I submit our manuscript "Evaluating the relative importance of precipitation, temperature and land-cover change in the hydrologic response to extreme meteorological drought conditions over the North American High Plains" with minor revisions for your consideration in HESS.

On the following pages, please find our point-by-point response to reviewers and list of all changes, followed by the marked-up manuscript. A clean version of the revised manuscript has been uploaded separately.

We appreciate your time and feedback on the manuscript.

Sincerely,

Annette Hein

Hein et al, list of revisions for second review:

The authors thank the reviewers and editor for their constructive feedback on the manuscript and present the revisions of the manuscript in response to the second round of comments. Note that line numbers in the reviewer comments refer to the revised submission, whereas line numbers in the present response refer to the accompanying second revision with tracked changes. In presenting our revisions, unchanged text is shown in normal format, deleted text is presented with a strikethrough, and added text is underlined.

The list of responses and revisions is presented first, followed by the manuscript showing markup.

Editor Comments:

**Comments to the Author:**

**The authors have made significant modifications to the manuscript thereby resolving most of the issue pointed out by the reviewers. After my own reading of the manuscript I would suggest the authors include the final remarks provided by both reviewers, namely to mention the short-comings of the meteorological resampling and the definition of drought compared to existing hydrological drought literature.**

**Regards,**

**Niko**

Author Response: We have incorporated the final comments of the reviewers.

Revisions: Revisions are detailed below in response to the reviewers and include an adjustment of the land cover discussion and more detail on the definition of hydrologic drought. We appreciate the reviewer's feedback on meteorological forcing, and believe that we have already addressed these points in the passage starting page 11, line 10 to page 12, line 2.

Reviewer 1:

**I thank the authors for their detailed response and the modifications of the manuscript. The authors have addressed all my comments, and some changes were made to the manuscript. These changes improved the presentation and clarity of the methods and results and increased comprehensibility. The distinction between meteorological and hydrological drought is useful and the title is much more representative of the content.**

**However, unfortunately, I remain disappointed by the changes made. This is mainly because the manuscript still deals with 'drought severity' (e.g. abstract, referring to the hydrological system, and throughout the text) without actually addressing or evaluating a hydrological drought.**

**The analysis that the authors provide is valid, so is the application of meteorological drought factors to a hydrological system in the state of an average water year. However, it has to be clear from the manuscript that the authors do not evaluate the hydrological drought response. They evaluate the differences to a baseline scenario, but 'decreased runoff or soil moisture' are not necessarily representative of a drought. A negative difference relative to a baseline (what the authors call 'anomaly') does not necessarily represent a drought and especially not during one 'average water**

year'. I.e. the difference of ~60 mm in soil moisture relative to 330mm of soil moisture in Figure 4 does not show the 'hydrological drought impact' but simply the difference arising from meteorological drought. To evaluate the severity of the drought, these values should be anomalies relative to a climatology, exceeding critical thresholds of e.g. soil moisture. To evaluate a drought severity, the authors should apply an integrated measure of severity (any of the indices out there would do), that addresses the state. In my previous comment, I was also noting the 'extent of drought', which referred to the spatial extent of drought, e.g. as depicted by such a grid cell based index. Similarly, the authors could evaluate the effects on temporal characteristics of drought, such as the onset.

To be clear: I am not asking that the authors re-run their experiments to start from hydrological drought conditions. I am asking them to clarify the terminology of drought in the manuscript (as they do not evaluate the hydrological drought) or add some analysis, as indicated above, that addresses the simulation of hydrological drought. After these changes, the manuscript should be publishable.

Author Response:

We thank the reviewer for the detailed feedback that has improved the clarity and quality of the manuscript. We have added some context about drought metrics to our discussion of hydrological drought.

Revisions:

Page 12, Line 12: The following text was added: "We acknowledge that these calculated anomalies are not equivalent to published drought metrics that are often used to define hydrologic drought. These calculations were chosen for this analysis in order to specifically evaluate the research questions being addressed, through specific focus on hydrological variables such as soil moisture, water table level, runoff and ET. "

Reviewer 2:

The paper now titled "Evaluating the relative importance of precipitation, temperature, and land-cover change in the hydrologic response to extreme meteorological drought conditions over the North American High Plains" uses ParFlow-CLM to understand how perturbations of the meteorology of the High Plains over the contiguous United States impact the hydrologic cycle. The authors have provided satisfactory responses to the majority of my comments. Although I still disagree that it is ok to simply perturb one variable at a time; I agree that this approach is common and therefore should not prevent this work from being published. In the end, however, these types of assumptions (similar to the limitation on the land cover database use) will limit how much these results can be generalized and therefore the larger impact of this study; but again that should not limit this work from being published at this time. I have one minor comment below that can be addressed at the proof stage and a more general response regarding the meteorology challenge that doesn't necessarily need to be addressed.

Page 7; Line 16: The climate modeling community have actually put a lot of effort into developing historical land use/land cover datasets (https://link.springer.com/article/10.1007/s10584-011-0153-2) so this statement is not truly accurate. You could have chosen to use these data instead. This should at least be mentioned in the paper.

**Table 1: You are correct in stating that others also use the approach of just perturbing one variable at a time. But again, this can be completely inconsistent. All the meteorology is inherently correlated; as such, varying one variable at a time does not represent a realistic scenario and can lead to misleading conclusions. In the end, I am ok with moving ahead with the simulations as is. However, I think it is a missed opportunity. Just because it has been done before does not mean there isn't a better way. For the case of the temperature, it would have been pretty straightforward to modify both specific humidity (assume constant relative humidity) and longwave radiation (assume constant emissivity). In the end, whether it will change things dramatically is certainly debatable, but I don't agree with the statement that changing "three at a time" would be problematic. It would still be only change temperature and then adjust variables that are strongly dependent on air temperature.**

Author Response:

We thank the reviewer for the detailed feedback that has improved the clarity and quality of the manuscript.

We appreciate the reviewer's feedback on meteorological forcing, and believe that we have already addressed these points in the passage starting page 11, line 10 to page 12, line 2.

We have added the suggested reference and revised the sentence on land cover datasets.

Revisions:

Page 7, line 18: The use of modern day vegetation is not temporally consistent with pre-industrial water management (Hurtt,2011), . However, in  this project, we are not reconstructing any specific historical drought, so it is less important in addressing the research questions to match all forcings and settings to one period of time.

Reference list, page 34, line 11:

Hurtt, G.C., Chini, L.P., Frolking, S., Betts, R. A., Feddema, J., Fischer, G., Hibbard, K., Houghton, R.A., Janetos, A., Jones, C.D., Kindermann, G., Kinoshita, T., Goldewijk, K.K., Riahi,K., Shevliakova, E., Smith, S., Stehfest, E., Thomson, A., Thornton, P., van Vuuren, D.P., Wang, Y.P.: Harmonization of land use scenarios for the period 1500-2100: 600 years of global gridded annual land-use transitions, wood harvest, and resulting secondary lands. Climatic Change (109:117), 2011.

Other Revisions:

Affiliation was updated for the first author; acknowledgments were updated to include editor and all reviewers.

**Evaluating the relative importance of precipitation, temperature and land-cover change in the hydrologic response to extreme meteorological drought conditions over the North American High Plains**

5    Annette Hein[1*], Laura Condon[2], Reed Maxwell[1]

[1]Department of Geology and Geological Engineering, Colorado School of Mines, Golden, 80401, USA
*Now at SS Papadopulos and Associates, Boulder, 80303, USA
[2]Department of Hydrology and Atmospheric Science, University of Arizona, Tucson, 85721, USA

*Correspondence to*: Annette Hein (ahein@alumni.mines.edu; ahein@sspa.com)

10    **Abstract.** Drought is a natural disaster that may become more common in the future under climate change. It involves changes to temperature, precipitation, and/or land cover, but the relative contributions of each of these factors to overall drought severity is not clear. Here we apply a high-resolution integrated hydrologic model of the High Plains to explore the individual importance of each of these factors and the feedbacks between them. The model was constructed using ParFlow-CLM, which represents surface and subsurface

15    processes in detail with physically based equations. Numerical experiments were run to perturb vegetation, precipitation and temperature separately and in combination. Results show that decreased precipitation caused larger anomalies in evapotranspiration, soil moisture, stream flow and water table levels than did increased temperature or disturbed land cover. However, these factors are not linearly additive when applied in combination; some effects of multi-factor runs came from interactions between temperature,

20    precipitation and land cover. Spatial scale was important in characterizing impacts, as unpredictable and nonlinear impacts at small scales aggregate to predictable, linear large scale behaviour.

**1 Introduction**

Improved understanding of drought is important to sustainably manage water resources and agricultural production worldwide. Agriculture depends on rainfall, especially in arid and semiarid regions,

25    so large droughts can devastate global agriculture.

Because there are many ways to characterize drought, researchers often make a distinction between meteorological and hydrologic drought (Van Loon, 2015). Meteorological drought is defined as weather changes such as decreased precipitation or increased temperature. These changes may produce hydrologic drought, which is defined as impacts to the hydrologic system such as decreased runoff or soil moisture.

As climate continues to change, meteorological droughts are projected to occur more often and with greater severity than we have seen in the past. These meteorological changes will propagate to increased hydrologic drought as watersheds become increasingly stressed(IPCC 2014; Diffenbaugh et al. 2017).

Within the United States, the High Plains is a key agricultural region that is also drought-prone. Drought affected that region on many occasions during the 20th century, including the Dustbowl of the 1930s (Hong and Kalnay 2000, Schubert et al. 2004) and the more recent 2012 drought that dried soils and lowered crop yields across most of the area (Otkin et al., 2012). Forecasts for the High Plains predict similar or worse droughts in the future (Cook et al., 2015) that could result in significant declines in crop yields (Glotter and Elliot, 2016). In the past, groundwater pumping has been used to buffer the region against hydrologic drought impacts, but groundwater is becoming depleted (Scanlon et al 2012, McGuire 2017). A better understanding of the effects of meteorological drought and the resulting hydrologic drought gained from modeling studies will be valuable for meeting future sustainability challenges.

Meteorological droughts are often characterized by some combination of decreased precipitation and increased temperature (Van Loon 2015). Hydrologic drought occurs when these changes propagate through the watershed resulting in streamflow losses, changes in evapotranspiration (ET) and decreased soil moisture levels within a watershed. Sustained hydrologic drought can ultimately lead to crop failure for managed systems or changes in vegetation for natural systems.

There are a variety of pathways by which a meteorological drought can evolve to a hydrologic drought. When precipitation decreases, less water is available for any part of the water cycle including ET. If the system is already wet (energy limited), this change may cause only minor impacts if the remaining water is still sufficient to supply potential ET. If the system is water limited, then a meteorological drought with a decrease in precipitation will cause a hydrologic drought in which ET and soil moisture both decrease. Some of the energy previously used to evaporate the water (latent heat of phase change) will instead go to sensible heat, causing a shift in the Bowen ratio (Eltahir 1998,Seneviratne 2010). Although feedbacks between the land surface and atmosphere are outside the scope of the current study, it should be noted that this change in the surface energy balance can carry over into atmospheric instability and changes to circulation (Eltahir 1998) creating feedbacks to meteorology (Brubaker and Entekhabi 1996) at a variety of timescales (Betts et al1996). In the present study area of the High Plains, an ensemble of climate models found a strong connection between soil moisture and the atmosphere (Koster 2004).

In contrast to the precipitation decrease, temperature increases cause hydrologic drought more indirectly, through an increase in potential ET. In an energy limited system, the available water will supply a higher actual ET (McEvoy 2016). In a water limited system, the increase in ET is bounded by the available water. Initially, ET can still increase, but as the soils dry ET is eventually expected to drop due to

water limitations. This initial increase of ET is the opposite direction of the effect predicted for precipitation decrease, so in a drought where both occur, there will be a competing effect on the hydrologic system (Livneh and Hoerling 2016). If vegetation is disturbed, its buffering effect on the system is removed. Vegetation is expected to have a buffering effect against impacts to ET because it can reach deeper sources of water to satisfy ET demands when the surface soil moisture is depleted (Maxwell and Condon 2016).

Many studies have used models to explore the driving factors and possible effects of future droughts. Otkin et al. (2012) examined US Department of Agriculture metrics and Noah, Mosaic and Variable Infiltration Capacity (VIC) models to show that hot and dry conditions in the 2012 drought dried High Plains soils within a few months. Gosling et al. (2017) used an ensemble of local and global hydrologic models and a variety of climate change scenarios to conclude there was no definite prediction for runoff in the upper Mississippi basin. Crosbie et al. (2012) also found no definite prediction for recharge in the High Plains under scenarios from 16 global climate models. Chien et al. (2012) predicted with a Soil Water Assessment Tool (SWAT) model that streamflow in Illinois watersheds will decrease under climate change. Naz et al. (2016) modeled hydrologic response to climate change across the entire continental US with a VIC model. They found large regional differences in runoff, SWE, and rain-to-snow ratio across the country under various Climate Model Intercomparison Project 5 model projections. Meixner et al. (2015) reviewed studies across 8 representative aquifers in the United States to anticipate effects of climate change on recharge. Recharge increased slightly in the northern High Plains, and decreased in the south.

Modelling studies typically include some combination of temperature, precipitation and land cover changes as forcing factors to drought. However, since the preceding studies are either reconstructing a natural event or forecasting future droughts, they involve many simultaneous changes in forcing variables. Although the broad theoretical importance of each variable is clear, multiple simultaneous changes in one study obscure the details of exact mechanisms or interactions between factors. To address this limitation, other studies have taken the approach of isolating and comparing factors using numerical experiments instead of reconstructing real-world events.

Livneh and Hoerling (2016) argued that precipitation was more important than temperature in causing hydrologic drought impacts in the High Plains based on results from historical reconstruction and sensitivity experiments using VIC and the Unified Land Model (ULM). Maxwell and Kollet (2008) ran a ParFlow-CLM model over the Little Washita basin in Oklahoma and found that a 2.5 degree C temperature increase reduced saturation and potential recharge. Effects were much more extreme with precipitation decreases than if temperature increased alone. They showed that this relationship was caused by shallow

groundwater supported by lateral convergence in the subsurface, which allowed local regions of the model to maintain saturation and potential recharge regardless of the climate perturbations. These studies suggest that precipitation changes, typical of observed droughts, outweigh the effects of typical temperature or land cover changes in water limited systems. However, if precipitation is stable, these secondary factors can be important; and when considered together with precipitation changes they may mitigate or exacerbate the effect of precipitation.

Previous work has identified precipitation and temperature as the most important controls of watershed drought response, with vegetation changes as a secondary impact. The studies reviewed here often reconstruct historical events, which does not allow for isolation of individual factors and their effects. Here we focus on isolating individual drought factors using an advanced and flexible hydrologic model to ensure that the results are as physically based as possible. This study builds on previous studies that compare different meteorological factors and their impact on hydrology by quantifying those impacts in detail. In particular, the project addresses three specific questions:

1) What is the relative importance of precipitation, temperature and land cover change in response of ET, runoff, soil moisture and water table levels to meteorological drought?

2) How do the hydrologic impacts of precipitation, temperature and land cover change differ when driving factors are considered together rather than in isolation?

3) How do impacts of the main drought factors and their interactions change across spatial scales?

**2 Methods**

This study explores how temperature, precipitation and land cover affect the water and energy balance of the High Plains through a series of numerical experiments where the driving factors (precipitation and temperature) and land cover are systematically perturbed.  While land cover change can be viewed as a response to drought, it can also exacerbate system response to further drought. We include land cover change in our perturbations here to incorporate systemic watershed changes in addition to the meteorological forcing difference. The scenarios developed for these numerical experiments were modeled after an example of extreme drought in the region, the Dustbowl of the 1930s.  The goal of the study is not to reconstruct the Dustbowl or produce operational forecasting, rather to exploit the capabilities of large-scale modeling to illuminate major features of the hydrologic system using a real world extreme drought as a test case.

The numerical simulations were conducted with ParFlow-CLM, an integrated hydrologic model. ParFlow-CLM was selected because it employs a more extensive and physically based representation of subsurface processes than many other hydrologic models and is therefore well suited to simulate the water

and energy dynamics that occur during drought. Here we provide more details on the modeling platform (2.1), study domain (2.2), drought scenarios (2.3) and metrics of analysis (2.4). Selected model inputs and outputs are presented in the model data (Hein et al 2018) on the Harvard Dataverse.

**2.1 Model Selection**

5   The model was constructed using ParFlow, an integrated hydrologic modeling code, coupled to the Common Land Model (CLM), a land surface modeling code. The terminology "integrated hydrologic model" used here refers to the integration of variably saturated subsurface and overland flow processes and is not intended to imply that the model includes anthropogenic or biologic processes. ParFlow simulates saturated and unsaturated flow in three dimensions using Richards' Equation, and relationships between

10   pressure and relative saturation or permeability defined by the Van Genuchten pressure-saturation and pressure-relative permeability relationships (Van Genuchten, 1980). Overland flow is modeled with the kinematic wave equation, with velocity found by Manning's equation. Energy and water balances at the surface are represented with CLM (Dai et al., 2003; Maxwell and Miller 2005, Jefferson et al. 2017). CLM is coupled to ParFlow by passing the land surface water flux to ParFlow as a forcing in the top layer, and

15   substituting ParFlow's computations for infiltration and streamflow routing within CLM (Maxwell and Miller 2005).

ParFlow has a number of differences with commonly used models in other drought-related studies. It is instructive to compare ParFlow with the Variable Infiltration Capacity (VIC) model (VIC 2016; Liang et al. 1994) and the Soil Water Assessment Tool (SWAT, Neisch et al. 2011), not in order to criticize any model,

20   but to illustrate the reasoning for model choice in this study. ParFlow allows any number of subsurface layers with any specified conductivity, and has vertical and lateral flow driven by pressure gradients. Soil moisture and groundwater are not distinguished; both are represented through pressure in a cell and solved for using Richards' Equation. This 3 dimensional variably saturated flow is the main difference between ParFlow and other models. While this detailed representation is an advantage for this study, it also leads to

25   higher computational expense in ParFlow runs when compared to other models. VIC typically has three soil layers and does not simulate lateral flow between macroscale grid cells, although it includes a baseflow term for water leaving a cell to enter a stream. VIC is often applied to larger scale modeling, while ParFlow can be used at any resolution. SWAT partitions groundwater into a "deep aquifer" which can have lateral flow to other subbasins and a "shallow aquifer" which contributes only to the stream. Soil moisture

30   and groundwater are modeled separately. In contrast, ParFlow combines all of these processes at a variety of scales; in this model, scales are included from km-scale to subcontinental. (This does omit meter and cm-scale processes such as biogeochemical cycles.) The detailed representation of subsurface processes

makes ParFlow-CLM a suitable model to run numerical experiments whose results depend on physical processes and their interactions, as opposed to statistical fitting or simplified parameterizations.

**2.2 Model Configuration**

The model domain covers the southern High Plains and Rocky Mountains, including portions of the Arkansas and Missouri river basins (Figure 1) at a 1-km resolution. The domain is 1200 by 1124 km and extends to a depth of 102 m, with 5 layers for a total of 6,744,000 computational cells. The lowest layer is 100 m thick and the other 4 layers are 1 m, 0.6 m, 0.3 m and 0.1m thick, listed from base to top. An overland-flow boundary condition, allowing free development of a stream network, was imposed at the top layer. A no-flow boundary condition was specified at the bottom and on all sides, with the exception of surface streams which can exit the domain.  Due to computational expense, the runs in the present study were performed on the supercomputer Cori at the National Energy Research Scientific Computing Center (NERSC). One year of the model required over 20,000 processor hours to calculate and completed in about a week of wall clock time.

[Figure]

**Figure 1: The model domain (box) covers the southern High Plains of the United States. Blue shading indicates the Missouri and Arkansas continental river basins.**

Inputs for the study were developed from previous work of Maxwell and Condon (2016) modeling hydrology across the continental US (CONUS). The basic input data and initial conditions follow Maxwell and Condon (2016).  Inputs include slopes, soil types, vegetation, attributes of soils and geologic units, and initial pressure conditions (see Figure 2). The slopes in the x (east-west) and y (north-south) directions were derived from a digital elevation model and sink-filled to ensure the entire domain was connected. The simulations presented here use the kinematic wave approximation of the overland flow equations and

therefore require a domain with a connected drainage network, as slopes control surface flow routing. While natural depressions in the landscape exist, literature suggests that it is difficult to distinguish these natural depressions and sinks from noise within the processed DEM (eg Kenny et al 2008). Soil types were taken from the SSURGO database. Important soil attributes include porosity, permeability, specific storage

5    and van Genuchten parameters, which control saturated and unsaturated flow. Initial subsurface pressure conditions were taken from the original CONUS run, to increase spinup efficiency. The vegetation dataset was taken from the USGS land cover trends dataset (Soulard et al. 2014). Important vegetative parameters include leaf and stem area index, roughness length and displacement heights, rooting distribution parameters, and reflectance and transmittance for leaves and stems (Maxwell and Condon 2016). While

10   most inputs were drawn from Condon and Maxwell 2016, the geologic layer of that study contained features that were geologically less realistic at the scale of the High Plains. The geology of the base layer was updated for this project using local data from the US Geological Survey (USGS, 1998; USGS, 2005).

It is important to note that water management such as groundwater pumping, surface water storage and diversion, and irrigation are not included in these simulations. This means that results of the project

15   represent the system in a pre-development state not including anthropogenic impacts to the hydrology (Maxwell and Condon 2016). The only management impact represented was land use and its changes applied by setting and changing the land cover type in the model. The use of modern day vegetation is not temporally consistent with pre-industrial water management (Hurtt et al, 2011)., but unfortunately, there is limited pre-industrial data on land use. However, in In the case of this project, we are not reconstructing

20   any specific historical drought, so it is less important in addressing the research questions to match all forcings and settings to one period of time. Additionally, different crops may affect the details of drought evolution in different ways, but this is not represented here because ParFlow-CLM is not an agricultural model. Each cell is assigned 1 vegetation type and all crops are represented as the same "croplands" vegetation type. Analyzing the details of crop type and its impact is outside the scope of the present study

25   and not critical to address the study questions.

The initial conditions for the simulation were obtained from the existing continental scale simulations (Maxwell and Condon 2016) and include 4 additional years of spinup prior to this project. The pressure file was subset to the High Plains domain and the geologic layer was updated as described in the Appendix. The model was run repeatedly with water year 1984 North American Land Data Assimilation

30   System (NLDAS-2) forcing until average subsurface storage change in one year was less than 1% of precipitation (achieved after three repeated runs). Ajami et al. (2014) showed that change in subsurface storage is one of the most rigorous spinup metrics for integrated hydrologic modelling. Holding this value

below 1% of precipitation means that effects seen in numerical experiments can be interpreted as meaningful, i.e. something besides spinup noise if they exceed the threshold.

[Figure]

Figure 2: Model inputs include a) geology to characterize the bottom model layer, b) soils to characterize upper layers, c) topography for flow routing and d) vegetation for surface energy and water exchanges.

**2.3 Numerical Experiments**

A suite of synthetic drought scenarios were developed to explore the importance of precipitation, temperature and land cover change on regional drought response in the High Plains. All simulations are developed using the hourly observed NLDAS-2 historical atmospheric forcings from Water Year 1984 as a baseline (precipitation, temperature, pressure, humidity, short wave radiation, long wave radiation, wind speed). The experiments are outlined in Table 1 and Table 2 and include a baseline run, three one-perturbation experiments exploring the effect of precipitation, temperature and land cover changes in isolation, a combined experiment with both temperature and precipitation changes, and a worst-case scenario which also includes land cover changes.

Two further runs were also conducted to explore the importance of lateral flow as a mechanism within the model as part of addressing the third research question on spatial scaling and complexity. Commonly used models including VIC and SWAT do not allow lateral flow within the model, and including this process

makes the model computationally more expensive. Creating normal runs with lateral flow and free-draining runs (i.e., without lateral flow) allows exploration of how this process affects model results. To construct a free-draining run, the water table was set at the base of the domain and all overland and subsurface lateral flow processes were turned off, while vertical flow through the soil column and water table remained. (No

5   separate spinup was conducted for the free draining run, which has a lower water table than the other runs. To account for this, the free draining drought run was compared to a free draining baseline.) With these settings, ParFlow-CLM mimics a traditional land surface model as described in Maxwell and Condon (2016).

Definitions and specific implementation of each numerical experiment are shown in Table 2, and the exact

10   perturbations used are quantified in Figure 3. The baseline run and both free draining runs were conducted for one year; the drought runs were conducted for three years of repeated drought forcing to simulate a transient time period a few years into a hypothetical severe drought. Although the simulation included three years, the analysis focuses on annually averaged results in the third and last year of model simulations in order to emphasize spatial scaling and factor interactions. Temporal evolution of drought is a large topic in

15   itself and while interesting, falls outside the scope of the present study.

**Table 1 Numerical experiments were implemented through changes to the model temperature, precipitation, vegetative cover and internal settings.**

| Term | Definition | Implementation |
|------|-----------|----------------|
| Hot | Temperature above normal | Forcing input variable Temp was changed in each cell on a monthly basis. |
| Dry | Precipitation below normal | Forcing input variable APCP was changed in each cell on a monthly basis. |
| Crops | Land cover changed | Static input vegetation type was set to bare soil wherever it was normally crop or crop mosaic. |
| Free draining | Lateral flow forbidden | ParFlow keys allowing lateral flow were turned off. |

**Table 2: Model scenarios including run name and perturbations applied relative to the baseline scenario.**

| Run | Temperature | Precipitation | Vegetation | Lateral Flow |
|---|---|---|---|---|
| 1 Baseline | | | | |
| 2 Baseline free draining | | | | Off |
| 3 Hot | Hot | | | |
| 4 Dry | | Dry | | |
| 5 Crops | | | Crops set to bare soil | |
| 6 Hot and Dry | Hot | Dry | | |
| 7 Worst case | Hot | Dry | Crops set to bare soil | |
| 8 Worst case free draining | Hot | Dry | Crops set to bare soil | Off |

These experiments are synthetic drought drought scenarios that are not an exact reconstruction of any historical drought, but rather an example of severe meteorological drought based on the Dustbowl

5  drought of the 1930s. They begin with a baseline water year, then add perturbations singly and in combination, as shown in Table 1. We used NLDAS forcing from Water Year 1984 as the baseline because it is one of the most average water years in the United States in recent decades. We then increased temperature and decreased precipitation using examples drawn from a major drought in the region. Temperature and precipitation perturbations of the NLDAS forcing are based on PRISM reconstructions of

10  the Dustbowl drought (the most extreme drought in the modern historical record). Since PRISM data has a coarser resolution than the model grid, the PRISM rasters were resampled to the model grid before preparing forcing data.

Perturbations were applied at monthly timescale for each cell of the domain. To find these meteorological drought changes, a spatial map of changes was prepared for each month of water year 1934,

15  one of the worst drought years historically recorded in the region, relative to the baseline decade of 1920-1929. Months of water year 1934 were taken to represent a "drought January" "drought February" etc. Months of the 1920s were averaged to arrive at a baseline for that region at that time, a "non-drought January" "non-drought February" etc. Then these months were subtracted to create anomalies. For example, "non-drought January" temperature was subtracted from "drought January" temperature to find

the January temperature anomaly. The averaging and subtraction was done for each grid cell of the model grid, producing a spatial map for each month.

Lastly, these spatial maps were used to perturb the baseline and produce forcings for drought runs. Temperature was perturbed by adding an absolute temperature change to each cell of the hourly forcing for the relevant month. Precipitation was perturbed by multiplying each cell of the hourly forcing by a relative change for the appropriate month. Lastly, vegetation was disturbed for the crops runs by setting all crop or crop mosaic cells to bare soil (this approximation was inspired by the documented massive crop failure during the historic Dustbowl). Figure 3 shows plots of the resulting annual anomalies in temperature, precipitation and land cover which were used to drive the perturbed simulations.

In real droughts, we know that drought perturbations will not occur in isolation. For instance, changing temperature would generally accompany changes in radiation, humidity, etc. Additionally the vegetative changes could be more complex, as shifting temperature and rainfall patterns can produce a range of responses across several vegetative types. Finally, real droughts could exceed the ranges of drought simulated within the study (for example, if temperature was increased far enough, it could have a larger impact than a small precipitation change.) However, in our simulations we are isolating the primary driving factors to perturb the model and we do not include such secondary effects. This is an approximation, but the forcing is still suitable to address the research questions in this paper because the goal is not to simulate a projected climate change (with, for example, a Global Climate Model prediction) nor to predict an actual hydrologic drought.

By making a single change at a time, the analysis can attribute any differences between the baseline and the perturbed run to the single variable that was perturbed. This is an advantage of modeling studies over real world observations, as we can assess process interaction with much greater precision and detail. Changing humidity and pressure with associated temperature changes would be modifying three things at once, which limit the linearity arguments and the strength of the experiment as such. For example, Maxwell and Kollet (2008) perturbed temperature without changing other meteorological variables for a study of drought in the Little Washita watershed. Rasmussen et al (2011) also employed a simplified approach they called "pseudo-global warming", adding estimated climate perturbations in temperature, vapor mixing ratio, boundary layer height and windspeed to a forcing dataset. Markovich et al (2016) perturbed temperature alone in a similar study of climate change in California, and Pribulick et al (2016) perturbed temperature and land cover in studying the impacts of vegetation change under global warming in a Colorado watershed. While changing one meteorological variable does not fully represent the physical system, it is a documented simplification used in multiple published papers. This study uses single-factor

perturbations to the model (forcing, land cover) in a systematic way to better quantify system sensitivity and evaluate how watershed response varies between driving factors.

**2.4 Metrics of Drought Analysis**

Several metrics are applied to quantify drought impacts. First standard anomalies were calculated for all drought scenarios relative to the baseline by simply subtracting the bBaseline values. The term anomaly is used in this discussion to describe changes to hydrologic processes resulting from the perturbations applied to the simulations. The subtraction produces a simple metric of the model impact. Averaging the anomaly across the domain produced a measure of the total impact of a given factor. The single perturbation model runs allowed the calculation of average impacts ($I$) for each factor alone: $I$(temperature), $I$(precipitation) and $I$(land cover). The multi-perturbation runs allowed calculation of the impacts for the combined factors: $I$(temperature + precipitation + land cover) and $I$(temperature + precipitation).

We acknowledge that these calculated anomalies are not equivalent to published drought metrics that are often used to define hydrologic drought. These calculations were chosen for this analysis in order to specifically evaluate the research questions being addressed, through specific focus on hydrological variables such as soil moisture, water table level, runoff and ET.

The individual run impacts make it possible to assess whether impacts were linearly additive. If impacts are linearly additive, then the impacts of multi-perturbation runs (e.g. $I$(temperature + precipitation) ) should equal the sum of the composite individual perturbation runs (e.g. $I$(temperature) + $I$(precipitation)). Here we quantify the nonlinearity in the combined drought response as a percent difference between the multi-perturbation impact and the expected impact assuming linear addition.

In analyzing model results, it is important to note that these results do not constitute a direct prediction or reconstruction of a specific historical drought, therefore we do not directly validate the simulated drought impacts against observations. Comparison to data is an important and challenging step of model studies. There are observations that could be used to explore the same research questions that are addressed here; however, none of them are directly comparable to the present model because the present model is not a reconstruction of any historical drought. The findings of this model, therefore, cannot be taken as a direct prediction for central North America. Rather, their value lies in suggesting system-scale phenomena such as the nonlinear combination of factors.

[Figure]

**Figure 3: Maps of the annual drought perturbations applied to the baseline scenario including (a) changes from cropland to bare soil, (b) absolute temperature increases, and (c) relative precipitation increases.**

**3 Results and Discussion**

5   Results are grouped into four sections. The first section provides a general overview of drought impacts in their spatial and temporal context. The second section focuses on attribution of drought impacts to specific factors of temperature, precipitation and land cover. The third part quantifies how these factors combine and interact in the multi perturbation simulations, with particular focus on whether the impacts are linearly

additive. Lastly, the fourth section explores the importance of spatial scale to the predictability and linearity of impacts.

**3.1 Description of Simulated Drought Impacts**

This section characterizes hydrologic drought impacts produced by the meteorological drought simulated in the hot and dry runs. Here we focus on ET and soil moisture impacts and put the simulated anomalies in the context of seasonal and spatial variability. The next section takes a broader approach to compare more runs and more variables, with less intensive detail.

Figure 4 shows that drought impacts on ET are relatively small compared to annual changes, but impacts to soil moisture are on the order of annual fluctuations. In the baseline year, ET is low in the fall, winter and early spring but becomes large in May-September, fluctuating between approximately 10 mm/month to 80 mm/month. The annual change under meteorological drought is a hydrological drought impact to ET of -10.5 mm/month in the dry scenario, or +2.0 mm/month in the hot scenario. This average change also shows seasonal variation, with the largest impact occurring in May for the hot scenario and in Jun-Aug for the dry scenario. In the baseline year, soil moisture rises during fall, winter and spring, then decreases over the summer in response to high ET. The annual change under meteorological drought is a hydrological drought impact to soil moisture of -64 mm in the dry scenario, or -7 mm in the hot scenario. For soil moisture, the largest impact occurs in June for the hot scenario and in Sep-Nov for the dry scenario.

[Figure]

**Figure 4: Seasonal variability of ET and soil moisture. Panels a) and c) show the variables in the baseline runs, while panels b) and d) show hydrologic drought anomalies.**

Figure 5 shows a map of ET and soil moisture anomalies in August for the dry run, when the anomalies were highest. The largest ET impacts were localized in the northeast and southeast area, with the west and central parts of the domain being much less affected. The largest soil moisture impacts were localized in the northeast and southwest parts of the domain. There was high variability in ET, with small scale anomalies up to 10 times the domain average, and in soil moisture, with small scale anomalies approximately double the domain average. However, while the largest impacts were localized, most of the domain was affected with impacts to ET or soil moisture caused by the meteorological drought forcing.

(a)

[Figure]

(b)

[Figure]

**Figure 5: Spatial snapshot of drought impacts in August. Panel a) shows ET anomalies and panel b) shows soil moisture anomalies.**

This variability illustrates that the average annual temperature or precipitation anomaly is not representative of every snapshot in time, or each individual grid cell. The value of an annual average impact is to capture high level information about the entire year and the whole domain in a single number which can be used to compare impacts between runs or variables.

**3.2 Attribution of Drought Impacts**

The simulated meteorological drought conditions applied in the hot and dry simulations as well as the prescribed land cover changes produced large hydrologic impacts to runoff, ET, soil water content and water table depth. Here we compare the hydrologic impacts of different perturbation combinations to evaluate the relative importance of temperature, land cover and precipitation changes in hydrologic drought

impacts. Figure 6 shows domain averaged annual values of runoff, ET, soil water content and water table depth. Relative to the baseline scenario, most of the drought scenarios have decreased runoff and ET, depleted soil water content and lower water tables (ie, water tables that are further below the land surface), as would be expected in a drought. The exception is the Hot and the Crops runs, which have slightly higher

[revised manuscript text omitted]

These runs show the impact of land use on hydrologic drought as it develops in response to

20  meteorological drought. Figure 10 shows that the hot/dry run lowered ET by 114 mm/yr and dried soils by 70 mm on average, when drought occurred with no land use changes. However, in the worst case run where meteorological drought was combined with removal of crops, ET was lowered less (102 mm/yr) and soils

were dried more (82 mm). The directions of these changes are consistent with the impact of land use alone: removing crops without drought increased ET by 27 mm/yr, and dried soils by 22 mm. Vegetation removal exacerbates the drying effects of drought on soil because the presence of vegetation shields soil from evaporation, but since there is limited total moisture, the two effects do not add linearly and the soil

5    moisture decrease is capped at 82 mm instead of 70 mm +22 mm = 92 mm in the worst case run. Similarly, vegetation removal tends to increase ET because more is evaporating from the soil, but drought decreases ET because less water is available in general. These water limitations may relate to nonlinear behavior as discussed later in the paper. The worst case run has less impact to ET at the expense of drier soils when compared to the hot/dry run.

10        It is instructive to examine the mismatch between modelled anomalies and those predicted by linearity using spatial plots. Figure 11 maps this nonlinearity across the domain for runoff, ET, soil moisture and the water table.  The left hand column compares the hot/dry run to the individual hot and dry runs, and the right hand column compares the worst case run to the hot/dry and crops runs. The nonlinearity is calculated as a percent difference, as described in the methods. Red colors mean that the multifactor

15    impact was smaller than expected, while blue colors mean that it was larger. Gray denotes a small number of outlier grid cells. On average in the hot/dry run, ET generally decreased more than predicted by linearity, runoff decreased less, while water table and soil moisture decreased less in the center, more in the north and south.  Nonlinearity between the worst case run compared to the hot/dry run was naturally localized to the areas of land cover disturbance (i.e. where there were differences between the two). Runoff changed in

20    either direction, ET generally decreased more than predicted by single factor runs, and soil moisture and water table levels decreased less than the combination of their single factor run impacts. Importantly, the nonlinearity spans a wide range of variation, with simulated multi-perturbation impacts being up to +/- 40% from the expected values in a linear system.

        Antecedent soil moisture and water limited behaviour may explain some of the nonlinearity shown

25    in Figure 11. Figure 12 shows the modeled ET anomalies versus antecedent soil moisture for a given forcing change (color scale), providing a general illustration of these mechanisms within the model. The top panel is the dry run and the lower panel is the hot run. In this figure, all cells are plotted regardless of soil texture. This is a slight simplification as soil texture changes water retention, so two soils with the same water content but different textures might have different amounts of water available for evaporation.

30    Thus, in a few cases, the same water content might lead one cell to be energy limited but another to be water limited as the demand for evaporation exceeds the available water. This does not change the overall finding that pre-existing soil moisture partially controls the response of model cells to forcing changes. The transition in both panels at about 350 mm of soil water content indicates the importance of antecedent soil

moisture. Above this transition point, increasing temperature produces the largest increases in ET. Decreasing precipitation produces only small anomalies, because enough water is available in the wettest cells to supply the continued ET demand. Below this point, however, the system is water limited. In the presence of a temperature increase, soil water content limits the possible ET increase. These observations from the single perturbation runs support a mechanism for the nonlinearities that has been suggested in previous research (e.g. McEvoy 2016, Seniviratne 2010): when precipitation decreases, there is less available water to supply ET demand even when a rising temperature increases potential ET. Thus the simulated multi-perturbation ET is smaller (in other words, the deficit in ET is larger) than would be expected by simply combining factors.

A final possible explanation is that the nonlinearity is an artifact of the modeling. This model captures important processes of the hydrologic cycle using physically based equations, as discussed in the methods. Additionally, all runs were conducted with the same modeling environment and compared to a baseline from that model. However, we acknowledge there could be additional nonlinearities and feedbacks in the system that are not captured in our model.

[Figure]

**Figure 11: Nonlinear behaviour in both multi-perturbation runs shows spatial patterns. Panels (a) and (b) show runoff, (c) and (d) show ET, (e) and (f) show soil moisture, and (g) and (h) show water table levels. The left hand column compares the hot/dry run to the single perturbation hot and dry runs. The right hand column compares the worst case run to the crops run and the hot/dry run.**

[Figure]

**Figure 12: Antecedent soil moisture is an important control on model ET response. The ET anomaly is plotted versus antecedent soil moisture and colored by model forcing. Panel (a) shows data from the dry run (color scale is precipitation change) and panel (b) shows data from the hot run (color scale is temperature change). Each point is one model cell. Below about 350 mm of soil water content, the cells show water limited behaviour in which drought causes decreased ET depending on severity.**

**3.3 Importance of Spatial Scale**

This section examines the importance of scale in assessing these processes.  Impacts of individual factors show less variability and more dependence on model forcing at larger scales. As was discussed in section 3.1, impacts of individual factors can be more variable at small scales (Figure 8); in other words, a given forcing change can produce a wide range of impacts to ET. Figure 13shows that this variability is

greatly reduced as soon as ET anomalies are aggregated to small (HUC-8) drainage basins. By the time the ET response is aggregated to subcontinental watersheds on the scale of the Arkansas or Red river, these spatial differences cancel out and variability is greatly reduced.

[Figure]

Figure 13: Impacts of individual factors become less random at larger scales. Panel (a) shows ET anomalies of the dry run and panel (b) shows ET anomalies of the hot run, plotted against their respective forcing anomalies. Impacts begin at the individual cell level and are aggregated to a series of larger scales.

Section 3.2 showed that nonlinearity (i.e. the portion of the response not accounted for by the linear combination of the individual perturbations)  can span +/- 40% for a given grid cell (Figure 11 ) while being much less at the entire domain scale (Figure 9and Figure 10).

Figure 14 examines this nonlinearity across a variety of spatial scales, comparing the multifactor hot/dry run to the single factor hot and dry runs. The boxplots show the spread of the data plotted in the left-hand column of Figure 11, averaged at several different scales. Overall, Figure 14 shows that the nonlinearity is much decreased in moderately sized (HUC-6) river basins, and small in subcontinental river basins. (HUC-8 basins are smaller basins that nest within HUC-6 drainages. The term major basins is used here to mean the Arkansas, Red, and Missouri subcontinental basins.) This is especially important because it shows that treating the system as linear would fail to capture the most severe impacts occurring in individual grid cells. Closer inspection of Figure 14 shows that the median nonlinearity becomes more positive for runoff, and more negative for ET as scale increases. In a subcontinental basin as a whole, there is a small positive nonlinearity in runoff, meaning that the change in runoff under both temperature and precipitation increase is slightly larger than that due to the separate effects of temperature and precipitation. The interpretation that responses are more linear at larger scales focuses not on the median magnitude, but rather on the decreasing interquartile range of the boxplots at larger scales.

[Figure]

**Figure 14: Factors combine in a more linear way at larger scales. The multi-factor hot/dry run is compared to the single factor hot and dry runs. Each panel shows boxplots that characterize deviations from linearity in the hot/dry run from model cell to subcontinental scales. Panel (a) shows runoff, (b) shows ET, (c) shows soil water content and (d) shows the water table.**

The previous two figures may seem to contradict the message of earlier sections. They suggest that model responses at large scales are basically linear and predictable from the single factor runs. If the impacts of individual factors are actually straightforward and combine in a basically linear way at subcontinental scales, perhaps simplified models would be adequate to answer big-picture questions at large scales without involving the full complexity of an integrated hydrologic model. The free draining run provides insight to address this question by removing interactions between cells. Without lateral flow, a free draining run can be considered as a package of single column models, run across a spectrum of soil types, slopes and land cover.

The free draining run allows us to directly test the impact of lateral flow on the simulated drought response across spatial scales. We find that the simulated changes from the landcover change simulation vary significantly between the standard and free draining runs. As previously noted, ET tends to increase when croplands change to bare soil. Figure 15 compares ET in the worst case simulation to a baseline in both the free draining and typical configuration. Inspecting the area of crop disturbance in the lower panel of Figure 15 confirms that in the normal model configuration, ET increases in the locations where crops change to bare soil. However, in the same area of the upper panel, exactly the same forcing changes cause ET to decrease for the free draining run. This is largely explained by the nature of the free draining run. In the normal configuration, lateral flow of groundwater can sustain ET and soil moisture, similar to the results found by Maxwell and Condon (2016). However, when no lateral flow is allowed, every grid cell is restricted to the soil moisture within it.

The difference may also be due to the differences in water table between the simulations; as the free draining run did not use a constant water table or a separate spinup for the free draining run, and therefore it has a lower water table than the other runs. (Impacts were calculated by comparison to the free draining baseline run, which will partially adjust for this. However, it is still possible that a generally lower water table resulted in a water limited system and decreased ET once plant transpiration stopped.) The results of the free draining run relate to the third research question on spatial scaling and complexity by showing that even linear-seeming, large scale predictions depend on representation of lateral flows and interactions within the model.

Overall, this section shows that nonlinear kilometre-scale impacts aggregate to large scale changes which can be well predicted by linear combinations of single factor simulations. The complexity of model response depends on the scale of the area of interest, with individual km-scale grid cells being complex and aggregated subcontinental river basins response being much more simple. Responses at any scale nonetheless depend on representation of the processes and feedbacks at smaller scales. At coarse resolution,

single factor simulations may provide usable results; but as the resolution increases, accounting for the nonlinearities arising from multiple factors becomes more important. This means that coarser scale simulations such as those often run on VIC may capture big picture drought-related impacts, but may miss the finer scale local variation.

[Figure]

[Figure]

**Figure 15: Large scale impacts of crop changes depend on representation of small scale processes. Spatial maps are calculated by subtracting (a) the free draining baseline (Run 2) from the free draining worst case run (Run 8) and (b) subtracting the baseline with lateral flow (Run 1) from the worst case run (Run 7). Panel (a) shows the free draining run where ET decreases in areas of land cover disturbance. Panel (b) shows the typical configuration for comparison, where ET increases in areas of land cover disturbance.**

**Conclusions**

This study explored impacts of drought-related drivers and relevant mechanisms through a series of numerical experiments using a ParFlow-CLM model. Meteorological and landcover perturbations are based on the example of the Dustbowl drought of the 1930s individually perturbed temperature, precipitation and land cover, followed by multi-perturbation runs that combined these changes.

       Attribution of drought effects to single factors showed that lowered precipitation caused more severe effects than increased temperature, within ranges of variation typical of major droughts. All impacts are ultimately due to forcing changes, but Sections 3.2 and 3.3 showed that moisture limitations and scale also influenced responses and produced more complex behavior. Soil types and land cover had minimal effect. The complex behavior described above produced nonlinear impacts at small scales when comparing single factor to multifactor simulations, however, these impacts combined more linearly at larger scales. Although large-scale behavior appears more simple than grid scale responses, including complex small-scale processes such as lateral flow between cells was crucial to representing the large-scale responses.

       In response to the first research question on the relative importance of precipitation, temperature and land cover change in hydrologic response to drought, we conducted single-factor simulations. The results show that precipitation is relatively more important than temperature or land cover change in hydrologic response to drought. The effects of precipitation are on the order of 3 times greater than the effects of temperature or land cover change, for ranges plausibly seen in extreme droughts. This is consistent with results of prior studies including Livneh and Hoerling (2016) and Maxwell and Kollett (2008). However, the exact effects of forcing change are still highly variable and this broad result may not hold true for individual grid cells.

       Next we took these individual effects and combined them to see whether, and under what conditions, the impacts of the drought perturbations tested here (precipitation, temperature and land cover change) are linearly additive. We find that the effects can be linearly additive on a large scale for variables such as soil water content, but they are slightly less linear for variables such as ET or runoff. For individual model grid cells, the effects can be +/- 40% of the expected value.  This agrees with expected system feedbacks such as those described by Eltahir (1998), and expands on the results in Maxwell and Condon (2016).

       Finally we evaluate  how the impacts of the main drought factors and their interactions change across spatial scales. Results showed that highly variable and nonlinear impacts modelled at small scales aggregate to much less random, linear large scale behaviour, even though these large scale predictions

depend on representation of the small scale processes and interactions. This extends the work of Maxwell and Condon (2016), which showed that lateral flow affects ET thresholds within the system.

Future studies could build on the work shown here by incorporating more detail. Including surface water management such as dams or irrigation diversions would make streamflow more representative of a present-day water year on the High Plains where streams are heavily managed. Inclusion of irrigation and groundwater pumping would allow for the study of human impacts to groundwater and surface energy balance. Further updates to the available geologic datasets would allow the model's existing detailed representation of subsurface processes to be based on better-supported parameters. Future studies could also combine an individual-factor approach as done here, with a more realistic approach where meteorological variables like pressure or humidity change with temperature, or with complete future climate scenarios.

Results of this study indicate that when regional drought is occurring, local impacts may be many times smaller or larger. This matters because the most severe and costly impacts may occur in such small scale, nonlinear responses. While the exact location and size of these small scale anomalies is not predictable with a model like the present, their general existence is. Even without specific predictions, plans for responding to regional drought will be more resilient and adaptive if they anticipate small-scale, severe impacts like those modeled here.

**Competing interests**

The authors declare that they have no conflicts of interest.

**Acknowledgments**

The simulations presented here were completed using resources of the National Energy Research Scientific Computing Center (NERSC), a U.S. Department of Energy Office of Science User Facility operated under Contract No. DE-AC02-05CH11231. Funding was provided by the National Science Foundation through its Water, Sustainability and Climate (WSC) program and NSF award number 1204787. The authors thank the editor, Dr. Niko Wanders, Dr. Peter LaFollette and two anonymous reviewers for their constructive comments.

---

## Author Response (AR3)

Dear Dr Wanders,

Please find attached the final version of our manuscript "Evaluating the relative importance of precipitation, temperature and land-cover change in the hydrologic response to extreme meteorological drought conditions over the North American High Plains" with the corrections that you requested for publication in HESS.

As requested, we have updated the in-text citation and reference listing for the VIC model.

We appreciate your time and feedback on the manuscript.

Sincerely,

Annette Hein